# Coresets for Clustering Under Stochastic Noise

**Lingxiao Huang**[*]
Nanjing University

**Zhize Li**
Singapore Management University

**Nisheeth K. Vishnoi**
Yale University

**Runkai Yang**
Nanjing University

**Haoyu Zhao**
Princeton University

## Abstract

We study the problem of constructing coresets for $(k, z)$-clustering when the input dataset is corrupted by stochastic noise drawn from a known distribution. In this setting, evaluating the quality of a coreset is inherently challenging, as the true underlying dataset is unobserved. To address this, we investigate coreset construction using surrogate error metrics that are tractable and provably related to the true clustering cost. We analyze a traditional metric from prior work and introduce a new error metric that more closely aligns with the true cost. Although our metric is defined independently of the noise distribution, it enables approximation guarantees that scale with the noise level. We design a coreset construction algorithm based on this metric and show that, under mild assumptions on the data and noise, enforcing an $\varepsilon$-bound under our metric yields smaller coresets and tighter guarantees on the true clustering cost than those obtained via classical metrics. In particular, we prove that the coreset size can improve by a factor of up to $\text{poly}(k)$, where $n$ is the dataset size. Experiments on real-world datasets support our theoretical findings and demonstrate the practical advantages of our approach.

## 1 Introduction

Clustering is a foundational tool in machine learning, with applications ranging from image segmentation and customer behavior analysis to sensor data summarization [67, 80, 5, 20]. An important class of clustering problems is called $(k, z)$-CLUSTERING where, given a dataset $P \subset \mathbb{R}^d$ of $n$ points and a $k \geq 1$, the goal is to find a set $C \subset \mathbb{R}^d$ of $k$ points that minimizes the cost $\text{cost}_z(P, C) := \sum_{x \in P} d^z(x, C)$. Here $d^z(x, C) := \min \{d^z(x, c) : c \in C\}$ is the distance of $x$ to the center set $C$ and $d^z$ denotes the $z$-th power of the Euclidean distance. Examples of $(k, z)$-CLUSTERING include $k$-MEDIAN (when $z = 1$) and $k$-MEANS (when $z = 2$). In many applications, the dataset $P$ is large, and it is desirable to have a small representative subset that requires less storage and computation while allowing us to solve the underlying clustering problem. Coresets have been proposed as a solution towards this [48] – a coreset is a subset $S \subseteq P$ that approximately preserves the clustering cost for all center sets. Coresets have found further applications in sublinear models, including streaming [48, 16], distributed [8, 56], and dynamic settings [51] due to the ability to merge and compose them (see, e.g., [83, Section 3.3]).

Yet, a critical limitation of existing coreset constructions is their reliance on exact, noise-free data—a condition rarely met in practice. In practical applications, data is frequently corrupted by measurement error, transmission artifacts, or deliberate noise insertion for privacy and robustness. One reason is that the measurement process may itself introduce noise into the data, or corruption may occur during

---

[*]Alphabetical order. Correspondence to: `huanglingxiao@nju.edu.cn`, `zhizeli@smu.edu.sg`, `nisheeth.vishnoi@gmail.com`

39th Conference on Neural Information Processing Systems (NeurIPS 2025).

the recording or reporting processes [46, 3, 75, 58]. Further, noise can be introduced intentionally in data due to privacy concerns [44, 31, 43], or to ensure robustness [84, 66]. In these scenarios, instead of the true dataset $P$, one observes a noisy dataset $\widehat{P} \subset \mathbb{R}^d$. Various types of noise can emerge depending on the context: stochastic noise, adversarial noise, and noise due to missing data [7, 11, 58].

The degree of knowledge about the noise may range from complete uncertainty to full specification of its distributional parameters. Stochastic noise, in particular, has been studied in problems such as clustering [58] and regression [81], and is commonly encountered in fields like the social sciences [12, 72, 39], economics [34], and machine learning [31, 84, 66]. When the attributes of data interact weakly with each other, independent and additive stochastic noise is considered [86, 40, 65]. In such case, for each point $p \in P$ and every attribute $j \in [d]$, the observed point is $\widehat{p}_j = p_j + \xi_{p,j}$ where $\xi_{p,j}$ is drawn from a known distribution $D_j$. Various choices for $D_j$ have been explored, including Gaussian distribution [77, 49], Laplace distribution [17], uniform distribution [3, 74], and Dirac delta distribution [87]. Such noise can reflect inherent individual variability—for example, fluctuations in STEM scores across repeated exams [12, 72], where the mean and variance can be estimated by multiple exams, or from employers making decisions based on statistical information about the groups individuals belong to [34]. In settings focused on privacy or robustness, noise with known parameters might be deliberately added to data, such as the use of i.i.d. Gaussian noise in the Gaussian mechanism [30], or the introduction of i.i.d. Gaussian noise to enhance robustness against adversarial attacks in deep learning [84, 66]. The noise level might also be well known or estimable when data is collected using sensors [73, 79]; see Sections 2 and B.2 for further discussions. Numerous studies have examined the effects of modeled noise, specifically Gaussian noise [85, 58], on clustering tasks [29, 42, 58]. They analyze the relationship between the level of noise and the performance of clustering algorithms, showing that a small amount of noise can actually benefit centroid-based clustering methods [58].

Given the widespread use of coresets in clustering problems, it is crucial to explore the possibility of constructing compact coresets that remain effective in the presence of noise. The effectiveness of a coreset $S$ is usually measured via the approximation quality of $S$'s optimal center set $\mathsf{C}(S)$ in $P$:

$$r_P(\mathsf{C}(S)) := \frac{\mathrm{cost}_z(P, \mathsf{C}(S))}{\min_{C \subset \mathbb{R}^d : |C| = k} \mathrm{cost}_z(P, C)} = \frac{\mathrm{cost}_z(P, \mathsf{C}(S))}{\mathsf{OPT}_P}. \tag{1}$$

In the noise-free setting, challenges for evaluating quality $r_P(\mathsf{C}(S))$ lie in the computation of the optimal clustering cost $\mathsf{OPT}_P$, which is NP-hard. To address this, one traditionally considers a surrogate error metric that bounds the maximum (over all possible center sets) ratio [36, 24]:

$$\mathsf{Err}(S, P) := \sup_{C \subset \mathbb{R}^d : |C| = k} \frac{|\mathrm{cost}_z(S, C) - \mathrm{cost}_z(P, C)|}{\mathrm{cost}_z(S, C)} \tag{2}$$

This ratio, serving as an upper bound for $r_P(\mathsf{C}(S))$ (see Section 2), helps derive the minimum size necessary for coreset construction. There is a long and productive line of work focused on analyzing the optimal tradeoff between the coreset size $|S|$ and the associated estimation error [21, 22, 55] in the noise-free setting. However, in the noisy setting, there is an additional challenge for evaluating $r_P(\mathsf{C}(S))$: the true underlying dataset $P$ is unobservable. This leads to a natural question: *Can the traditional surrogate error* $\mathsf{Err}$ *guide coreset construction when the observed data is noisy?*

**Our contributions.** We study the problem of coreset construction for clustering in the presence of noise. Motivated by the applications mentioned above, we consider a stochastic additive noise model where each $D_j$ (as defined earlier) is parameterized by a known *noise-level* $\theta$, constrained by a bounded-moment condition (see Definition 2.1). Our first result is adapting the use of the $\mathsf{Err}$ measure for coreset construction to this noise model, along with a bound for the coreset's quality $r_P(\mathsf{C}(S))$ (Theorem 3.1). To our knowledge, this is the first result to study how coreset performance degrades under noise. We show that $\mathsf{Err}$ can significantly overestimate coreset error under noise, resulting in overly pessimistic guarantees (Section 2).

To address this limitation of using $\mathsf{Err}$, we introduce a new surrogate metric $\mathsf{Err}_\alpha$, termed *approximation-ratio* (Equation (6)). Using this new metric, we design a cluster-wise sampling algorithm (Algorithm 1) that partitions the given noisy dataset into $k$ clusters and takes a uniform sample from each cluster. We show that, under natural and necessary assumptions on $P$, this new algorithm yields smaller coresets (by a factor of up to $\mathrm{poly}(k)$) and tighter quality guarantees for $r_P(\mathsf{C}(S))$ (Theorem 3.3). These improvements hinge on distinguishing the influence of noise on the clustering cost and the location of the optimal center set $\mathsf{C}(S)$, and may be of independent interest.

Empirical results, in Section 4, support our theoretical findings even for datasets that do not meet the assumptions required by our theoretical analysis (see, e.g., Table 1), and in scenarios involving non-i.i.d. noise across dimensions (see, e.g., Table 7). Overall, our algorithm effectively generates small coresets with theoretical quality bounds, which can be integrated into clustering frameworks, enhancing robustness in noisy environments.

## 2   Noise models and metrics for coreset quality

This section formalizes the noisy data models we consider, defines the ideal (but unobservable) metric for coreset quality, and introduces two surrogate metrics-Err (standard) and $\mathsf{Err}_\alpha$ (ours). We compare their behavior under noise and motivate our new construction.

**Noise distribution and models.** Given a probability distribution $D$ on $\mathbb{R}$ with mean $\mu$ and variance $\sigma^2$, $D$ is said to satisfy *Bernstein condition* [15, 14, 33] if there exists some constant $b > 0$ such that for every integer $i \geq 3$, $\mathbb{E}_{X \sim D}\left[|X - \mu|^i\right] \leq \frac{1}{2}i!\sigma^2 b^{i-2}, i = 3, 4, \ldots$. This condition imposes an upper bound on each moment of $D$, allowing control over tail behaviors, and we consider such noise distributions in this paper. Several well-known distributions satisfy the Bernstein condition, including the Gaussian distribution, Laplace distribution, sub-Gaussian distributions, sub-exponential distributions, and so on [82]; see Section B.1 for a discussion. We begin with a probabilistic noise model that reflects real-world data corruption: with some probability, each point either remains untouched or receives independent additive noise on each coordinate.

**Definition 2.1 (Noise model I).** Let $\theta \in [0, 1]$ be a noise parameter and $D_1, \ldots, D_d$ be probability distributions on $\mathbb{R}$ with mean 0 and variance 1 that satisfy the Bernstein condition.[2] Every point $\widehat{p}$ ($p \in P$) is i.i.d. drawn from the following distribution: 1) with probability $1 - \theta$, $\widehat{p} = p$; 2) with probability $\theta$, for every $j \in [d]$, $\widehat{p}_j = p_j + \xi_{p,j}$ where $\xi_{p,j}$ is drawn from $D_j$.

When $\theta = 0$, $\widehat{P} = P$ and as $\theta \to 1$, $\widehat{P}$ becomes increasingly noisy. Note that this noise model roughly selects a fraction $\theta$ of underlying points and adds an independent noise to each feature $j \in [d]$ that is drawn from a certain distribution $D_j$.

We also consider the following model, called *noise model II*: For every $i \in [n]$ and $j \in [d]$, $\widehat{p}_{i,j} = p_{i,j} + \xi_{p,j}$, where $\xi_{p,j}$ is drawn from $D_j$, a probability distribution on $\mathbb{R}$ with mean 0, variance $\sigma^2$, and satisfying the Bernstein condition. The main difference from noise model I is that we add noise to every coordinate of each point with a changeable variance $\sigma^2$ instead of 1. Moreover, we consider more general noise models where the noise is non-independent across dimensions. For example, the covariance matrix of each noise vector $\xi_p$ is $\Sigma \in \mathbb{R}^{d \times d}$, which extends $\Sigma = \sigma^2 \cdot I_d$ when each $D_j = N(0, \sigma^2)$ under the noise model II.

Several applications mentioned in Section 1 use these noise models. For example, setting $\theta = 1$ and each $D_j$ as a Gaussian distribution corresponds to the Gaussian mechanism in differential privacy [30, 66]. The noise parameter $\theta$ is usually known in real-world scenarios. In domains like healthcare, location services, and financial analytics, adding Laplace or Gaussian noise to data points is a common strategy to protect privacy [73, 79], creating a noisy dataset, $\widehat{P}$. In such cases, $\theta$ is known in advance, removing the need to compute it during coreset construction. Other scenarios where $\theta$ is known include: 1) measurement errors in sensor data, and 2) noise in STEM exam scores (see Section B.2).

**The ideal metric.** Let $\mathcal{C}$ denote the collection of all subsets $C \subset \mathbb{R}^d$ of size $k$, termed *center sets*. For any dataset $X \subset \mathbb{R}^d$, let $\mathsf{C}(X)$ denote the optimal center set for $(k, z)$-CLUSTERING, i.e., $\mathsf{C}(X) := \arg\min_{C \in \mathcal{C}} \text{cost}_z(X, C)$. Given a dataset $P \subset \mathbb{R}^d$ of size $n$, a coreset is a weighted set $S \subset \mathbb{R}^d$ with a function $w : S \to \mathbb{R}_{\geq 0}$ such that for all $C \in \mathcal{C}$,

$$\text{cost}_z(S, C) := \sum_{x \in S} w(x) \cdot d^z(x, C) \in (1 \pm \varepsilon) \cdot \text{cost}_z(P, C). \tag{3}$$

Ideally, we would measure the effectiveness of a coreset $S$ by how well the clustering solution it yields on $S$ generalizes to the true dataset $P$. This leads to the *ideal quality measure*:

$$r_P(\mathsf{C}(S)) := \frac{\text{cost}_z(P, \mathsf{C}(S))}{\mathsf{OPT}_P},$$

where $\mathsf{OPT}_P := \min_{C \in \mathcal{C}} \text{cost}_z(P, C)$. Note that $r_P(\mathsf{C}(S)) \geq 1$, with equality if $\mathsf{C}(S) = \mathsf{C}(P)$. However, computing $\mathsf{C}(S)$ is generally NP-hard, and even estimating $r_P(\mathsf{C}(S))$ is infeasible in

---

[2]The variance of each $D_j$ can be fixed to any $t > 0$ since we can scale each point in the dataset by $\frac{1}{t}$.

the noisy setting where $P$ is unobservable. To account for approximate clustering, we define for any $\alpha \geq 1$ the set of $\alpha$-approximate center sets for $S$: $\mathcal{C}_\alpha(S) := \{C \in \mathcal{C} : r_S(C) \leq \alpha\}$, where $r_S(C) := \text{cost}_z(S,C)/\text{OPT}_S$. We then define the *worst-case quality* over this set:

$$r_P(S, \alpha) := \max_{C \in \mathcal{C}_\alpha(S)} \frac{\text{cost}_z(P,C)}{\text{OPT}_P}. \tag{4}$$

This function is monotonically increasing in $\alpha$, and $r_P(S, 1) = r_P(\mathsf{C}(S))$. We focus on settings where $\alpha$ is close to 1, such as when a PTAS is available for $(k, z)$-CLUSTERING. As discussed in Section 1, directly evaluating $r_P(S, \alpha)$ is computationally hard and, in noisy settings, fundamentally infeasible—motivating the need for surrogate metrics.

**The metric** Err. In the noise-free setting, a standard surrogate for $r_P(S, \alpha)$ is the relative error:

$$\mathsf{Err}(S, P) := \sup_{C \in \mathcal{C}} \frac{|\text{cost}_z(S,C) - \text{cost}_z(P,C)|}{\text{cost}_z(S,C)}.$$

This quantity provides a bound on how much the clustering cost on $S$ deviates from that on $P$, uniformly over all center sets. In particular, for the optimal center set $\mathsf{C}(S)$ of $S$, we have: $\frac{|\text{cost}_z(S,\mathsf{C}(S)) - \text{cost}_z(P,\mathsf{C}(S))|}{\text{cost}_z(S,\mathsf{C}(S))} \leq \mathsf{Err}(S, P)$, which implies $\text{cost}_z(P, \mathsf{C}(S)) \in \left[\frac{1}{1+\mathsf{Err}(S,P)}, (1 + \mathsf{Err}(S, P))\right] \cdot \text{cost}_z(S, \mathsf{C}(S))$. Since $\mathsf{C}(S)$ is optimal for $S$, this yields a lower bound on $\text{OPT}_P$ and extends to all $\alpha$-approximate center sets:

$$\forall C \in \mathcal{C}_\alpha(S), \quad \text{cost}_z(P, C) \leq (1 + \mathsf{Err}(S, P)) \cdot \text{cost}_z(S, C) \leq (1 + \mathsf{Err}(S, P)) \cdot \alpha \cdot \text{cost}_z(S, \mathsf{C}(S)).$$

Combining these gives:

$$r_P(S, \alpha) \leq (1 + \mathsf{Err}(S, P))^2 \cdot \alpha. \tag{5}$$

This justifies the use of $\mathsf{Err}(S, P)$ as a surrogate for $r_P(S, \alpha)$ in the noise-free setting.

In the noisy setting, however, $P$ is unobservable. A natural alternative is to compute $\mathsf{Err}(S, \widehat{P})$ instead. This raises the question: how does $\mathsf{Err}(S, \widehat{P})$ relate to the true coreset quality $r_P(S, \alpha)$? We explore this relationship and show how $\mathsf{Err}(S, \widehat{P})$ can still guide coreset construction (see Theorem 3.1).

**The new metric.** While $\mathsf{Err}(S, P)$ is a valid surrogate in the noise-free setting, its adaptation to noisy data via $\mathsf{Err}(S, \widehat{P})$ is problematic: noise inflates clustering costs on $\widehat{P}$, weakening its connection to the true quality $r_P(S, \alpha)$. To mitigate this, we introduce a new surrogate metric that compares the *relative* quality of a center set on $P$ versus $S$:

$$\mathsf{Err}_\alpha(S, P) := \sup_{C \in \mathcal{C}_\alpha(S)} \frac{r_P(C)}{r_S(C)} - 1. \tag{6}$$

Since $r_S(C) \leq \alpha$, we have $\mathsf{Err}_\alpha(S, P) \geq \sup_{C \in \mathcal{C}_\alpha(S)} \frac{r_P(C)}{\alpha} - 1$, and therefore $r_P(S, \alpha) \leq (1 + \mathsf{Err}_\alpha(S, P)) \cdot \alpha$. This bound justifies $\mathsf{Err}_\alpha$ as a surrogate for $r_P(S, \alpha)$. The metric is monotonic in $\alpha$, independent of the noise distribution, and aligns better with coreset quality under noise than $\mathsf{Err}$, which measures absolute cost deviation.

In practice, we compute $\mathsf{Err}_\alpha(S, \widehat{P})$ as a proxy for $\mathsf{Err}_\alpha(S, P)$. We analyze this in Theorem 3.3, showing that it can guide coreset construction under noise. Notably, $\mathsf{Err}_\alpha$ extends prior approximation-ratio-based coreset ideas [23, 53] to the noisy setting.

**Comparing two metrics under noise.** We illustrate the advantage of $\mathsf{Err}_\alpha$ over $\mathsf{Err}$ through a simple 1-MEANS example in $\mathbb{R}$, with $k = d = 1$, $z = 2$, and $\alpha = 1$.

Let $P = P_- \cup P_+$, where $P_-$ has $n/2$ points at $-1$ and $P_+$ has $n/2$ points at 1. The optimal center is $\mathsf{C}(P) = 0$ with $\text{OPT}_P = n$. Now consider $\widehat{P}$, generated under noise model I with $\theta = 1$ and $D_j = \mathcal{N}(0, 1)$. This adds i.i.d. Gaussian noise to each point, inflating clustering cost by roughly $2n$ for any center $c$, i.e., $\text{cost}_z(\widehat{P}, c) - \text{cost}_z(P, c) \approx 2n$. Hence, $\mathsf{Err}(\widehat{P}, P) \approx \frac{2n}{3n} = \frac{2}{3}$, yielding a bound:

$$r_P(\widehat{P}, 1) \leq (1 + \mathsf{Err}(\widehat{P}, P))^2 \lesssim \frac{25}{9}.$$

In contrast, because the noise $\xi_p$ averages out, the empirical center satisfies $\mathsf{C}(\widehat{P}) \in [-\sqrt{1/n}, \sqrt{1/n}]$ with high probability, and: $\text{cost}_z(P, \mathsf{C}(\widehat{P})) \leq n + 1$. This implies $\mathsf{Err}_1(\widehat{P}, P) \leq \frac{1}{n}$, and therefore:

$$r_P(\widehat{P}, 1) \leq 1 + \frac{1}{n}.$$

Thus, $\mathsf{Err}_\alpha$ provides a much tighter estimate of $r_P$ under noise, by compensating for the uniform cost inflation that $\mathsf{Err}$ fails to account for. We elaborate on this example and provide additional comparisons in Section B.3.

Finally, we note that our setting differs fundamentally from "robust" coreset models [38, 52, 54], which assume direct access to the clean dataset $P$. In contrast, we construct coresets directly from noisy observations $\widehat{P}$; see Section A for a detailed comparison.

## 3  Theoretical results

This section gives theoretical guarantees for coreset construction under noise. We present two algorithms for $k$-MEANS ($z = 2$) using noise model I, based on the surrogate metrics $\mathsf{Err}$ (Theorem 3.1) and $\mathsf{Err}_\alpha$ (Theorem 3.3). While both yield bounds on coreset quality, the $\mathsf{Err}_\alpha$-based method achieves smaller coresets and tighter guarantees under mild structural assumptions. We write $\mathrm{cost}$ for $\mathrm{cost}_2$ throughout. Extensions to other noise models and $(k, z)$-CLUSTERING are in Section F.

We begin with $\mathsf{Err}$-based coresets. Theorem 3.1 extends its use to noisy data and bounds $r_P(S, \alpha)$ in terms of $\theta$, $d$, and $n$. See Section C for the proof.

**Theorem 3.1** (**Coreset using** $\mathsf{Err}$). *Let $\widehat{P}$ be drawn from $P$ via noise model I with known $\theta \geq 0$. Let $\varepsilon \in (0, 1)$ and fix $\alpha \geq 1$. Let $\mathcal{A}$ be an algorithm that constructs a weighted subset $S \subset \widehat{P}$ for $k$-MEANS of size $\mathcal{A}(\varepsilon)$ and with guarantee $\mathsf{Err}(S, \widehat{P}) \leq \varepsilon$. Then with probability at least 0.9,*

$$\mathsf{Err}(S, P) \leq \varepsilon + O(\tfrac{\theta n d}{\mathsf{OPT}_P} + \sqrt{\tfrac{\theta n d}{\mathsf{OPT}_P}}) \text{ and } r_P(S, \alpha) \leq (1 + \varepsilon + O(\tfrac{\theta n d}{\mathsf{OPT}_P} + \sqrt{\tfrac{\theta n d}{\mathsf{OPT}_P}}))^2 \cdot \alpha.$$

To ensure $\mathsf{Err}(S, \widehat{P}) \leq \varepsilon$, we may use an importance sampling algorithm [10] with coreset size

$$\mathcal{A}(\varepsilon) = \widetilde{O}(\min\{k^{1.5}\varepsilon^{-2}, \, k\varepsilon^{-4}\}), \tag{7}$$

which matches the state of the art in the noise-free setting. However, the resulting bounds for $\mathsf{Err}(S, P)$ and $r_P(S, \alpha)$ incur an additive term $O(\tfrac{\theta n d}{\mathsf{OPT}_P} + \sqrt{\tfrac{\theta n d}{\mathsf{OPT}_P}})$, due to the gap $\mathsf{Err}(\widehat{P}, P)$ (see Lemma C.2). As discussed in Section 2, this gap can be overly conservative—especially when noise uniformly inflates clustering cost. In such cases, $\mathsf{Err}_\alpha(\widehat{P}, P) \ll \mathsf{Err}(\widehat{P}, P)$, as shown in our earlier example. While this inflation always occurs when $k = 1$, it may not persist for general $k$, where noise can change point-to-center assignments between $P$ and $\widehat{P}$. To address this, we introduce structural assumptions that preserve assignments under noise.

**Assumptions on data.** To theoretically separate the performance of $\mathsf{Err}$ and $\mathsf{Err}_\alpha$, we impose mild structural assumptions on the dataset to ensure that point-to-center assignments remain stable under noise. A natural but strong assumption is to posit a generative model, such as a Gaussian mixture $\sum_{\ell=1}^{k} \frac{1}{k} N(\mu_\ell, 1)$, where the means $\mu_\ell \in \mathbb{R}^d$ are well separated, e.g., $\|\mu_\ell - \mu_{\ell'}\| \geq n$ [35, 57]. This would make the assignments between $P$ and $\widehat{P}$ nearly identical. However, this is more than we require—we instead use structural assumptions that capture the relevant properties directly.

The first is *cost stability*, a widely studied notion in clustering and coreset literature [71, 6, 59, 2, 25, 10]. Let $\mathsf{OPT}_P(m)$ denotes the optimal cost of $m$-means on $P$. For $\gamma > 0$, a dataset $P$ is $\gamma$-cost-stable if

$$\frac{\mathsf{OPT}_P(k-1)}{\mathsf{OPT}_P(k)} \geq 1 + \gamma.$$

As $\gamma$ increases, the clusters are more well-separated, making assignments more robust to noise. In our setting, cost stability ensures that the assignment changes between $P$ and $\widehat{P}$ remain limited, and we specify the required value of $\gamma$ as a function of the noise level $\theta$ in Assumption 3.2. This assumption is also necessary to distinguish $\mathsf{Err}(\widehat{P}, P)$ from $\mathsf{Err}_\alpha(\widehat{P}, P)$. As shown in Appendix E.1, when cost stability is weak, the two metrics can behave similarly; e.g., in a 3-means instance with $\gamma = 1$, we find $\mathsf{Err}(\widehat{P}, P) \approx \mathsf{Err}_1(\widehat{P}, P)$.

We also assume that $P$ does not contain strong outliers. Let $P_1, \ldots, P_k$ denote the partition of $P$ induced by its optimal center set $\mathsf{C}(P)$. For each cluster $P_i$, define the average and maximum radius:

$$\bar{r}_i := \sqrt{\tfrac{1}{|P_i|} \sum_{p \in P_i} d^2(p, \mathsf{C}(P)_i)}, \quad r_i := \max_{p \in P_i} d(p, \mathsf{C}(P)_i).$$

We assume $r_i \leq 8\overline{r}_i$ for all $i$, which rules out heavy-tailed clusters and helps distinguish noise from genuine outliers. This choice of 8 is made to simplify analysis and is satisfied by real-world datasets; see Table 2.

**Assumption 3.2** (**Cost stability and limited outliers**). Given $\alpha \geq 1$ and $\theta \in [0, 1]$, assume $P$ is $\gamma$-cost-stable with

$$\gamma = O(\alpha) \cdot \left(1 + \tfrac{\theta nd \log^2(kd/\sqrt{\alpha-1})}{\mathsf{OPT}_P}\right),$$

and that $r_i \leq 8\overline{r}_i$ for all $i \in [k]$.

Under these assumptions, the following theorem gives performance guarantees for a coreset algorithm based on the $\mathsf{Err}_\alpha$ metric. The proof appears in Section D.

**Theorem 3.3** (**Coreset using the $\mathsf{Err}_\alpha$ metric**). *Let $\widehat{P}$ be an observed dataset drawn from $P$ under the noise model I with known parameter $\theta \in [0, \frac{\mathsf{OPT}_P}{nd}]$. Let $\varepsilon \in (0, 1)$ and fix $\alpha \in [1, 2]$. Under Assumption 3.2, there exists a randomized algorithm that constructs a weighted $S \subset \widehat{P}$ for $k$-MEANS of size $O(\frac{k \log k}{\varepsilon - \frac{\sqrt{\alpha-1}\theta nd}{\alpha \mathsf{OPT}_P}} + \frac{(\alpha-1)k \log k}{(\varepsilon - \frac{\sqrt{\alpha-1}\theta nd}{\alpha \mathsf{OPT}_P})^2})$ and with guarantee $\mathsf{Err}_\alpha(S, \widehat{P}) \leq \varepsilon$ with probability at least 0.99. Moreover,*

$$\mathsf{Err}_\alpha(S, P) \leq \varepsilon + O(\tfrac{\theta kd}{\mathsf{OPT}_P} + \tfrac{\sqrt{\alpha-1}}{\alpha} \cdot \tfrac{\sqrt{\theta kd \mathsf{OPT}_P} + \theta nd}{\mathsf{OPT}_P}) \ and$$

$$r_P(S, \alpha) \leq (1 + \varepsilon + O(\tfrac{\theta kd}{\mathsf{OPT}_P} + \tfrac{\sqrt{\alpha-1}}{\alpha} \cdot \tfrac{\sqrt{\theta kd \mathsf{OPT}_P} + \theta nd}{\mathsf{OPT}_P})) \cdot \alpha.$$

We consider the regime $\theta \leq \frac{\mathsf{OPT}_P}{nd}$, ensuring that noise does not dominate the clustering cost, i.e., $\mathrm{cost}(\widehat{P}, C) = O(\mathrm{cost}(P, C))$. The coreset size depends on the knowledge of $\mathsf{OPT}_P$, but we later show how to remove this dependence. In the special case $\varepsilon = 0$ and $\alpha = 1$, the bound becomes $\mathsf{Err}_\alpha(S, P) = \mathsf{Err}_1(\widehat{P}, P) = O\left(\frac{\theta kd}{\mathsf{OPT}_P}\right)$, which is a factor $k/n$ smaller than the bound $\mathsf{Err}(\widehat{P}, P) = O\left(\frac{\theta nd}{\mathsf{OPT}_P} + \sqrt{\frac{\theta nd}{\mathsf{OPT}_P}}\right)$ from Theorem 3.1. This supports the use of $\mathsf{Err}_\alpha$ under Assumption 3.2. Subsequently, we also provide an interpretation for different terms in the bounds of $\mathsf{Err}$ and $\mathsf{Err}_\alpha$.

**Comparison of coreset performance using two metrics.** We now compare the coreset size and error bounds for $r_P(S, \alpha)$ achieved by Theorem 3.1 (**CN**) and Theorem 3.3 (**CN$_\alpha$**). Let $\varepsilon = 1/\mathrm{poly}(k)$ and $\alpha = 1 + c\varepsilon$ for a constant $0 < c < 0.5$, ensuring that an $\alpha$-approximate center set can be efficiently computed [62]. Under this setting, the stability parameter in Assumption 3.2 becomes $\gamma = O(1 + \log^2(kd/\varepsilon))$, since $\theta \leq \mathsf{OPT}_P/(nd)$.

*Coreset size.* When $\frac{\sqrt{\alpha-1}\theta nd}{\alpha \mathsf{OPT}_P} < c\varepsilon/2$, the coreset size from **CN$_\alpha$** is $\widetilde{O}(k/\varepsilon)$, which improves over the size $\widetilde{O}(\min\{k^{1.5}/\varepsilon^2, \ k/\varepsilon^4\})$ of **CN** by a factor of $\sqrt{k}/\varepsilon$.

*Bound on $r_P$.* The error bound in Theorem 3.3 includes a term $O\left(\frac{\theta kd}{\mathsf{OPT}_P} + \frac{\sqrt{\alpha-1}}{\alpha} \cdot \frac{\sqrt{\theta kd \mathsf{OPT}_P} + \theta nd}{\mathsf{OPT}_P}\right)$, whose dominant component, when $n \gg \mathrm{poly}(k)$, is at most $O\left(\frac{1}{\mathrm{poly}(k)} \cdot \frac{\theta nd}{\mathsf{OPT}_P}\right)$. This is again tighter than the bound from Theorem 3.1, which scales as $O\left(\frac{\theta nd}{\mathsf{OPT}_P} + \sqrt{\frac{\theta nd}{\mathsf{OPT}_P}}\right)$, by a factor of at least $\mathrm{poly}(k)$.

For a general $\varepsilon \in (0, 1)$, we provide another example that demonstrates improved coreset performance for **CN$_\alpha$**. Let $\alpha = 1 + \varepsilon$ and $\theta = \frac{\mathsf{OPT}_P}{nd \cdot \mathrm{poly}(k)}$. Following the same analysis as above, we observe that the coreset size of **CN$_\alpha$** improves over that of **CN** by a factor of $\sqrt{k}/\varepsilon$, while the error bound improves by at least a $\mathrm{poly}(k)$ factor.

Overall, **CN$_\alpha$** yields smaller coresets and tighter theoretical guarantees for $r_P(S, \alpha)$, improving over **CN** by at least a factor of $\mathrm{poly}(k)$, owing to the more noise-aware nature of the $\mathsf{Err}_\alpha$ metric.

**Applying Theorems 3.1 and 3.3 in practice.** In practical settings, we often aim to construct a coreset such that $r_P(S, \alpha) \leq (1 + \varepsilon) \cdot \alpha$. There are two ways to achieve this:

1. Use **CN**$(\varepsilon')$ from Theorem 3.1, with $\varepsilon' = \varepsilon - O\left(\frac{\theta nd}{\mathsf{OPT}_P} + \sqrt{\frac{\theta nd}{\mathsf{OPT}_P}}\right)$.

2. Use $\mathbf{CN}_\alpha(\varepsilon_\alpha)$ from Theorem 3.3, with $\varepsilon_\alpha = \varepsilon - O\left(\frac{\theta kd}{\mathsf{OPT}_P} + \frac{\sqrt{\alpha-1}}{\alpha} \cdot \frac{\sqrt{\theta kd\mathsf{OPT}_P} + \theta nd}{\mathsf{OPT}_P}\right)$.

Both approaches require an estimate of $\mathsf{OPT}_P$, which can be obtained by computing an $O(1)$-approximate center set $\widehat{C}$ for $\widehat{P}$ and using $\mathrm{cost}(\widehat{P}, \widehat{C})$ as a proxy; see discussion in Section E.3. When $\theta \leq \mathsf{OPT}_P/(nd)$, this yields a valid approximation. Notably, both $\mathbf{CN}$ and $\mathbf{CN}_\alpha$ already compute such a $\widehat{C}$ as a first step, so no additional overhead is incurred.

A practical question is when to prefer $\mathbf{CN}_\alpha(\varepsilon_\alpha)$ over $\mathbf{CN}(\varepsilon')$. This depends on whether $P$ satisfies Assumption 3.2. Although $P$ is unobserved, the observed dataset $\widehat{P}$ often satisfies an approximate version of the assumption. We discuss how to verify this in Section E.2, allowing practitioners to choose the tighter construction when applicable. In practice, we note that $\mathbf{CN}_\alpha$ performs well even when the assumptions are violated; see Section 4.

**Key ideas in the proof of Theorem 3.1.** The goal is to relate $\mathsf{Err}(S, \widehat{P})$ to $\mathsf{Err}(S, P)$, from which the bound on $r_P(S, \alpha)$ follows via Equation (5). The Err metric satisfies a standard composition bound: $\mathsf{Err}(S, P) \leq \mathsf{Err}(S, \widehat{P}) + O(\mathsf{Err}(\widehat{P}, P))$ (Lemma C.3). Thus, it suffices to bound $\mathsf{Err}(\widehat{P}, P)$. We show $\mathsf{Err}(\widehat{P}, P) = O\left(\frac{\theta nd}{\mathsf{OPT}} + \sqrt{\frac{\theta nd}{\mathsf{OPT}}}\right)$ (Lemma C.2). This is obtained by controlling the difference:

$$\mathrm{cost}(\widehat{P}, C) - \mathrm{cost}(P, C) \leq \sum_{p \in P} \left(\|\xi_p\|_2 + \langle \xi_p, p - c\rangle\right)$$

for all $C \in \mathcal{C}$, and showing that the sum on the r.h.s. above normalized by $\mathrm{cost}(\widehat{P}, C)$ is $O\left(\frac{\theta nd}{\mathsf{OPT}} + \sqrt{\frac{\theta nd}{\mathsf{OPT}}}\right)$. The proof uses concentration from the independence of noise $\{\xi_p\}$, combined with bounded higher moments ensured by the Bernstein condition.

We note that the first term, $O\left(\frac{\theta nd}{\mathsf{OPT}_P}\right)$, is information-theoretically necessary and cannot be improved in the worst case; see Section C.2. The second term, $O\left(\sqrt{\frac{\theta nd}{\mathsf{OPT}_P}}\right)$, arises from a loose upper bound in our analysis and is not known to be tight. This term is introduced when bounding the cumulative error over all points and center sets $C$ by applying Cauchy-Schwarz:

$$\sum_{p \in P} \langle \xi_p, p - c \rangle \leq \sqrt{\left(\sum_p \|\xi_p\|_2^2\right) \cdot \left(\sum_p d^2(p, C)\right)} \leq \sqrt{\theta nd \cdot \mathrm{cost}(P, C)}.$$

This worst-case analysis assumes full alignment of all error contributions, which may be overly pessimistic. A more refined analysis could potentially tighten or remove this term by accounting for error cancellation.

**Key ideas in the proof of Theorem 3.3.** The proof has two parts: (1) designing a coreset algorithm that guarantees $\mathsf{Err}_\alpha(S, \widehat{P}) \leq \varepsilon$, and (2) bounding the gap between $\mathsf{Err}_\alpha(S, \widehat{P})$ and $\mathsf{Err}_\alpha(S, P)$.

*Coreset design.* We first construct a coreset $S$ in the noise-free setting ($\widehat{P} = P$) such that $\mathsf{Err}_\alpha(S, \widehat{P}) \leq \varepsilon$. Under Assumption 3.2, we partition $\widehat{P}$ into $k$ well-separated clusters $\widehat{P}_1, \ldots, \widehat{P}_k$, each of diameter at most $2r_i$. From each $\widehat{P}_i$, the algorithm samples a uniform subset $S_i$ of size $O\left(\frac{\log k}{\varepsilon - \Delta} + \frac{(\alpha-1)\log k}{(\varepsilon - \Delta)^2}\right)$, where $\Delta := \frac{\sqrt{\alpha-1}\theta nd}{\alpha\mathsf{OPT}_P}$. Let $S = \bigcup_i S_i$. Standard concentration bounds imply that $\mathsf{C}(S)$ is close to $\mathsf{C}(\widehat{P})$, and $\mathsf{OPT}_S \lesssim \mathsf{OPT}_{\widehat{P}}$, which yield $\mathsf{Err}_1(S, \widehat{P}) \leq \varepsilon$ (Lemma D.4).

To extend this to $\mathsf{Err}_\alpha(S, \widehat{P}) \leq \varepsilon$, we leverage a geometric property: for any $C \in \mathcal{C}_\alpha(S)$, each center $c_i$ lies within distance $O\left(\sqrt{\frac{\alpha(\mathsf{OPT} + \theta nd \log^2(kd/\sqrt{\alpha-1}))}{n_i}}\right)$ of $\mathsf{C}(S)_i$ (Lemma D.3). This follows from cost stability, which ensures consistent cluster structure for all such $C$.

Directly applying this algorithm to the noisy dataset introduces several analytical challenges. First, noise can significantly increase the diameter of each cluster $\widehat{P}_i$, weakening the closeness guarantee between $\mathsf{C}(S)$ and $\mathsf{C}(\widehat{P})$. Second, highly noisy points may shift cluster assignments across partitions $\widehat{P}i$, breaking the geometric structure required for all $C \in \mathcal{C}_\alpha(S)$. To address both issues, we eliminate extremely noisy points from $\widehat{P}$ (see Line 3 of Algorithm 1), which is the key innovation of our

---

**Algorithm 1** A coreset algorithm $\mathbf{CN}_\alpha$ using the $\mathsf{Err}_\alpha$ metric under the noise model I

---

**Input:** a noisy dataset $\widehat{P}$ derived from $P$ under noise model $I$ with $\theta > 0$, $\varepsilon \in (0,1)$, $\alpha \in [1,2]$, and an $O(1)$-approximate center set $\widehat{C} = \{\widehat{c}_1, \ldots, \widehat{c}_k\} \in \mathcal{C}$.
**Output:** a weighted set $S \subseteq \widehat{P}$

1: Decompose $\widehat{P}$ into $k$ clusters $\widehat{P}_i$ by $\widehat{C}$, where $\widehat{P}_i = \left\{ p \in \widehat{P} : \arg\min_{c \in \widehat{C}} d(p,c) = \widehat{c}_i \right\}$.

2: For each $i \in [k]$, compute $\widehat{r}_i = \sqrt{\mathrm{cost}(\widehat{P}_i, \widehat{c}_i)/|\widehat{P}_i|}$.

3: Compute $P'_i = \widehat{P}_i \cap B_i$, where ball $B_i := B(\widehat{c}_i, R_i)$ with $R_i := 3\widehat{r}_i + O(\sqrt{d} \log \frac{1+\theta k d}{\sqrt{\alpha - 1}})$.

4: For each $i \in [k]$, take a uniform sample $S_i$ of size $O(\frac{\log k}{\varepsilon - \frac{\sqrt{\alpha-1}\theta n d}{\alpha \mathsf{OPT}}} + \frac{(\alpha-1)\log k}{(\varepsilon - \frac{\sqrt{\alpha-1}\theta n d}{\alpha \mathsf{OPT}})^2})$.

5: Return $S = \bigcup_{i \in [k]} S_i$ with $w(p) = \frac{|P'_i|}{|S_i|}$ for $p \in S_i$.

---

algorithm. In contrast to the existing approaches, our method explicitly excludes high-noise points and focuses the coreset on points whose assignments are reliable. This noise-aware sampling step is critical to preserving geometric stability. Moreover, we show that the effect of the removed points on the location of $\mathsf{C}(P')$ is negligible (Lemma D.5). Together, these steps ensure that Algorithm 1 achieves the desired bound $\mathsf{Err}_1(S, \widehat{P}) \leq \varepsilon$.

*Metric bounds.* A natural approach is to compose $\mathsf{Err}_\alpha(S, \widehat{P})$ with $\mathsf{Err}_\alpha(\widehat{P}, P)$, as with the Err metric. However, this fails because a set $C \in \mathcal{C}_\alpha(S)$ may only satisfy $C \in \mathcal{C}_{\alpha(1+\varepsilon)}(\widehat{P})$, preventing direct use of $\mathsf{Err}_\alpha(\widehat{P}, P)$. To resolve this, we instead compose $\mathsf{Err}_\alpha(S, \widehat{P})$ with $\mathsf{Err}_{\alpha(1+\varepsilon)}(\widehat{P}, P)$, yielding:

$$\mathsf{Err}_\alpha(S, P) \leq \varepsilon + O(\mathsf{Err}_{\alpha(1+\varepsilon)}(\widehat{P}, P)).$$

The remaining task is to bound $\mathsf{Err}_{\alpha(1+\varepsilon)}(\widehat{P}, P)$. Assumption 3.2 allows us to analyze this quantity cluster by cluster. For simplicity, we illustrate the argument on one cluster $P_i$ and its noisy counterpart $\widehat{P}_i$. Note that $\mathsf{C}(\widehat{P}_i) = \mathsf{C}(P_i) + \frac{1}{|P_i|} \sum_{p \in P_i} \xi_p$ by the optimality condition of 1-MEANS. Then:

$$\mathrm{cost}(P_i, \mathsf{C}(\widehat{P}_i)) = \mathsf{OPT}_{P_i} + \frac{1}{|P_i|} \left\| \sum_{p \in P_i} \xi_p \right\|_2^2 \text{ and hence,}$$

$$\mathsf{Err}_1(\widehat{P}_i, P_i) = \frac{\mathrm{cost}(P_i, \mathsf{C}(\widehat{P}_i))}{\mathsf{OPT}_{P_i}} - 1 = O\left( \frac{\theta d}{\mathsf{OPT}_{P_i}} \right),$$

using standard concentration for sums of independent noise vectors (Lemma D.12). This is significantly smaller than $\mathsf{Err}(\widehat{P}_i, P_i)$, which lacks cancellation.

Aggregating across clusters and plugging into the composition bound gives:

$$\mathsf{Err}_\alpha(S, P) \leq \varepsilon + O\left( \frac{\theta k d}{\mathsf{OPT}_P} + \frac{\sqrt{\alpha(1+\varepsilon) - 1}}{\alpha} \cdot \frac{\sqrt{\theta k d \mathsf{OPT}_P} + \theta n d}{\mathsf{OPT}_P} \right).$$

This bound matches Theorem 3.3, up to an extra $(1+\varepsilon)$ factor inside the square root. This artifact can be removed by composing through $P'$ instead of $\widehat{P}$ (see Lemmas D.4 and D.5). Note that the term $O\left( \frac{\theta k d}{\mathsf{OPT}_P} \right)$ accounts for center drift due to noise. Intuitively, each optimal center in $\mathsf{C}(P)$ may shift by $O(\theta d)$ under noise, resulting in a total movement of $O(\theta k d)$. This implies that $\mathsf{C}(\widehat{P})$ forms a $(1 + \frac{\theta k d}{\mathsf{OPT}_P})$-approximate center set for $P$, yielding the corresponding error term. The final term,

$$O\left( \frac{\sqrt{\alpha - 1}}{\alpha} \cdot \frac{\sqrt{\theta k d \cdot \mathsf{OPT}_P} + \theta n d}{\mathsf{OPT}_P} \right),$$

captures the additional approximation error from using $\alpha$-approximate center sets rather than exact optima. It scales with the gap between $\alpha$ and $1$, and vanishes as $\alpha \to 1$.

Thus, we complete the proof of Theorem 3.3. Note that other noise models primarily affect the concentration bounds of the term $\left\| \sum_{p \in P_i} \xi_p \right\|_2^2$ in the analysis; see details in Section F.

# 4 Empirical results

We now evaluate the empirical performance of our proposed coreset algorithms on real-world datasets under varying noise levels and tolerance thresholds. The goal is to test whether our theoretical guarantees translate into practical improvements in coreset size, accuracy, and robustness. We also assess how well the theoretical bounds track actual performance in both clean and noisy data regimes.

**Setup.** We consider the $k$-MEANS problem on the `Adult` [61] and `Census1990` [68] datasets from the UCI Repository. Both satisfy the limited outlier assumption but exhibit small cost-stability constants $\gamma$; see Table 2. We set $k = 10$. We perturb each dataset under noise model I, using Gaussian noise with $\theta \in \{0, 0.01, 0.05, 0.25\}$, where $\theta = 0$ denotes the noise-free case. For varying tolerance levels $\varepsilon \in \{0.1, 0.15, 0.2, 0.25, 0.3\}$, we construct a coreset $S$ from $\widehat{P}$ using **CN** and $\mathbf{CN}_\alpha$. For the initialization of our algorithms, we run $k$-means++ with $\max\_\text{iter} = 5$ on $\widehat{P}$ to obtain a fast $O(1)$-approximate solution. Implementation details appear in Section G.1.

**Metrics.** For each coreset $S$, we report: (i) coreset size $|S|$, (ii) empirical approximation ratio $\widetilde{r}_S := \frac{\text{cost}(P, C_S)}{\text{cost}(P, C_P)}$, and (iii) tightness ratio $\kappa_S := \frac{\widetilde{r}_S}{u_S}$, where $u_S$ is the theoretical bound for $r_P(C_S)$ from Theorems 3.1 and 3.3. The approximation ratio $\widetilde{r}_S$ measures how well the coreset solution approximates the true clustering cost on $P$. To calculate $\widetilde{r}_S$, we obtain $C_S$ and $C_P$ by running $k$-means++ 10 times (default settings, varied seeds) on $S$ and $P$ separately and selecting the best solution for each. Letting $\widehat{\text{OPT}} = \text{cost}(\widehat{P}, C_{\widehat{P}})$, we set:

$$
u_S = \begin{cases} (1 + \varepsilon + \frac{\theta n d}{\widehat{\text{OPT}}} + \sqrt{\frac{\theta n d}{\widehat{\text{OPT}}}})^2 & (\mathbf{CN}) \\ 1 + \varepsilon + \frac{\theta k d}{\widehat{\text{OPT}}} + \frac{\theta n d}{\widehat{\text{OPT}}} & (\mathbf{CN}_\alpha). \end{cases}
$$

A value $\kappa_S \leq 1$ implies the empirical ratio is below the theoretical bound; values closer to 1 indicate tighter guarantees. All experiments are repeated 10 times, and average metrics are reported.

**Analysis.** Table 1 reports results on the `Adult` dataset across noise levels. Results on `Census1990` appear in Section G.2 and follow similar trends. In all settings, $\mathbf{CN}_\alpha$ consistently produces smaller coresets and achieves $\kappa_S$ values closer to 1 than **CN**. For example, at $\varepsilon = 0.2, \theta = 0.01$, $\mathbf{CN}_\alpha$ yields a coreset of size 1940 (82% of **CN**'s 2371), with $\kappa_S = 0.937$ vs. 0.596 for **CN** —indicating much tighter empirical bounds.

We also find that for higher tolerance levels ($\varepsilon \geq 0.2$), $\mathbf{CN}_\alpha$ often yields better empirical approximation: e.g., for $\varepsilon = 0.2, \theta = 0.01$, $\mathbf{CN}_\alpha$ attains $\widetilde{r}_S = 1.156$ vs. 1.193 for **CN**. This suggests that $\mathbf{CN}_\alpha$ outperforms even in noise-free settings, indicating its potential value beyond noisy data applications. Moreover, in the noise-free case ($\theta = 0$), empirical ratios $\widetilde{r}_S$ consistently satisfy $\widetilde{r}_S \leq 1 + \varepsilon$ for $\mathbf{CN}_\alpha$ —further validating the theoretical guarantees.

Additional results under Laplace, uniform, non-i.i.d., and noise model II appear in Section F.1. These confirm the robustness and broader applicability of $\mathbf{CN}_\alpha$ across diverse noise regimes.

**Summary.** Overall, the empirical results demonstrate that $\mathbf{CN}_\alpha$ consistently achieves tighter approximation guarantees with smaller coresets across a range of noise levels. These findings validate the practical utility of the $\text{Err}_\alpha$-based construction and suggest that it remains effective even when the theoretical assumptions (e.g., exact cost-stability) are only approximately satisfied. The robustness of $\mathbf{CN}_\alpha$ across multiple datasets and noise models underscores its suitability for integration into practical data preprocessing pipelines.

# 5 Conclusion, limitations, and future work

This paper studies the practically relevant problem of coreset construction for clustering in the presence of noise. The main contributions are two new algorithms that construct coresets with provable guarantees relative to the true (unobserved) dataset, one based on adapting the traditional surrogate metric Err, and the other introducing a new metric $\text{Err}_\alpha$. We prove that the algorithm based on $\text{Err}_\alpha$ yields smaller coreset sizes and tighter performance bounds, assuming the dataset satisfies certain natural assumptions that are necessary in worst-case scenarios. Our analysis quantifies how noise impacts clustering costs and perturbs the optimal center sets, relying on properties such as noise cancellation and the concentration of the empirical center $\mathsf{C}(\widehat{P})$. The new metric $\text{Err}_\alpha$ may

Table 1: Results of `Adult` dataset under noise model I with Gaussian noise. $|S|$ represents the coreset size, $\widetilde{r}_S$ represents its empirical approximation ratio, and $\kappa_S$ denotes the tightness ratio of its empirical approximation ratio over the theoretical bound.

(a) $\theta = 0$

| $\varepsilon$ | | 0.1 | 0.15 | 0.2 | 0.25 | 0.3 |
|---|---|---|---|---|---|---|
| $|S|$ | **CN** | 9486 | 4216 | 2371 | 1517 | 1054 |
| | $\mathbf{CN}_\alpha$ | 6445 | 3178 | 1940 | 1318 | 960 |
| $\widetilde{r}_S$ | **CN** | 1.040 | 1.080 | 1.183 | 1.278 | 1.200 |
| | $\mathbf{CN}_\alpha$ | 1.085 | 1.115 | 1.150 | 1.197 | 1.124 |
| $\kappa_S$ | **CN** | 0.859 | 0.817 | 0.821 | 0.818 | 0.710 |
| | $\mathbf{CN}_\alpha$ | 0.986 | 0.969 | 0.959 | 0.958 | 0.865 |

(b) $\theta = 0.01$

| $\varepsilon$ | | 0.1 | 0.15 | 0.2 | 0.25 | 0.3 |
|---|---|---|---|---|---|---|
| $|S|$ | **CN** | 9486 | 4216 | 2371 | 1517 | 1054 |
| | $\mathbf{CN}_\alpha$ | 6445 | 3178 | 1940 | 1320 | 960 |
| $\widetilde{r}_S$ | **CN** | 1.037 | 1.081 | 1.193 | 1.187 | 1.244 |
| | $\mathbf{CN}_\alpha$ | 1.114 | 1.069 | 1.156 | 1.154 | 1.145 |
| $\kappa_S$ | **CN** | 0.600 | 0.581 | 0.596 | 0.554 | 0.543 |
| | $\mathbf{CN}_\alpha$ | 0.984 | 0.904 | 0.937 | 0.899 | 0.859 |

(c) $\theta = 0.05$

| $\varepsilon$ | | 0.1 | 0.15 | 0.2 | 0.25 | 0.3 |
|---|---|---|---|---|---|---|
| $|S|$ | **CN** | 9486 | 4216 | 2371 | 1517 | 1054 |
| | $\mathbf{CN}_\alpha$ | 6445 | 3178 | 1940 | 1320 | 960 |
| $\widetilde{r}_S$ | **CN** | 1.027 | 1.061 | 1.217 | 1.152 | 1.200 |
| | $\mathbf{CN}_\alpha$ | 1.120 | 1.108 | 1.133 | 1.186 | 1.154 |
| $\kappa_S$ | **CN** | 0.369 | 0.359 | 0.389 | 0.348 | 0.343 |
| | $\mathbf{CN}_\alpha$ | 0.886 | 0.844 | 0.830 | 0.839 | 0.788 |

(d) $\theta = 0.25$

| $\varepsilon$ | | 0.1 | 0.15 | 0.2 | 0.25 | 0.3 |
|---|---|---|---|---|---|---|
| $|S|$ | **CN** | 9486 | 4216 | 2371 | 1517 | 1054 |
| | $\mathbf{CN}_\alpha$ | 6445 | 3178 | 1940 | 1320 | 960 |
| $\widetilde{r}_S$ | **CN** | 1.044 | 1.049 | 1.067 | 1.234 | 1.341 |
| | $\mathbf{CN}_\alpha$ | 1.123 | 1.163 | 1.173 | 1.158 | 1.222 |
| $\kappa_S$ | **CN** | 0.131 | 0.127 | 0.125 | 0.139 | 0.146 |
| | $\mathbf{CN}_\alpha$ | 0.585 | 0.590 | 0.581 | 0.559 | 0.576 |

also have independent utility beyond the noisy setting, for instance, enabling further coreset size reduction in noise-free tasks where preserving near-optimal solutions suffices—such as in regression. Empirical evaluations strongly support the theoretical findings, demonstrating robust performance across a broad range of real-world datasets (which may violate the theoretical assumptions) and under diverse noise models. These results suggest that the proposed algorithms can be readily integrated into existing coreset-based clustering pipelines to improve robustness in noisy environments.

One limitation lies in extending the use of the $\mathsf{Err}_\alpha$ metric to coreset construction in the streaming model. Although $\mathsf{Err}_\alpha$ satisfies a composition property akin to $\mathsf{Err}$, it may lack the *mergeability* property (the union of coresets is a coreset for the union of datasets), which is crucial for enabling coreset construction in streaming settings. As a result, adapting the $\mathsf{Err}_\alpha$ metric to the streaming model remains an technically challenging problem. As an initial step toward this challenge, we present a relaxed version of mergeability for the $\mathsf{Err}_\alpha$ metric in Section B.4, which may be applicable in certain scenarios.

Besides this, our work opens several promising avenues for future research. A natural next step is to explore weaker assumptions that are applicable beyond worst-case scenarios, thereby improving the generalizability of our results. Another key direction is to extend our analysis to more realistic noise models, including those with dependencies across data points, heavy-tailed noise, or adversarially structured noise. Furthermore, our work focuses on clustering problems in Euclidean spaces. Extending coreset construction under noise to general metric spaces presents an interesting direction for future research. However, without access to Euclidean coordinates, it becomes nontrivial to define additive noise on individual points. A natural alternative in such settings is to model noise additively on pairwise distances (edges) rather than on point coordinates. In addition to clustering, it would be valuable to investigate how noise affects coreset construction in other learning tasks such as regression and classification. Furthermore, studying connections between coreset constructions and other robustness notions in clustering could yield new insights. Finally, our empirical results suggest that coresets based on the $\mathsf{Err}_\alpha$ metric may outperform those based on $\mathsf{Err}$ even in the absence of noise. Characterizing the conditions under which $\mathsf{Err}_\alpha$ provides tighter guarantees than $\mathsf{Err}$ in the noise-free setting is an intriguing direction for future theoretical study.

We anticipate positive societal impact from this work by enabling more accurate and reliable data analysis pipelines. These improvements could benefit various sectors, including healthcare, finance, and technology, by enabling more robust data-driven decisions.

## Acknowledgments

LH acknowledges support from the State Key Laboratory of Novel Software Technology, the New Cornerstone Science Foundation, and NSFC Grant No. 625707396. ZL was supported by the Singapore Ministry of Education (MOE) Academic Research Fund (AcRF) Tier 1 grant. NKV was supported in part by NSF Grant CCF-2112665.

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

# Contents

# A   Related work

**Coresets for clustering (with noise).** There is a substantial body of work on coreset construction for $(k, z)$-CLUSTERING across various metric spaces, including Euclidean spaces, doubling metrics, graph shortest-path metrics, and general discrete metrics [48, 36, 37, 16, 52, 24, 21, 55]. An alternative notion called *weak coresets* has also been studied [70, 24, 27], which only require preservation of $O(1)$-approximate solutions. However, weak coresets offer no significant improvement in size compared to standard coresets in practice.

In the context of noise, most prior work focuses on identifying and removing outliers [38, 52, 54]. These approaches assume that noise manifests as identifiable outliers, and the goal is to build a robust coreset by filtering them out. By contrast, our model assumes full observation of only noisy data, with no oracle access to the clean underlying distribution. This represents a conceptual shift: instead of excluding noise, our algorithms construct coresets that directly accommodate it. This setting reflects realistic scenarios in which noise and signal cannot be reliably disentangled—necessitating coreset constructions that remain robust without preprocessing or filtering steps.

**Clustering under noise.** Clustering with noisy data has been studied extensively, typically under two paradigms. The first assumes noise is generated stochastically from a known distribution [29, 42], while the second considers adversarial noise with bounded magnitude or cardinality [7, 13, 9, 63, 64]. Our work aligns more closely with the first setting.

Broadly, this literature branches into two directions. The first investigates the *robustness* of existing clustering algorithms to noise—quantifying their performance degradation in noisy environments [28, 47, 50, 1, 4]. The second direction designs new algorithms that can tolerate or adapt to noise [26, 29, 13, 64]. A related line of work in robust clustering allows the algorithm to discard a fraction of points as outliers [18, 19, 45, 76, 41, 74, 78]. While our work shares a similar structural noise model with some previous works, the goal is significantly different. Prior works mostly ask whether a standard algorithm can efficiently solve the problem on the noisy dataset $\widehat{P}$. For instance, [4] uses Gaussian perturbations for $k$-means and shows that $k$-means converges quickly on the noisy data. In contrast, we aim to construct a coreset from $\widehat{P}$ that approximates the clustering cost on the original, unobserved dataset $P$. Here, noise is the main challenge, not a tool for tractability.

# B   Additional discussion on assumptions and error metrics

This section provides additional context and justification for the modeling choices made in the main text. We first clarify the Bernstein condition and show it is satisfied by a wide class of noise distributions. We then discuss practical scenarios where the noise variance $\theta$ is known or can be estimated. Next, we compare the standard error metric Err with the proposed $\mathrm{Err}_\alpha$, using concrete examples to highlight the tighter guarantees enabled by our approach. Finally, we present a weak mergeability property for $\mathrm{Err}_\alpha$ measure, which may be useful for streaming settings.

## B.1   Distributions satisfying the Bernstein condition

The Bernstein condition plays a central role in our theoretical analysis, as it enables control over higher-order moments and supports concentration arguments under noise. We show that this condition is satisfied by several widely used distribution families.

**Sub-Gaussian and sub-exponential distributions.** A distribution $D$ is called *sub-Gaussian* if there exists a constant $K > 0$ such that for all $t > 0$,

$$\Pr_{X \sim D}[|X| \geq t] \leq 2e^{-t^2/K^2}.$$

Similarly, $D$ is called *sub-exponential* if there exists $K > 0$ such that

$$\Pr_{X \sim D}[|X| \geq t] \leq 2e^{-t/K}.$$

Gaussian distributions are sub-Gaussian; Laplace distributions are sub-exponential.

**Lemma B.1** (**Moment bounds [82, Sections 2.5 and 2.7]**)**.** *If $D$ is sub-Gaussian, there exists a constant $K > 0$ such that for all integers $i \geq 1$,*

$$\mathbb{E}_{X \sim D}[|X|^i]^{1/i} \leq K\sqrt{i}.$$

*If $D$ is sub-exponential, then for some $K > 0$,*

$$\mathbb{E}_{X \sim D}[|X|^i]^{1/i} \leq Ki.$$

Combining Lemma B.1 with Stirling's approximation, we conclude that both sub-Gaussian and sub-exponential distributions satisfy the Bernstein condition. On the other hand, heavy-tailed distributions such as the Pareto, log-normal, and Student's $t$-distributions generally do not.

## B.2 Practical scenarios with known or estimable $\theta$

Several real-world applications provide natural access to the noise parameter $\theta$, either directly or through side-information.

**Sensor networks.** In settings such as environmental monitoring using sensor arrays (e.g., for air quality), each measurement $\widehat{p}_{i,j} = p_{i,j} + \xi_j$ is perturbed by Gaussian noise $\xi_j \sim \mathcal{N}(0, \theta)$ [60]. Although the exact $\theta$ may be unknown, sensor specifications often report an upper bound $\theta'$, which can be directly used in our algorithm without additional estimation.

**Standardized assessments.** In educational assessments (e.g., STEM exam scores), observed outcomes are noisy proxies for latent ability [32]. Here, $\widehat{p}_{i,j} = p_{i,j} + \xi_j$, with $\xi_j$ representing unknown variation. Historical data or variance analyses over large student cohorts often enable institutions to estimate $\theta$ empirically, which can then be reused to construct robust coresets for future datasets.

## B.3 Comparing Err and Err$_\alpha$

We now illustrate how Err$_\alpha$ yields significantly tighter guarantees than Err, even in simple 1D settings.

**Noisy setting.** Consider 1-MEANS in $\mathbb{R}$, with $k = d = 1$. Let $P$ consist of $n/2$ points at $-1$ and $n/2$ points at $+1$. Then $\mathsf{C}(P) = 0$ and $\mathsf{OPT} = n$. Both $-1$ and $+1$ are 2-approximate centers: $\mathrm{cost}(P, -1) = \mathrm{cost}(P, 1) = 2n$.

Let $\widehat{P}$ be the noisy version of $P$ under model I, with $\theta = 1$ and $\xi_p \sim \mathcal{N}(0, 1)$. Using the decomposition:

$$\mathrm{cost}(\widehat{P}, c) - \mathrm{cost}(P, c) = \sum_{p \in P} \|\xi_p\|_2^2 + 2 \sum_{p \in P} \langle \xi_p, p - c \rangle,$$

the first term concentrates around $n$. Bounding the second term naively:

$$2 \sum_{p \in P} \langle \xi_p, p - c \rangle \leq \sum_{p \in P} \left( \|\xi_p\|_2^2 + \|p - c\|_2^2 \right),$$

we get $\mathsf{Err}(\widehat{P}, P) \approx \frac{2n}{3n} = \frac{2}{3}$, yielding $r_P(\widehat{P}, 1) \leq (1 + \frac{2}{3})^2 = \frac{25}{9}$.

Now consider $\mathsf{Err}_\alpha(\widehat{P}, P)$. We have:

$$\mathsf{C}(\widehat{P}) = \frac{1}{n} \sum_{p \in P} \xi_p, \quad |\mathsf{C}(\widehat{P}) - \mathsf{C}(P)| = O(n^{-1/2}), \quad \Rightarrow \mathrm{cost}(P, \mathsf{C}(\widehat{P})) = n + O(1).$$

Thus, $\mathsf{Err}_1(\widehat{P}, P) \lesssim \frac{1}{n}$, giving $r_P(\widehat{P}, 1) \lesssim 1 + \frac{1}{n}$—a much sharper guarantee.

**Noise-free setting.** Consider the 1-Median problem in $\mathbb{R}$, with $n - 1$ points at 0 and one point at 1. Then $\mathsf{C}(P) = 0$ and $\mathsf{OPT} = 1$. Let $S$ have $n - 1$ points at 0 and one at $1/n$. Then:

$$\mathsf{C}(S) = 0, \quad \mathsf{OPT}_S = 1/n, \quad r_P(\mathsf{C}(S)) = 1, \quad r_S(\mathsf{C}(S)) = 1 \Rightarrow \mathsf{Err}_1(S, P) = 0.$$

However,

$$\mathsf{Err}(S, P) = \frac{|\mathrm{cost}(P, \mathsf{C}(S)) - \mathrm{cost}(S, \mathsf{C}(S))|}{\mathrm{cost}(S, \mathsf{C}(S))} = n.$$

This example underscores that even in noise-free settings, Err$_\alpha$ can yield dramatically smaller errors than Err, especially when datasets differ slightly but preserve the same optimal solution.

### B.4 Weak mergeability under $\mathsf{Err}_\alpha$

We discuss the mergeability under our proposed measure $\mathsf{Err}_\alpha$. In the ideal case, mergeability means that if two coresets $S_1, S_2$ for disjoint datasets $P_1, P_2$ each satisfy $\mathsf{Err}_\alpha(S_\ell, P_\ell) \leq \varepsilon$, then their union also satisfies

$$\mathsf{Err}_\alpha(S_1 \cup S_2, \ P_1 \cup P_2) \leq \varepsilon.$$

However, this guarantee can fail with our new metric $\mathsf{Err}_\alpha$. Consider the cases that $S_1$ and $S_2$ summarize datasets with significantly different cluster structures. In such cases, the optimal center set for $S_1 \cup S_2$ may differ substantially from the union of the individual optimal, and the combined coreset may not encode enough information to capture this emergent structure.

Mergeability remains plausible when the two coresets are structurally similar. If there exists $1 \leq \alpha' \leq \alpha$ such that

$$\mathcal{C}_{\alpha'}(S_1) \subseteq \mathcal{C}_\alpha(S_2) \quad \text{and} \quad \mathcal{C}_{\alpha'}(S_2) \subseteq \mathcal{C}_\alpha(S_1),$$

then $S_1$ and $S_2$ approximately share the same set of near-optimal solutions. This overlap enables a weaker form of mergeability under $\mathsf{Err}_\alpha$.

*Claim* B.2 (**Weak mergeability under** $\mathsf{Err}_\alpha$). Let $\varepsilon > 0$ and $\alpha, \alpha' \geq 1$ with $\alpha' \leq \alpha$. Given datasets $P_1$ and $P_2$ and weighted subsets $S_1$ and $S_2$, suppose $\mathsf{Err}_\alpha(S_\ell, P_\ell) \leq \varepsilon$ and $\mathcal{C}_{\alpha'}(S_\ell) \subseteq \mathcal{C}_\alpha(S_{3-\ell})$ for $\ell = 1, 2$. Define:

$$\kappa = \frac{\min\{\mathsf{OPT}_{S_1}, \mathsf{OPT}_{S_2}\}}{\mathsf{OPT}_{S_1 \cup S_2}}, \quad \tau = \max\left\{\frac{\mathsf{OPT}_{S_1}/\mathsf{OPT}_{P_1}}{\mathsf{OPT}_{S_2}/\mathsf{OPT}_{P_2}}, \frac{\mathsf{OPT}_{S_2}/\mathsf{OPT}_{P_2}}{\mathsf{OPT}_{S_1}/\mathsf{OPT}_{P_1}}\right\}.$$

Then:

$$\mathsf{Err}_{1+(\alpha-1)\kappa}(S_1 \cup S_2, \ P_1 \cup P_2) < \alpha' \cdot \tau \cdot (1 + \varepsilon) - 1.$$

In the limiting case where $\alpha', \alpha \to 1$ and $\tau \to 1$, this bound recovers the ideal mergeability condition:

$$\mathsf{Err}_\alpha(S_1 \cup S_2, \ P_1 \cup P_2) \leq \varepsilon.$$

*Proof of Claim B.2.* We first prove that for any center set $C \in \mathcal{C}_{1+(\alpha-1)\kappa}(S_1 \cup S_2)$, $C \in \mathcal{C}_\alpha(S_1) \cap \mathcal{C}_\alpha(S_2)$. By contradiction if there exists some $C \in \mathcal{C}_{1+(\alpha-1)\kappa}(S_1 \cup S_2)$ such that $C \notin \mathcal{C}_\alpha(S_1) \cap \mathcal{C}_\alpha(S_2)$. Then

$$
\begin{aligned}
& \mathrm{cost}(S_1 \cup S_2, C) \\
> \ & \mathsf{OPT}_{S_1 \cup S_2} + (\alpha - 1) \cdot \min\{\mathsf{OPT}_{S_1}, \mathsf{OPT}_{S_2}\} && (C \notin \mathcal{C}_\alpha(S_1) \cap \mathcal{C}_\alpha(S_2)) \\
\geq \ & \mathsf{OPT}_{S_1 \cup S_2} + (\alpha - 1)\kappa \cdot \mathsf{OPT}_{S_1 \cup S_2} && (\text{Defn. of } \kappa) \\
= \ & (1 + (\alpha - 1)\kappa) \cdot \mathsf{OPT}_{S_1 \cup S_2}.
\end{aligned}
$$

Thus, $C \notin \mathcal{C}_{1+(\alpha-1)\kappa}(S_1 \cup S_2)$, which is a contradiction.

Fix a center set $C \in \mathcal{C}_{1+(\alpha-1)\kappa}(S_1 \cup S_2)$. We now have $C \in \mathcal{C}_\alpha(S_1) \cap \mathcal{C}_\alpha(S_2)$. Since $\mathsf{Err}_\alpha(S_\ell, P_\ell) \leq \varepsilon$, we have

$$\frac{r_{P_\ell}(C) - r_{S_\ell}(C)}{r_{S_\ell}(C)} \leq \varepsilon,$$

implying that $r_{P_\ell}(C) \leq (1 + \varepsilon) r_{S_\ell}(C)$. Moreover, let $\mathsf{C}(P)$ denote the optimal center set of $S_1$ and we have $\mathsf{C}(P) \in \mathcal{C}_\alpha(S_1) \subseteq \mathcal{C}_{\alpha'}(S_2)$. Thus, we have

$$\mathsf{OPT}_{S_1 \cup S_2} \leq \mathrm{cost}(S_1 \cup S_2, \mathsf{C}(P)) \leq \mathsf{OPT}_{S_1} + \alpha' \cdot \mathsf{OPT}_{S_2} \leq \alpha'(\mathsf{OPT}_{S_1} + \mathsf{OPT}_{S_2}). \quad (8)$$

Thus,

$$\frac{r_{P_1 \cup P_2}(C) - r_{S_1 \cup S_2}(C)}{r_{S_1 \cup S_2}(C)}$$

$$= \frac{r_{P_1 \cup P_2}(C)}{r_{S_1 \cup S_2}(C)} - 1$$

$$= \frac{\text{cost}(P_1, C) + \text{cost}(P_2, C)}{\text{cost}(S_1, C) + \text{cost}(S_2, C)} \cdot \frac{\text{OPT}_{S_1 \cup S_2}}{\text{OPT}_{P_1 \cup P_2}} - 1$$

$$= \frac{\text{OPT}_{P_1} \cdot r_{P_1}(C) + \text{OPT}_{P_2} \cdot r_{P_2}(C)}{\text{OPT}_{S_1} \cdot r_{S_1}(C) + \text{OPT}_{S_2} \cdot r_{S_2}(C)} \cdot \frac{\text{OPT}_{S_1 \cup S_2}}{\text{OPT}_{P_1 \cup P_2}} - 1$$

$$\leq (1 + \varepsilon) \frac{\text{OPT}_{P_1} \cdot r_{S_1}(C) + \text{OPT}_{P_2} \cdot r_{S_2}(C)}{\text{OPT}_{S_1} \cdot r_{S_1}(C) + \text{OPT}_{S_2} \cdot r_{S_2}(C)} \cdot \frac{\text{OPT}_{S_1 \cup S_2}}{\text{OPT}_{P_1 \cup P_2}} - 1$$

$$(r_{P_\ell}(C) \leq (1 + \varepsilon) r_{S_\ell}(C))$$

$$\leq (1 + \varepsilon) \max\left\{\frac{\text{OPT}_{P_1}}{\text{OPT}_{S_1}}, \frac{\text{OPT}_{P_2}}{\text{OPT}_{S_2}}\right\} \cdot \frac{\text{OPT}_{S_1 \cup S_2}}{\text{OPT}_{P_1 \cup P_2}} - 1$$

$$\leq (1 + \varepsilon) \max\left\{\frac{\text{OPT}_{P_1}}{\text{OPT}_{S_1}}, \frac{\text{OPT}_{P_2}}{\text{OPT}_{S_2}}\right\} \cdot \frac{\alpha'(\text{OPT}_{S_1} + \text{OPT}_{S_2})}{\text{OPT}_{P_1} + \text{OPT}_{P_2}} - 1 \qquad \text{(Ineq. (8))}$$

$$\leq \alpha'(1 + \varepsilon) \cdot \max\left\{\frac{\text{OPT}_{P_1}}{\text{OPT}_{S_1}}, \frac{\text{OPT}_{P_2}}{\text{OPT}_{S_2}}\right\} \cdot \max\left\{\frac{\text{OPT}_{S_1}}{\text{OPT}_{P_1}}, \frac{\text{OPT}_{S_2}}{\text{OPT}_{P_2}}\right\} - 1$$

$$\leq \alpha'\tau(1 + \varepsilon) - 1. \qquad \text{(Defn. of } \tau)$$

$\square$

## C  Proof of Theorem 3.1: Using Err

In this section, we prove Theorem 3.1. Our proof primarily relies on bounding $\text{Err}(\widehat{P}, P)$, which we will show in Section C.1. Additionally, we provide a lower bound for $\text{Err}(\widehat{P}, P)$ in Section C.2. This helps explain each additive term in Theorem 3.1.

**Theorem C.1 (Restatement of Theorem 3.1).** *Let $\widehat{P}$ be drawn from $P$ via noise model I with known $\theta \geq 0$. Let $\varepsilon \in (0, 1)$ and fix $\alpha \geq 1$. Let $\mathcal{A}$ be an algorithm that constructs a weighted subset $S \subset \widehat{P}$ for $k$-MEANS of size $\mathcal{A}(\varepsilon)$ and with guarantee $\text{Err}(S, \widehat{P}) \leq \varepsilon$. Then with probability $p > 0.9$,*

$$\text{Err}(S, P) \leq \varepsilon + O(\tfrac{\theta n d}{\text{OPT}_P} + \sqrt{\tfrac{\theta n d}{\text{OPT}_P}}) \text{ and } r_P(S, \alpha) \leq (1 + \varepsilon + O(\tfrac{\theta n d}{\text{OPT}_P} + \sqrt{\tfrac{\theta n d}{\text{OPT}_P}}))^2 \cdot \alpha.$$

For simplicity, we use $\text{OPT}$ to denote $\text{OPT}_P$ in the following discussion. For preparation, we provide the following lemmas.

**Lemma C.2 (Bounding $\text{Err}(\widehat{P}, P)$).** *For any $\widehat{P}$ derived from an $n$-point dataset $P \in \mathbb{R}^d$ using noise model I, with parameter $\theta$ we have that with probability $p > 0.9$,*

$$\text{Err}(\widehat{P}, P) \leq O(\tfrac{\theta n d}{\text{OPT}} + \sqrt{\tfrac{\theta n d}{\text{OPT}}}). \qquad (9)$$

**Lemma C.3 (Composition property).** *Given $P, \widehat{P}, S \subset \mathbb{R}^d$, suppose $\text{Err}(S, \widehat{P}) \in (0, 1)$, then we have*

$$\text{Err}(S, P) \leq \text{Err}(S, \widehat{P}) + 2\text{Err}(\widehat{P}, P)$$

*Proof.* Suppose $\text{Err}(S, \widehat{P}) = \varepsilon$ and $\text{Err}(\widehat{P}, P) = \varepsilon'$. For every center set $C$, we have

$$|\text{cost}(\widehat{P}, C) - \text{cost}(S, C)| \leq \varepsilon \cdot \text{cost}(S, C),$$

and

$$|\text{cost}(P, C) - \text{cost}(\widehat{P}, C)| \leq \varepsilon' \cdot \text{cost}(\widehat{P}, C).$$

Combine these inequalities above, we have

$$
\begin{aligned}
&|\text{cost}(P, C) - \text{cost}(S, C)| \\
&\leq |\text{cost}(\widehat{P}, C) - \text{cost}(S, C)| + |\text{cost}(P, C) - \text{cost}(\widehat{P}, C)| \quad \text{(Triangle Inequality)} \\
&\leq \varepsilon \cdot \text{cost}(S, C) + \varepsilon' \cdot \text{cost}(\widehat{P}, C) \\
&\leq (\varepsilon + \varepsilon'(1 + \varepsilon)) \cdot \text{cost}(S, C) \\
&\leq (\varepsilon + 2\varepsilon') \cdot \text{cost}(S, C) \quad\quad\quad\quad\quad\quad\quad\quad\quad\quad\quad (\varepsilon < 1)
\end{aligned}
$$

Thus $\text{Err}(S, P) = \sup_{C \in \mathcal{C}} \frac{|\text{cost}_z(S,C) - \text{cost}_z(P,C)|}{\text{cost}_z(S,C)} \leq \text{Err}(S, \widehat{P}) + 2\text{Err}(\widehat{P}, P)$. $\qquad\square$

We are ready to prove Theorem 3.1.

*Proof of Theorem 3.1.* Suppose a weighted subset $S \subset \widehat{P}$ constructed by Algorithm $\mathcal{A}$ satisfies that $\text{Err}(S, \widehat{P}) \leq \varepsilon$. By Lemma C.2, with probability $p > 0.9$,

$$
\text{Err}(\widehat{P}, P) \leq O\left(\frac{\theta n d}{\text{OPT}} + \sqrt{\frac{\theta n d}{\text{OPT}}}\right).
$$

By Lemma C.3,

$$
\text{Err}(S, P) \leq \text{Err}(S, \widehat{P}) + 2\text{Err}(\widehat{P}, P) = \varepsilon + O\left(\frac{\theta n d}{\text{OPT}} + \sqrt{\frac{\theta n d}{\text{OPT}}}\right).
$$

Moreover, for the optimal center set $\mathsf{C}(S)$ of $S$, we have

$$
\frac{|\text{cost}(S, \mathsf{C}(S)) - \text{cost}(P, \mathsf{C}(S))|}{\text{cost}(S, \mathsf{C}(S))} \leq \text{Err}(S, P),
$$

which implies

$$
\text{cost}_z(P, \mathsf{C}(S)) \in \left[\frac{1}{1 + \text{Err}(S, P)}, (1 + \text{Err}(S, P))\right] \cdot \text{cost}_z(S, \mathsf{C}(S)).
$$

For an $\alpha$-approximate center set $C$ of $S$, we have

$$
\text{cost}(P, C) \leq (1 + \text{Err}(S, P)) \cdot \text{cost}(S, C) \leq (1 + \text{Err}(S, P)) \cdot \alpha \cdot \text{cost}(S, \mathsf{C}(S)).
$$

Combining these gives:

$$
r_P(S, \alpha) \leq (1 + \text{Err}(S, P))^2 \cdot \alpha. \tag{10}
$$

This implies that $r_P(S, \alpha) \leq (1 + \varepsilon + O(\frac{\theta n d}{\text{OPT}} + \sqrt{\frac{\theta n d}{\text{OPT}}}))^2 \cdot \alpha$. $\qquad\square$

## C.1 Proof of Lemma C.2: Bounding $\text{Err}(\widehat{P}, P)$

For each $p \in P$, recall that $\xi_p = \widehat{p} - p$ is the noise vector. We first have the following claim that bounds norms of these noise vectors.

*Claim C.4* (**Bounding $\sum_{p \in P} \|\xi_p\|_2^2$**). With probability at least 0.95, $\sum_{p \in P} \|\xi_p\|_2^2 \leq 60\theta n d$.

*Proof.* Note that for every $p \in P$,

$$
\begin{aligned}
\mathbb{E}_{\xi_p}\left[\|\xi_p\|_2^2\right] &= \theta \mathbb{E}_{\xi_{p,j} \sim D_j \sim}\left[\sum_j \xi_{p,j}^2\right] \\
&= \theta \sum_j \text{Var}_{\xi_{p,j} \sim D_j \sim}[\xi_{p,j}] - \mathbb{E}_{\xi_{p,j} \sim D_j \sim}[\xi_{p,j}]^2 \\
&= \theta d. \quad\quad\quad\quad\quad\quad\quad\quad\quad\quad\quad\quad\quad\quad \text{(Defn. of } D_j)
\end{aligned}
$$

Thus, we have

$$\mathbb{E}_{\widehat{P}}\left[\sum_{p\in P}\|\xi_p\|_2^2\right]=\theta nd. \tag{11}$$

Furthermore,

$$
\begin{aligned}
&\mathrm{Var}_{\widehat{P}}\left[\sum_{p\in P}\|\xi_p\|_2^2\right]\\
=\ &n\cdot\mathrm{Var}_{\xi_p}\left[\|\xi_p\|_2^2\right]\\
=\ &n\cdot\left(\mathbb{E}_{\xi_p}\left[\|\xi_p\|_2^4\right]-\mathbb{E}_{\xi_p}\left[\|\xi_p\|_2^2\right]^2\right)\\
\leq\ &\theta n\cdot(2d+d^2-\theta d^2)\\
\leq\ &3\theta nd^2.
\end{aligned}
\tag{12}
$$

where $\chi^2(d)$ represents the chi-square distribution with $d$ degrees of freedom, whose variance is known to be $2d$ [69].

If $\theta\leq\frac{1}{20n}$, we have that

$$\Pr_{\widehat{P}}\left[\sum_{p\in P}\|\xi_p\|_2^2=0\right]=(1-\theta)^n\geq(1-\frac{1}{20n})^n\geq0.95,$$

implying that $\Pr_{\widehat{P}}\left[\sum_{p\in P}\|\xi_p\|_2^2\leq60\theta nd\right]\geq0.95$. Otherwise, if $\theta\leq\frac{1}{n}$, we have that

$$
\begin{aligned}
&\Pr_{\widehat{P}}\left[\sum_{p\in P}\|\xi_p\|_2^2>60\theta nd\right]\\
\leq\ &\Pr_{\widehat{P}}\left[\left|\sum_{p\in P}\|\xi_p\|_2^2-\mathbb{E}_{\widehat{P}}\left[\sum_{p\in P}\|\xi_p\|_2^2\right]\right|>33\sqrt{\theta n}\sqrt{\mathrm{Var}_{\widehat{P}}\left[\sum_{p\in P}\|\xi_p\|_2^2\right]}\right] &\text{(Eq. (11) and Ineq. (12))}\\
\leq\ &\frac{1}{1000\theta n} &\text{(Chebyshev's ineq.)}\\
\leq\ &0.05. &(\theta\geq\frac{1}{20n})
\end{aligned}
$$

Thus, we complete the proof of the claim. $\qquad\square$

Now we are ready to prove Lemma C.2.

*Proof of Lemma C.2.* By Claim C.4, with probability at least 0.95, $\sum_{p\in P}\|\xi_p\|_2^2\leq60\theta nd$, which we assume happens in the following. It suffices to prove that for any center set $C\in\mathcal{C}$,

$$|\mathrm{cost}(P,C)-\mathrm{cost}(\widehat{P},C)|\leq(\frac{60\theta nd}{\mathsf{OPT}}+4\sqrt{\frac{15\theta nd}{\mathsf{OPT}}})\cdot\mathrm{cost}(\widehat{P},C). \tag{13}$$

By the triangle inequality, we know that for each $p\in P$, $|d(p,C)-d(\widehat{p},C)|\leq\|\xi_p\|_2$, which implies that $|d^2(p,C)-d^2(\widehat{p},C)|\leq\|\xi_p\|_2^2+2\|\xi_p\|_2\cdot d(p,C)$.

By a similar analysis of Lemma D.8, we have $\text{cost}(\widehat{P}, C) = O(\text{cost}(P, C))$ since we assume $\mathsf{OPT} > \theta n d$, thus we have

$$
\begin{aligned}
&\frac{|\text{cost}(P,C) - \text{cost}(\widehat{P},C)|}{\text{cost}(\widehat{P},C)} \\
&\leq \frac{\sum_{p\in P} \|\xi_p\|_2^2 + 2\|\xi_p\|_2 \cdot d(p,C)}{\text{cost}(\widehat{P},C)} \\
&\leq \frac{60\theta n d}{\mathsf{OPT}} + \frac{2\sum_{p\in P}\|\xi_p\|_2 \cdot d(p,C)}{\text{cost}(\widehat{P},C)} && \text{(by assumption)} \\
&\leq \frac{60\theta n d}{\mathsf{OPT}} + \frac{2\sqrt{(\sum_{p\in P}\|\xi_p\|_2^2) \cdot (\sum_{p\in P} d^2(p,C))}}{\text{cost}(\widehat{P},C)} && \text{(Cauchy-Schwarz)} \\
&\leq \frac{60\theta n d}{\mathsf{OPT}} + 4\sqrt{\frac{15\theta n d}{\text{cost}(P,C)}} && \text{(by assumption)} \\
&\leq \frac{60\theta n d}{\mathsf{OPT}} + 4\sqrt{\frac{15\theta n d}{\mathsf{OPT}}}, && \text{(Defn. of } \mathsf{OPT}\text{)}
\end{aligned}
$$

which completes the proof of Inequality (13). $\qquad\square$

## C.2 Lower bound of $\text{Err}(\widehat{P}, P)$

We provide the following lower bound for $\text{Err}(\widehat{P}, P)$.

*Claim* C.5 (**Lower Bound of** $\text{Err}(\widehat{P}, P)$). With probability $p > 0.8$, $\text{Err}(\widehat{P}, P) = \Omega(\frac{\theta n d}{\mathsf{OPT}_P})$.

*Proof of Claim C.5.* We show this by a worst-case example. Let $\theta = 0.1$, $n = 10000$ and $k = n - 1$. Let $P = \{p_i = 100n e_i : i \in [n]\} \subset \mathbb{R}^n$ where $e_i$ is the $i$-th unit basis in $\mathbb{R}^n$. An optimal solution $\mathsf{C}(P) = \left\{p_1, \ldots, p_{n-2}, \frac{p_{n-1}+p_n}{2}\right\}$, and hence, $\mathsf{OPT} = 10000n^2$. Note that with probability at least 0.8, the following events hold: $\sum_{p\in P}\|\xi_p\|_2^2 = \Theta(\theta n d)$ and for every $p \in P$, $\|\xi_p\|_2^2 \leq 10d\log n$. Conditioned on these events, the optimal solution $\mathsf{C}(\widehat{P})$ of $\widehat{P}$ must consist of $n - 2$ points $\widehat{p} \in \widehat{P}$ and the average of the remaining two points in $\widehat{P}$. Assume $\mathsf{C}(\widehat{P}) = \left\{\widehat{p}_1, \ldots, \widehat{p}_{n-2}, \frac{\widehat{p}_{n-1}+\widehat{p}_n}{2}\right\}$. By calculation, we obtain that

$$
\text{cost}(P, \mathsf{C}(\widehat{P})) \geq (1 + \Omega(\frac{\theta n d}{\mathsf{OPT}})),
$$

implying that $\text{Err}(\widehat{P}, P) = \Omega(\frac{\theta n d}{\mathsf{OPT}_P})$. $\qquad\square$

# D Proof of Theorem 3.3: Using $\text{Err}_\alpha$

**Theorem D.1** (**Restatement of Theorem 3.3**). *Let $\widehat{P}$ be an observed dataset drawn from $P$ under the noise model I with known parameter $\theta \in [0, \frac{\mathsf{OPT}_P}{nd}]$. Let $\varepsilon \in (0,1)$ and fix $\alpha \in [1,2]$. Under Assumption 3.2, there exists a randomized algorithm that constructs a weighted $S \subset \widehat{P}$ for $k$-MEANS of size $O(\frac{k\log k}{\varepsilon - \frac{\sqrt{\alpha-1}\theta n d}{\alpha \mathsf{OPT}_P}} + \frac{(\alpha-1)k\log k}{(\varepsilon - \frac{\sqrt{\alpha-1}\theta n d}{\alpha \mathsf{OPT}_P})^2})$ and with guarantee $\text{Err}_\alpha(S, \widehat{P}) \leq \varepsilon$ with probability at least 0.99. Moreover,*

$$
\text{Err}_\alpha(S, P) \leq \varepsilon + O\left(\frac{\theta k d}{\mathsf{OPT}_P} + \frac{\sqrt{\alpha-1}}{\alpha} \cdot \frac{\sqrt{\theta k d \mathsf{OPT}_P} + \theta n d}{\mathsf{OPT}_P}\right),
$$

*and*

$$
r_P(S, \alpha) \leq \left(1 + \varepsilon + O\left(\frac{\theta k d}{\mathsf{OPT}_P} + \frac{\sqrt{\alpha-1}}{\alpha} \cdot \frac{\sqrt{\theta k d \mathsf{OPT}_P} + \theta n d}{\mathsf{OPT}_P}\right)\right) \cdot \alpha.
$$

Recall that we use Algorithm 1 to construct such a coreset $S$. For ease of analysis, we provide the following assumption for the center set $\widehat{C}$ obtained in Line 1 of Algorithm 1. We will justify this assumption in Section D.4.

**Assumption D.2** (**Locality**). Assume that for each $\widehat{c}_i \in \widehat{C}$ ($i \in [k]$), $d(\widehat{c}_i, c_i^\star) \leq O(r_i + \sqrt{d} \log \frac{1+\theta k d}{\sqrt{\alpha-1}})$. Here, $c_i^\star$ represents the $i$-th center in $\mathsf{C}(P)$.

In the following discussion, we denote $c_i^\star$ to be the $i$-th center in $\mathsf{C}(P)$, and use $\mu(X) = \frac{\sum_{p \in X} p}{|X|}$ to denote the mean point of any dataset $X$.

For preparation, we first show that given $C \in \mathcal{C}_\alpha(S)$, every center $c_i \in C$ lies within a local ball centered at an optimal center $c_i^\star \in \mathsf{C}(P)$.

**Lemma D.3** (**Structural properties of $\mathcal{C}_\alpha(S)$**). *With high probability, for every $C \in \mathcal{C}_\alpha(S)$ and every $i \in [k]$, there exist a center $c \in C$ such that $c \in B(c_i^\star, O(\sqrt{\frac{\alpha(\mathsf{OPT}+\theta nd \log^2(\frac{kd}{\sqrt{\alpha-1}}))}{n_i}}))$. Moreover, for each point $p \in P_i, i, j \in [k], i \neq j, d(p, c_i) < d(p, c_j)$.*

Next, we provide some useful geometric properties of $S$.

**Lemma D.4** (**Properties of $S$**). *With probability $p > 0.99$,*

- *For any $i \in [k]$, $\|\mu(P_i') - \mu(S_i)\|_2^2 \leq O(\frac{\varepsilon \mathsf{OPT}_i'}{n_i})$.*

- $\mathsf{OPT}_S \leq (1 + O(\frac{\varepsilon}{\sqrt{\alpha-1}}))\mathsf{OPT}'$.

- $\mathsf{Err}_\alpha(S, \widehat{P}) \leq \varepsilon$.

Finally, we provide an upper bound for the movement distance of centers from $P'$ to $P$, which helps bound the error caused by noise.

**Lemma D.5** (**Movement of centers of $P'$**). *With high probability, for any $i \in [k]$,*

$$\|\mu(P_i) - \mu(P_i')\|_2^2 \leq O(\frac{\theta d}{n_i} + \sqrt{\alpha-1}\theta d)$$

Now we are ready to prove Theorem 3.3.

*Proof of Theorem 3.3.* Suppose we run Algorithm 1 to construct a weighted $S \subset \widehat{P}$ for $k$-MEANS of size $m$. By Lemma D.4, we directly have $\mathsf{Err}_\alpha(S, \widehat{P}) \leq \varepsilon$. It remains to prove $r_P(S, \alpha) \leq (1 + O(\varepsilon + \frac{\theta kd}{\mathsf{OPT}} + \frac{\sqrt{\alpha-1}}{\alpha} \cdot \frac{\sqrt{\theta kd \mathsf{OPT}+\theta nd}}{\mathsf{OPT}})) \cdot \alpha$.

Given an $\alpha$-approximate center set $C_\alpha = \{c_1, \ldots, c_k\}$ of $S$, by Lemma D.3, we can assume $\min_{c \in C_\alpha} d(p, c) = c_i$ for any $i \in [k]$ and any point $p \in P_i$. This implies that any point $p \in P_i$ satisfies $d(p, c_i) < d(p, c_j)$ for any $j \neq i$.

Then we have

$$
\begin{aligned}
& r_P(C_\alpha) \\
={} & \frac{\text{cost}(P, C_\alpha)}{\text{OPT}} \\
={} & \frac{\sum_{i \in k} \text{cost}(P_i, c_i) - \text{cost}(P_i, \mu(P_i))}{\text{OPT}} + 1 \\
={} & \frac{\sum_{i \in k} n_i \|c_i - \mu(P_i)\|_2^2}{\text{OPT}} + 1 \\
\leq{} & \frac{\sum_{i \in k} n_i \cdot (\|\mu(P_i) - \mu(P_i')\| + \|\mu(P_i') - \mu(S_i)\| + \|\mu(S_i) - c_i\|)^2}{\text{OPT}} + 1) \\
& \hspace{7cm} \text{(Triangle inequality)} \\
\leq{} & \frac{O(\varepsilon \text{OPT}' + \theta k d + \sqrt{\alpha - 1}\theta n d + \sqrt{(\alpha-1)\theta k d \text{OPT}}) + (\alpha - 1)\text{OPT}_S}{\text{OPT}} + 1 \\
\leq{} & \frac{O(\varepsilon \text{OPT}' + \theta k d + \sqrt{\alpha - 1}\theta n d + \sqrt{(\alpha-1)\theta k d \text{OPT}}) + (\alpha - 1)(1 + O(\frac{\varepsilon}{\sqrt{\alpha-1}}))\text{OPT}}{\text{OPT}} + 1 \\
\leq{} & \alpha \cdot \left(1 + O\left(\varepsilon + \frac{\theta k d}{\text{OPT}} + \frac{\sqrt{\alpha - 1}}{\alpha} \cdot \frac{\sqrt{\theta k d \text{OPT}} + \theta n d}{\text{OPT}}\right)\right).
\end{aligned}
$$

This completes the proof.

$\square$

## D.1  Proof of Lemma D.3: structural properties of $\mathcal{C}_\alpha(S)$

**Useful properties of $\widehat{P}$.** For preparation, we first show some properties of $\widehat{P}$. In the following, Claim D.6 and Lemma D.7 show that the clustering cost for 1-MEANS remains stable under noise perturbation. It also helps bound the cost difference in each cluster in the general $k$-MEANS setting.

*Claim* D.6 (**Statistics of cost difference**). In 1-MEANS problem, for any center $c \in \mathbb{R}^d$, we have

$$
\text{cost}(\widehat{P}, c) - \text{cost}(P, c) = \sum_{p \in P} \|\xi_p\|_2^2 + 2 \sum_{p \in P} \langle \xi_p, p - c \rangle.
$$

Moreover, we have

- $\mathbb{E}_{\widehat{P}}\left[\sum_{p \in P} \|\xi_p\|_2^2\right] = \theta n d$ and $\text{Var}_{\widehat{P}}\left[\sum_{p \in P} \|\xi_p\|_2^2\right] \leq 3\theta n d^2$;

- $\mathbb{E}_{\widehat{P}}\left[\sum_{p \in P} \langle \xi_p, p - c \rangle\right] = 0$ and $\text{Var}_{\widehat{P}}\left[\sum_{p \in P} \langle \xi_p, p - c \rangle\right] = \theta \cdot \text{cost}(P, c)$.

*Proof.* We have $\text{cost}(\widehat{P}, c) - \text{cost}(P, c) = \sum_{p \in P} d^2(\widehat{p}, c) - d^2(p, c) = \sum_{p \in P} \|\xi_p\|_2^2 + 2 \langle \xi_p, p - c \rangle$. For the first error term $\sum_{p \in P} \|\xi_p\|_2^2$, we have $\mathbb{E}_{\widehat{P}}\left[\sum_{p \in P} \|\xi_p\|_2^2\right] = \theta n d$, and

$$
\begin{aligned}
& \text{Var}_{\widehat{P}}\left[\sum_{p \in P} \|\xi_p\|_2^2\right] \\
={} & n \cdot \text{Var}_{\xi_p}\left[\|\xi_p\|_2^2\right] \\
={} & n \cdot \left(\mathbb{E}_{\xi_p}\left[\|\xi_p\|_2^4\right] - \mathbb{E}_{\xi_p}\left[\|\xi_p\|_2^2\right]^2\right) \\
\leq{} & \theta n \cdot (2d + d^2 - \theta d^2) \\
\leq{} & 3\theta n d^2.
\end{aligned}
$$

where $\chi^2(d)$ represents the chi-square distribution with $d$ degrees of freedom, whose variance is known to be $2d$ [69].

For the second error term $\sum_{p\in P}\langle\xi_p, p-c\rangle$, its expectation is obvious 0 and we have

$$
\begin{aligned}
\mathrm{Var}_{\widehat{P}}\left[\sum_{p\in P}\langle\xi_p, p-c\rangle\right] &= \sum_{p\in P}\mathrm{Var}_{\xi_p}\left[\langle\xi_p, p-c\rangle\right]\\
&= \sum_{p\in P}\mathbb{E}_{\xi_p}\left[\langle\xi_p, p-c\rangle^2\right]\\
&= \theta\cdot\sum_{p\in P}\|p-c\|_2^2\\
&= \theta\cdot\mathrm{cost}(P,c).
\end{aligned}
$$

$\square$

By Claim D.6, it suffices to prove that

$$
\sup_{c\in\mathbb{R}^d}\frac{\left|\sum_{p\in P}\|\xi_p\|_2^2 + 2\sum_{p\in P}\langle\xi_p, p-c\rangle\right|}{\mathrm{cost}(P,c)} \le O\left(\frac{\theta nd}{\mathsf{OPT}} + \sqrt{\frac{\theta d}{\mathsf{OPT}}}\right). \tag{14}
$$

By Claim D.6 and Chebyshev's inequality, we directly have that $\sum_{p\in P}\|\xi_p\|_2^2 \le 2\theta nd$ happens with probability at least 0.95. For the second error term $2\sum_{p\in P}\langle\xi_p, p-c\rangle$, we provide an upper bound by the following lemma, which strengthens the second property of Claim D.6.

**Lemma D.7 (Error bound for 1-MEANS).** *In* 1-MEANS *problem, with probability at least* $1-0.05\delta$ *for* $0 < \delta \le 1$, *the following holds*

$$
\sup_{c\in\mathbb{R}^d}\frac{|\sum_{p\in P}\langle\xi_p, p-c\rangle|}{\mathrm{cost}(P,c)} = 10\cdot\sqrt{\frac{\theta d}{\delta\mathsf{OPT}}}.
$$

*Proof.* Let $X = |\{p\in P : \xi_p \ne 0\}|$ be a random variable. When $\theta n \le 0.01\delta$, we have
$$
\Pr[X = 0] = (1-\theta)^n \ge 1 - 0.02\delta.
$$
Conditioned on the event that $X = 0$, we have
$$
\sup_{c\in\mathbb{R}^d}\frac{|\sum_{p\in P}\langle\xi_p, p-c\rangle|}{\mathrm{cost}(P,c)} = 0,
$$
which completes the proof.

In the following, we analyze the case that $\theta n > 0.01\delta$. Let $E$ be the event that $\|\sum_{p\in P}\xi_p\|_2 \le O(\sqrt{\theta nd/\delta})$. We have the following claim:
$$
\Pr[E] \ge 1 - 0.02\delta. \tag{15}
$$
Note that $E[X] = \theta n$ and hence, $\Pr[X \le 100\theta n/\delta] \ge 1 - 0.01\delta$ by Markov's inequality. Hence, we only need to prove $\Pr[E \mid X \le 100\theta n/\delta]$ for Claim 15. Also note that $\sum_{p\in P}\xi_p$ has the same distribution as $N(0, X\cdot I_d)$. Then by Theorem 3.1.1 in [82],

$$
\Pr\left[\|\sum_{p\in P}\xi_p\|_2 \le 10\sqrt{\theta nd/\delta} \mid X \le 100\theta n\right] \ge 1 - 0.01\delta.
$$

Thus, we prove (15).

In the remaining proof, we condition on event $E$. Fix an arbitrary center $c \in \mathbb{R}^d$ and let $l = \|c - \mathsf{C}(P)\|_2$. By the optimality of $\mathsf{C}(P)$, it is well known that
$$
\mathrm{cost}(P,c) = \mathrm{cost}(P,\mathsf{C}(P)) + n\cdot\|c - \mathsf{C}(P)\|_2^2 = \mathsf{OPT} + nl^2.
$$
Note that $|\sum_{p\in P}\langle\xi_p, p-c\rangle| \le |\sum_{p\in P}\langle\xi_p, p-\mathsf{C}(P)\rangle| + |\sum_{p\in P}\langle\xi_p, c-\mathsf{C}(P)\rangle|$. By Chebyshev's inequality, we have

$$
\Pr_{\widehat{P}}[|\sum_{p\in P}\langle\xi_p, p-\mathsf{C}(P)\rangle| \ge 10\sqrt{\theta\cdot\mathsf{OPT}/\delta}] \le \frac{\theta\cdot\mathrm{cost}(P,\mathsf{C}(P))}{(10\sqrt{\theta\cdot\mathsf{OPT}/\delta})^2} = 0.01\delta. \tag{16}
$$

We also have

$$
\begin{aligned}
|\sum_{p\in P}\langle\xi_p, c - \mathsf{C}(P)\rangle| \leq \quad & \|\sum_{p\in P}\xi_p\|_2\|c - \mathsf{C}(P)\| \quad \text{(Cauchy-schwarz)} \\
\leq \quad & 10l \cdot \sqrt{\theta nd/\delta}
\end{aligned}
\tag{17}
$$

Combining with Inequalities 16 and 17, we conclude that

$$
\frac{|\sum_{p\in P}\langle\xi_p, p - c\rangle|}{\mathrm{cost}(P,c)} \leq 20 \cdot \frac{l \cdot \sqrt{\theta nd/\delta}}{\mathsf{OPT} + nl^2} \leq 10 \cdot \sqrt{\frac{\theta d}{\delta\mathsf{OPT}}},
$$

happens with probability at least 0.95, which completes the proof of Lemma D.7. $\qquad\square$

Recall that we use $P_i$ to denote the data clustered to $c_i$ in $P$, where $c_i$ is the $i$-th center in the optimal cluster of $P$. We use $\widetilde{P}_i$ to denote the points in $P_i$ with the presence of the noise, and $\widetilde{c}_i$ to denote the mean of $\widetilde{P}_i$. Now we bound $\mathsf{OPT}_{\widehat{P}}$ in the general $k$-MEANS case.

**Lemma D.8 (Bounding $\mathsf{OPT}_{\widehat{P}}$).** *With probability at least 0.9, we have for all $i \in [k]$*

$$
\mathrm{cost}(\widetilde{P}_i, \widetilde{c}_i) \leq \mathsf{OPT}_i + O\left(\theta n_i dk + \sqrt{\theta dk \cdot \mathsf{OPT}_i}\right) \leq 1.5\mathsf{OPT}_i + O(\theta n_i dk).
$$

*Besides, we also have with high probability*

$$
\mathsf{OPT}_{\widehat{P}} \leq \sum_i \mathrm{cost}(\widetilde{P}_i, \widetilde{c}_i) \leq \mathsf{OPT} + O(\theta nd + \sqrt{\theta d \cdot \mathsf{OPT}})
$$

*Proof.* Similar to the 1-MEANS setting (Claim D.6), we have the following decomposition of the error

$$
\mathrm{cost}(\widetilde{P}_i, \widetilde{c}_i) - \mathrm{cost}(P_i, c_i) \leq \mathrm{cost}(\widetilde{P}_i, c_i) - \mathrm{cost}(P_i, c_i) = \sum_{p\in P_i}\|\xi_p\|_2^2 + 2\sum_{p\in P_i}\langle\xi_p, p - c_i\rangle.
$$

Besides, we have the following variance of the error terms

- $\mathbb{E}_{\widetilde{P}_i}\left[\sum_{p\in P_i}\|\xi_p\|_2^2\right] = \theta n_i d$ and $\mathrm{Var}_{\widetilde{P}_i}\left[\sum_{p\in P_i}\|\xi_p\|_2^2\right] = O(\theta n_i d^2)$;

- $\mathbb{E}_{\widetilde{P}_i}\left[\sum_{p\in P_i}\langle\xi_p, p - c_i\rangle\right] = 0$ and $\mathrm{Var}_{\widetilde{P}_i}\left[\sum_{p\in P_i}\langle\xi_p, p - c_i\rangle\right] = \theta \cdot \mathsf{OPT}_i$.

From Lemma D.7 by choosing $\delta = 1/k$, we know that with probability at least $1 - \frac{0.05}{k}$, we have

$$
\sup_{c\in\mathbb{R}^d}\frac{|\sum_{p\in P_i}\langle\xi_p, p - c_i\rangle|}{\mathrm{cost}(P_i, c_i)} = 10 \cdot \sqrt{\frac{\theta dk}{\mathsf{OPT}_i}}.
$$

Besides from Chybeshev's inequality, we also have $\sum_{p\in P_i}\|\xi_p\|^2 \leq 2\theta n_i dk$ happens with probability at least $1 - \frac{0.05}{k}$. Then we conclude the proof by applying union bound on $i \in [k]$. $\qquad\square$

As shown in [10], under the stability assumption of the dataset, for any sufficiently good approximate $k$-MEANS solution $\{c_i, \ldots, c_k\}$, centers $c_i$ are pairwise well-separated.

**Lemma D.9 (Restatement of Lemma C.1 in [10]).** *Given a $\gamma$-stable dataset $P \subset \mathbb{R}^d$ and $\alpha \leq 1 + \frac{\gamma}{2}$, let $C = \{c_1, \ldots, c_k\}$ be a $\alpha$-approximation center set. Then for any $i, j \in [k]$ with $i \neq j$, it holds that*

$$
d^2(c_i, c_j) \geq \frac{\gamma\mathsf{OPT}}{2\min(n_i, n_j)}
$$

By $\gamma = \alpha \cdot O(1 + \frac{\theta nd \log^2(\frac{kd}{\sqrt{\alpha-1}})}{\mathsf{OPT}})$, we have $d^2(c_i, c_j) \geq \alpha \cdot O(\frac{\mathsf{OPT}+\theta nd \log^2(\frac{kd}{\sqrt{\alpha-1}})}{\min(n_i, n_j)})$.

Now we are ready to prove Lemma D.3

*Proof of Lemma D.3.* For ease of analysis, we prove the structural property of $P'$, and the property of $S$ naturally holds, as $S$ is obtained by uniform sampling from $P'$. We first show that with high probability, for all $i$, $|P'_i| > 0.5n_i$. Note that for any $\widehat{p} \in \widetilde{P}_i$, $\widehat{p} \notin P'_i$ implies that $\|\xi_p\| > R_i - d(\widehat{p}, \widehat{c}_i) > O(\sqrt{d} \log \frac{1+\theta kd}{\sqrt{\alpha-1}})$.

Note that since $\xi_{p,j}$ satisfies Bernstein condition for any $j \in [d]$ if $\xi_p \neq 0$, we have

$$\Pr\left[|\xi_{p,j}| \geq t | \xi_p \neq 0\right] \leq 2\exp\left(-\frac{t^2}{2(1+bt)}\right), \forall j \in [d], t > 0.$$

Thus we have for large enough $t$ (larger than $b$),

$$\Pr\left[|\xi_{p,j}| \geq t | \xi_p \neq 0\right] \leq 2\exp\left(-\Omega(t)\right).$$

It follows that

$$\Pr\left[\|\xi_p\| \geq t | \xi_p \neq 0\right] \leq d \cdot \Pr\left[|\xi_{p,j}| \geq \frac{t}{\sqrt{d}} | \xi_p \neq 0\right] \leq 2d\exp\left(-\Omega(\frac{t}{\sqrt{d}})\right).$$

Thus for any single point $p \in P_i$, with probability at most $o(1)$, $\widehat{p} \notin B(c_i^\star, R_i)$. Then we have $|P'_i| \geq 0.5n_i$ with high probability by Chernoff bound.

Note that for any points $\widehat{p}$ in $P'_i$,

$$
\begin{aligned}
d(\widehat{p}, c_i^\star) &\leq & d(\widehat{p}, \widehat{c}_i) + d(c_i^\star, \widehat{c}_i) & \\
&\leq & R_i + O(r_i) + O(\sqrt{d}\log\frac{1+\theta kd}{\sqrt{\alpha-1}}) & \text{(Assumption D.2)} \\
&\leq & O(\overline{r}_i) + O(\sqrt{d}\log\frac{1+\theta kd}{\sqrt{\alpha-1}}) & (r_i \leq 8\overline{r}_i) \\
&\leq & O(\sqrt{\frac{\mathsf{OPT}}{n_i}}) + O(\sqrt{d}\log\frac{1+\theta kd}{\sqrt{\alpha-1}}) & \\
&\leq & O(\sqrt{\frac{\alpha(\mathsf{OPT}+\theta nd\log^2(\frac{kd}{\sqrt{\alpha-1}}))}{n_i}}). &
\end{aligned}
$$

Suppose there exists $i \in [k]$ such that $C \cap B(c_i^\star, O(\sqrt{\frac{\alpha(\mathsf{OPT}+\theta nd\log^2(\frac{kd}{\sqrt{\alpha-1}}))}{n_i}})) = \emptyset$. Then

$$
\begin{aligned}
\mathrm{cost}(P', C) &\geq & \mathrm{cost}(P'_i, C) \\
&\geq & \sum_{p \in P'_i} d^2(C, c_i^\star) - d^2(p, c_i^\star) \\
&\geq & \frac{n_i}{2} \cdot O(\frac{\alpha(\mathsf{OPT}+\theta nd)}{n_i}) \\
&\geq & \alpha \cdot O(\mathsf{OPT}+\theta nd).
\end{aligned}
$$

By Lemma D.8, we have with high probability

$$\mathsf{OPT}' \leq O(\mathsf{OPT}+\theta nd).$$

Then we have

$$\frac{\mathrm{cost}(P', C)}{\mathsf{OPT}'} \geq \frac{\mathrm{cost}(P'_i, C)}{\mathsf{OPT}'} \geq \frac{\alpha \cdot O(\mathsf{OPT}+\theta nd)}{\mathsf{OPT}'} \geq \alpha.$$

Thus we prove it by contradiction. This directly implies that for any $i \in [k]$, there exists exactly one center $c_i \in C$ satisfying $c \in B(c_i^\star, O(\sqrt{\frac{\alpha(\mathsf{OPT}+\theta nd\log^2(\frac{kd}{\sqrt{\alpha-1}}))}{n_i}}))$. Moreover, by Lemma D.9, for any $j \in [k], j \neq i$,

$$d^2(c_i, c_j) \geq \alpha \cdot O(\frac{\mathsf{OPT}+\theta nd\log^2(\frac{kd}{\sqrt{\alpha-1}})}{\min(n_i, n_j)}).$$

Then we have

$$d(p, c_i) \leq r_i + d(c_i, c_i^\star) \leq O(d(c_i, c_j)).$$

Thus for any point $p \in P_i$,

$$d(p, c_i) < d(c_i, c_j) - d(p, c_i) < d(p, c_j),$$

which completes the proof. $\qquad\square$

## D.2 Proof of Lemma D.4: properties of $S$

**Lemma D.10** (**Movement of coreset centers**). *Let $P$ be a set of $n$ points and let $S$ be a set of $m$ points sampled uniformly at random with replacement from $P$, let $\mathsf{OPT} = \sum_{x \in P} \|x - \mu(P)\|_2^2$, then*

$$\mathbb{E}(\|\mu(P) - \mu(S)\|_2^2) = \frac{\mathsf{OPT}}{nm}.$$

*Proof.* For each $p \in P$, let $d_p := p - \mu(P)$. Obviously $\sum_{p \in P} d_p = 0$.

Thus

$$
\begin{aligned}
\mathbb{E}\left(\|\mu(P) - \mu(S)\|_2^2\right) &= \mathbb{E}\left(\left\|\frac{\sum_{p \in S}(p - \mu(P))}{m}\right\|_2^2\right) \\
&= \frac{1}{m^2}\mathbb{E}\left(\|\sum_{p \in S} d_p\|_2^2\right) \\
&= \frac{1}{m^2}\mathbb{E}\left(\sum_{p \in S}\|d_p\|_2^2 + 2\sum_{p_i, p_j \in S, i < j}\langle d_{p_i}, d_{p_j}\rangle\right) \\
&= \frac{\sum_{p \in P}\|d_p\|_2^2}{mn} + \frac{1}{\mathcal{P}(m,n)}\sum_{p \in P}\sum_{q \in P} <d_p, d_q> \\
&= \frac{\mathsf{OPT}}{nm} + \frac{1}{\mathcal{P}(m,n)}\left\langle\sum_{p \in P} d_p, \sum_{q \in P} d_q\right\rangle \\
&= \frac{\mathsf{OPT}}{nm},
\end{aligned}
$$

where $\mathcal{P}(m, n)$ is a polynomial in $m$ and $n$. $\qquad\square$

Now we prove Lemma D.4.

*Proof of Lemma D.4.* For any $i \in [k]$, by Lemma D.10, $\mathbb{E}(\|\mu(P_i') - \mu(S_i)\|^2) = \frac{\mathsf{OPT}_i'}{|P_i'| \cdot m_i}$.

Note that $|P_i'| > 0.5n_i$ and $m_i > O(\frac{1}{\varepsilon})$. Let $\varepsilon' = \varepsilon - \frac{\sqrt{\alpha - 1}\theta nd}{\mathsf{OPT}}$. Then by Hoeffding bound, with $p \geq 0.99$,

$$\|\mu(P_i') - \mu(S_i)\|^2 \leq O\left(\frac{\log k\mathsf{OPT}_i'}{|P_i'| \cdot m_i}\right) = O\left(\frac{\varepsilon'\mathsf{OPT}_i'}{n_i}\right).$$

Next we discuss the upper bound of $\mathsf{OPT}_S$. Let $C' = c_1', \ldots, c_k'$ be the optimal center set for $P'$ with cost $\mathsf{OPT}'$. Given an $\alpha$-approximate center set $C_\alpha = \{c_1, \ldots, c_k\}$, For each point $p \in S_i$, $w_p = \frac{|P_i'|}{|S_i|}$, thus

$$\mathbb{E}[\text{cost}(S_i, C')] = \frac{m_i}{|P_i'|}\sum_{x \in P_i'} d^2(x, C') \cdot w_x = \mathsf{OPT}_i'.$$

Then by Chernoff bound, we have for any $t > 0$ and $i \in [k]$,

$$\Pr\left[\left|\text{cost}(S_i, c_i') - \text{OPT}_i'\right| \geq t\text{OPT}_i'\right] \leq \exp\left(-\Omega(\frac{t^2}{m_i})\right).$$

Set $t = O\left(\sqrt{\frac{k \log k}{m}}\right)$ and by union bound, it follows that

$$\Pr\left[\left|\text{cost}(S, C') - \text{OPT}'\right| \geq t\text{OPT}'\right] \leq 0.01$$

Thus, with probability $> 0.99$,

$$\text{OPT}_S \leq \text{cost}(S, C') \leq \left(1 + O\left(\sqrt{\frac{k \log k}{m}}\right)\right)\text{OPT}' \leq (1 + O(\frac{\varepsilon}{\sqrt{\alpha - 1}}))\text{OPT}'.$$

Having the above properties, we conclude that

$$\begin{aligned}
\text{Err}_\alpha(S, \widehat{P}) &\leq \frac{\text{cost}(\widehat{P}, C_\alpha)}{\alpha\text{OPT}_{\widehat{P}}} - 1 \\
&\leq \frac{\sum_i \text{cost}(\widetilde{P}_i, c_i) - \text{cost}(\widetilde{P}_i, c_i')}{\alpha\text{OPT}'} - (1 - \frac{1}{\alpha}) \\
&\leq \frac{\sum_i n_i \|\mu(\widetilde{P}_i) - c_i\|_2^2}{\alpha\text{OPT}'} - (1 - \frac{1}{\alpha}) \\
&\leq \frac{\sum_i n_i(\|\mu(\widetilde{P}_i) - \mu(P_i')\| + \|\mu(P_i') - \mu(S_i)\| + \|\mu(S_i) - c_i\|)^2}{\alpha\text{OPT}'} - (1 - \frac{1}{\alpha})
\end{aligned}$$

(Triangle inequality)

$$\begin{aligned}
&\leq \frac{O(\varepsilon'\text{OPT}' + \sqrt{\alpha - 1}\theta nd) + (\alpha - 1)\text{OPT}_S}{\alpha\text{OPT}'} - (1 - \frac{1}{\alpha}) \\
&\leq O\left(\frac{\varepsilon}{\alpha} + \frac{(\alpha - 1)(\text{OPT}_S - \text{OPT}')}{\alpha\text{OPT}'}\right) \\
&\leq O\left(\frac{\varepsilon}{\alpha} + \frac{\sqrt{\alpha - 1}\varepsilon}{\alpha}\right) \\
&\leq \varepsilon
\end{aligned}$$

(Lemma D.4)

The last inequality is ensured by fixing a sufficient large constant factor in the sample size. This completes the proof. □

## D.3 Proof of Lemma D.5: movement of centers of $P'$

Recall that $\widetilde{P}_i := \{p + \xi_p | p \in P_i\} \subset \widehat{P}$ represent the cluster with noise corresponding to $P_i$. Let $O_i := \widetilde{P}_i \setminus B_i$, $O_{i \to j} := \widetilde{P}_i \cap B_j$ where $j \neq i$, $I_i := \cup_{j \neq i} O_{j \to i}$. By the above definition, we have $P_i' = (\widetilde{P}_i \setminus O_i) \cup I_i$.

For simplicity, we use $n_i, n_i^{in}, n_i^{out}$ to denote $|\tilde{P}_i|, |O_i|, |I_i|$ respectively.

**Lemma D.11 (Impact of removed points).** *With probability $p > 0.99$, for any $i \in [k]$,*

- $\frac{n_i^{out}}{n_i^2} \sum_{p \in O_i} \|p - \mu(P_i')\|_2^2 \leq O(\sqrt{\alpha - 1} \cdot \theta d)$.

- $\frac{n_i^{in}}{n_i^2} \sum_{p \in I_i} \|p - \mu(P_i')\|_2^2 \leq O(\sqrt{\alpha - 1} \cdot \theta d)$.

*Proof.* Note that since $\xi_{p,j}$ satisfies Bernstein condition for any $j \in [d]$ if $\xi_p \neq 0$, we have

$$\Pr\left[|\xi_{p,j}| \geq t | \xi_p \neq 0\right] \leq 2\exp\left(-\frac{t^2}{2(1 + bt)}\right), \forall j \in [d], t > 0.$$

Thus we have for large enough $t$ (larger than $b$),

$$\Pr\left[|\xi_{p,j}| \geq t | \xi_p \neq 0\right] \leq 2\exp\left(-\Omega(t)\right).$$

By union bound, with high probability,

$$\Pr\left[\|\xi_p\| \geq t | \xi_p \neq 0\right] \leq d \cdot \Pr\left[|\xi_{p,j}| \geq \frac{t}{\sqrt{d}} \middle| \xi_p \neq 0\right] \leq 2d\exp\left(-\Omega(\frac{t}{\sqrt{d}})\right).$$

Note that for any $p \in O_i$,

$$\|p - \mu(P_i')\|_2 \leq \|p - \xi_p - \mu(P_i')\|_2 + \|\xi_p\|_2 \leq 2(r_i + O(\sqrt{d}\log\frac{1+\theta kd}{\sqrt{\alpha-1}})) + \|\xi_p\|_2$$

Then we can decompose the cost as

$$\mathbb{E}\left[\sum_{p\in O_i}\|p - \mu(P_i')\|_2^2\right]$$

$$\leq n_i \cdot \theta \cdot \int_{r_i+O(\sqrt{d}\log\frac{1+\theta kd}{\sqrt{\alpha-1}})} \left(2(r_i + O(\sqrt{d}\log\frac{1+\theta kd}{\sqrt{\alpha-1}})) + t\right)^2 \rho(\|\xi_p\| = t)dt$$

$$\leq n_i \cdot \theta \cdot \int_{r_i+O(\sqrt{d}\log\frac{1+\theta kd}{\sqrt{\alpha-1}})} 9t^2\rho(\|\xi_p\| = t)dt.$$

Note that when $\Pr\left[\|\xi_p \geq t | \xi_p \neq 0\right] \leq 2d\exp\left(-\Omega(\frac{t}{\sqrt{d}})\right)$, for a constant $c$, $\rho(\|\xi_p\| = t)$ satisfies that

$$\rho(\|\xi_p\| = t) \leq 2c\sqrt{d}\exp\left(-\frac{ct}{\sqrt{d}}\right).$$

Thus

$$\int_{r_i+O(\theta\sqrt{d}\log k)} t^2\rho(\|\xi_p\| = t)dt \leq d(r_i + O(\sqrt{d}\log\frac{1+\theta kd}{\sqrt{\alpha-1}}))^2 \cdot \exp[\Omega(-\frac{r_i + O(\sqrt{d}\log\frac{1+\theta kd}{\sqrt{\alpha-1}})}{\sqrt{d}})]$$

$$\leq O(\sqrt{\alpha-1}d).$$

Then applying Markov's inequality, we have with probability $p > 0.99$,

$$\frac{n_i^{out}}{n_i}\sum_{p\in O_i}\|p - \mu(P_i')\|_2^2 \leq \frac{1}{n_i}\sum_{p\in O_i}\|p - \mu(P_i')\|_2^2 \leq O(\sqrt{\alpha-1}\cdot\theta d).$$

By the definition of $I_i$, $p \in B(\widetilde{c}_i, R_i)$ for any $p \in I_i$, thus

$$\max_{p\in I_i}\|p - \mu(\widetilde{P}_i))\|_2 \leq 2R_i.$$

Note that for any point $p \in I_i \cap \widetilde{P}_j$,

$$\|\widehat{c}_i - \widehat{c}_j\|_2 \leq R_i + \|p - \widehat{c}_j\|_2 \leq R_i + R_j + \|\xi_p\|_2,$$

By the stability of dataset $P$, every point $p \in I_i \cap \widetilde{P}_j$ satisfies that

$$\|\xi_p\|_2^2 \geq \alpha \cdot O(\frac{\mathsf{OPT} + \theta nd\log^2(\frac{kd}{\sqrt{\alpha-1}})}{\min(n_i, n_j)}) - O(R_i + R_j) \geq O(\frac{\mathsf{OPT} + \theta nd\log^2(\frac{kd}{\sqrt{\alpha-1}})}{\min(n_i, n_j)})$$

Thus

$$\mathbb{E}(\frac{n_i^{in}}{n_i^2} \sum_{p \in I_i} \|p - \mu(P_i')\|_2^2) \le \mathbb{E}(\frac{n_i^{in}}{n_i^2} \cdot 4n_i^{in} R_i^2)$$

$$\le \frac{4R_i^2}{n_i^2} \mathbb{E}\left((n_i^{in})^2\right)$$

$$\le \frac{4R_i^2 n^2 \theta^2}{n_i^2} \exp\left[-\Omega(R_i)\right] \cdot \exp\left[-\Omega(\frac{\theta n \log^2(\frac{kd}{\sqrt{\alpha-1}})}{n_i})\right]$$

$$\le O((\frac{\theta n}{n_i})^2) \cdot \exp\left[-\Omega(\frac{\theta n \log^2(\frac{kd}{\sqrt{\alpha-1}})}{n_i})\right]$$

$$\le O(\theta d \cdot \sqrt{\alpha-1})$$

Then we complete the proof by Markov inequality. $\qquad\square$

For the center of $P_i$ and $\tilde{P}_i$, we have the following properties:

**Lemma D.12 (Movement of noisy centers).** *With high probability, for any $i \in [k]$,*

$$\|\mu(P_i) - \mu(\tilde{P}_i)\|_2^2 \le O(\frac{\theta d}{n_i}).$$

*Proof.* Note that

$$\mu(\tilde{P}_i) = \frac{\sum_{\hat{p} \in \tilde{P}_i} \hat{p}}{n_i} = \mu(P_i) + \frac{\sum_{p \in P_i} \xi_p}{n_i}.$$

Thus it remains to show

$$\|\frac{\sum_{p \in P_i} \xi_p}{n_i}\|_2^2 \le O(\frac{\theta d}{n_i}).$$

As shown in Claim D.6, $\mathbb{E}_{\tilde{P}_i}\left[\sum_{p \in P_i} \|\xi_p\|_2^2\right] = \theta n_i d$ and $\text{Var}_{\tilde{P}_i}\left[\sum_{p \in P_i} \|\xi_p\|_2^2\right] \le 3\theta n_i d^2$.

We also note that $\mathbb{E}_{\tilde{P}_i}\left[\|\sum_{p \in P_i} \xi_p\|_2\right] \le O(\sqrt{\theta n_i d})$ and

$$\text{Var}_{\tilde{P}_i}\left[\|\sum_{p \in P_i} \xi_p\|_2\right] \le \mathbb{E}_{\tilde{P}_i}\left[\|\sum_{p \in P_i} \xi_p\|_2^2\right] = \mathbb{E}_{\tilde{P}_i}\left[\sum_{p \in P_i} \|\xi_p\|_2^2\right] = \theta n_i d,$$

which implies that $\left\|\sum_{p \in P_i} \xi_p\right\|_2$ is $O(\sqrt{\theta n_i d})$ with high probability.

Therefore, with high probability, we have:

$$\left\|\mu(\tilde{P}_i) - \mu(P_i)\right\|_2^2 \le O\left(\frac{\theta d}{n_i}\right).$$

This completes the proof. $\qquad\square$

Now we prove Lemma D.5.

*Proof of Lemma D.5.* For any $i \in [k]$, we have

$$
\begin{aligned}
\|\mu(\widetilde{P}_i) - \mu(P_i')\|_2^2 &= \|\frac{\sum_{p \in \widetilde{P}_i} p}{n_i} - \mu(P_i')\|_2^2 \\
&= \|\frac{\sum_{p \in P_i'} p + \sum_{p \in O_i} p - \sum_{p \in I_i} p}{n_i} - \mu(P_i')\|_2^2 \\
&= \|\frac{\sum_{p \in O_i}(p - \mu(P_i')) - \sum_{p \in I_i}(p - \mu(P_i'))}{n_i}\|_2^2 \\
&\leq \frac{2n_i^{out}}{n_i^2} \cdot \sum_{p \in O_i} \|p - \mu(P_i')\|_2^2 + \frac{2n_i^{in}}{n_i^2} \cdot \sum_{p \in I_i} \|p - \mu(P_i')\|_2^2 \\
&\leq O(\theta d \cdot \sqrt{\alpha - 1}) \qquad\qquad\qquad\qquad\qquad\qquad \text{(Lemma D.11)}
\end{aligned}
$$

By Lemma D.12, with high probability,

$$
\|\mu(P_i) - \mu(\tilde{P}_i)\|_2^2 \leq O(\frac{\theta d}{n_i}).
$$

Thus

$$
\|\mu(P_i) - \mu(P_i')\|_2^2 \leq 2\|\mu(P_i) - \mu(\tilde{P}_i)\|_2^2 + 2\|\mu(\widetilde{P}_i) - \mu(P_i')\|_2^2 \leq O(\theta d \cdot \sqrt{\alpha - 1} + \frac{\theta d}{n_i}),
$$

which completes the proof. $\qquad\square$

### D.4 Justifying Assumption D.2: Finding center sets within local balls

Lemmas D.8 and D.12 provide us with control of the clustering cost and the location of the optimal center set $\mathsf{C}(\widehat{P})$ of $\widehat{P}$. Together with Assumption 3.2, we know that $\widehat{P}$ is $O(\alpha)$-cost-stable. By [71, 59], we can efficiently compute an $O(1)$-approximate center set $C \in \mathcal{C}$ for such stable $\widehat{P}$. Partition $\widehat{P}$ into $\widehat{P}_1, \ldots, \widehat{P}_k$ of this $C$. Using a similar argument as for Lemma D.11, we can show that $\widehat{P} \cap B(c_i^\star, O(r_i + \sqrt{d} \log \frac{1 + \theta k d}{\sqrt{\alpha - 1}})) \subseteq \widehat{P}_i$. Furthermore, by a similar argument as for D.12, we can show that $\mu(\widehat{P}_i) \in B(c_i^\star, O(r_i + \sqrt{d} \log \frac{1 + \theta k d}{\sqrt{\alpha - 1}}))$. Thus, letting $\widehat{C}$ be the collection of $\mu(\widehat{P}_i)$'s satisfies Assumption D.2.

## E    Missing results and discussions from Section 3

This appendix provides additional results and clarifications supporting our theoretical analysis. We first examine the necessity of the cost-stability assumption in worst-case settings and then illustrate how to examine Assumption 3.2 in practice. Finally, we show the impact of using an approximate optimal solution of $P$ in the algorithms.

### E.1    Necessity of the stability assumption in the worst case

We present an example to demonstrate the necessity of the cost-stability assumption (Assumption 3.2) for ensuring a meaningful separation between the traditional error Err and the proposed $\text{Err}_\alpha$.

Consider the 3-Means problem in $\mathbb{R}$ (i.e., $k = 3$, $d = 1$). Let $P \subset \mathbb{R}$ consist of $\frac{n}{4}$ points each at positions $-3.25\sqrt{2}, -1.25\sqrt{2}, 1.25\sqrt{2}$, and $3.25\sqrt{2}$. A simple calculation shows that the optimal center set $\mathsf{C}(P)$ is either $\{-3.25\sqrt{2}, -1.25\sqrt{2}, 2.25\sqrt{2}\}$ or $\{3.25\sqrt{2}, 1.25\sqrt{2}, -2.25\sqrt{2}\}$, yielding $\text{OPT}_P(k) = n$ in both cases. Moreover, for the 2-Means problem, the optimal centers are at $\{\pm 2.25\sqrt{2}\}$, giving $\text{OPT}_P(k - 1) = 2n = 2 \cdot \text{OPT}_P(k)$. Hence, $P$ is $\gamma$-cost-stable with $\gamma = 1$.

Now let $\widehat{P}$ be a noisy version of $P$ generated under noise model I with $\theta = 1$, where each perturbation is sampled from $N(0, 1)$. Using the known expectation $\mathbb{E}_{x \sim N(0,1)}[|x| \mid x \geq 0] = \sqrt{2/\pi}$, we can approximate the means of the three clusters in $\widehat{P}$ as $-2.25\sqrt{2} - \sqrt{2/\pi}$, $0$, and $2.25\sqrt{2} + \sqrt{2/\pi}$,

respectively. Thus, the likely optimal center set for $\widehat{P}$ is $\mathsf{C}(S) = \{-2.25\sqrt{2} - \sqrt{2/\pi}, 0, 2.25\sqrt{2} + \sqrt{2/\pi}\}$, leading to:

$$\mathsf{Err}_1(\widehat{P}, P) = \frac{\mathrm{cost}(P, \mathsf{C}(S))}{\mathsf{OPT}_P(k)} - 1 \approx 0.82.$$

By further computation, we approximate $\widehat{P}$ by assuming $\xi_p = \pm\mathbb{E}(|\xi_p|) = \pm\sqrt{2/\pi}$ for each point $p$, then we observe that $\mathsf{OPT}_{\widehat{P}} \approx 1.24n$, thus $\mathsf{Err}(\widehat{P}, P) \approx \frac{\left|\mathsf{OPT}_{\widehat{P}} - \mathrm{cost}(P, \mathsf{C}(S))\right|}{\mathsf{OPT}_{\widehat{P}}} \approx 0.47$. This implies $\mathsf{Err}(\widehat{P}, P) \lesssim \mathsf{Err}_1(\widehat{P}, P)$.

This example underscores the importance of Assumption 3.2: in worst-case scenarios without cost-stability, the two errors may become comparable, and the benefit of using $\mathsf{Err}_\alpha$ may diminish.

**Empirical comparison of metrics.** To further compare the behavior of $\mathsf{Err}_\alpha$ and $\mathsf{Err}$, we conduct simulations on synthetic datasets with varying separation levels $\beta$.

Let $P$ consist of $\frac{n}{4}$ points each at positions $p_1 = -2\sqrt{2} - \frac{\beta\sqrt{2}}{2}$, $p_2 = \frac{-\beta\sqrt{2}}{2}$, $p_3 = \frac{\beta\sqrt{2}}{2}$, and $p_4 = 2\sqrt{2} + \frac{\beta\sqrt{2}}{2}$. When $\beta = 2.5$, this setup matches the example discussed above. Note that the distances $|p_2 - p_1| = |p_4 - p_3| = 2\sqrt{2}$ remain fixed, while only the inter-cluster gap $|p_3 - p_2|$ varies. This ensures that $\mathsf{OPT}_P = \theta nd$ and $\gamma = 1$ hold for all $\beta \geq 2$.

We set $n = 10,000$, $k = 3$, and vary $\beta$ from 2 to 3 in steps of 0.05. The dataset $P$ is perturbed under noise model I with $\theta = 1$ to obtain $\widehat{P}$.

We compute near-optimal $k$-means++ centers for $P$ and $\widehat{P}$, denoted $C^\star$ and $\widehat{C^\star}$, respectively. We approximate $\mathsf{Err}_1(\widehat{P}, P)$ by computing $\frac{\mathrm{cost}(P, \widehat{C^\star})}{\mathrm{cost}(P, C^\star)} - 1$.

To estimate $\mathsf{Err}(\widehat{P}, P)$, we randomly sample 500 candidate $k$-center sets $C_1, \dots, C_{500}$, each constructed by uniformly sampling $k$ points from the interval $[\min(\widehat{P}), \max(\widehat{P})]$. We define

$$\widehat{\mathsf{Err}}(\widehat{P}, P) := \max_{1 \leq i \leq 500} \frac{|\mathrm{cost}(\widehat{P}, C_i) - \mathrm{cost}(P, C_i)|}{\mathrm{cost}(\widehat{P}, C_i)}$$

as a proxy for the classical $\mathsf{Err}$ metric.

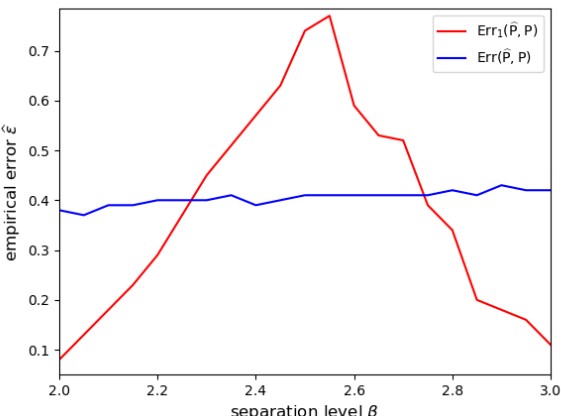

Figure 1: Empirical comparison of error metrics $\mathsf{Err}_1(\widehat{P}, P)$ and $\mathsf{Err}(\widehat{P}, P)$ on synthetic data as the separation level $\beta$ varies. While $\mathsf{Err}_1$ responds to changing cluster geometry, $\mathsf{Err}$ remains nearly constant, illustrating its insensitivity to structural properties in noisy settings.

Figure 1 presents the results. At $\beta = 2.5$, we observe that $\mathsf{Err}(\widehat{P}, P) \approx 0.41$ and $\mathsf{Err}_1(\widehat{P}, P) \approx 0.74$, consistent with our theoretical discussion. The overall trend confirms that $\mathsf{Err}$ and $\mathsf{Err}_\alpha$ can diverge significantly in value.

**Interpretation.** This simulation supports our comparison between the classical error metric $\mathrm{Err}(\widehat{P}, P)$ and the proposed surrogate $\mathrm{Err}_\alpha(\widehat{P}, P)$. It shows that $\mathrm{Err}$ is relatively insensitive to changes in the structural properties of the dataset, potentially underrepresenting approximation error as data becomes less separable. In contrast, $\mathrm{Err}_\alpha$, approximated here via $\mathrm{Err}_1$, decreases with $\beta$ (when $\beta > 2.5$), reflecting the growing challenge of summarizing data as cluster separation decreases.

Even when the optimal clustering is unaffected by noise—i.e., $\mathrm{OPT}_P$ and cluster geometry remain stable—$\mathrm{Err}$ remains nearly unchanged, while $\mathrm{Err}_\alpha$ adapts to the increased representational difficulty. This divergence (e.g., 0.41 vs. 0.74 at $\beta = 2.5$) underscores their differing sensitivities.

These findings reinforce our theoretical argument: $\mathrm{Err}_\alpha$ is a structurally aware and more reliable metric for noisy settings. This justifies its use in guiding coreset construction and motivates algorithms such as $\mathbf{CN}_\alpha$ that explicitly target this measure.

Interestingly, we observe that $\mathrm{Err}_1$ remains small even when the separation is minor (e.g., $\mathrm{Err}_1 < 0.1$ at $\beta = 2$), suggesting that $\mathrm{Err}_\alpha$ may also better align with the ideal metric $r_P$ in the presence of noise when clusters are close. This opens a promising direction for strengthening our theoretical guarantees in low-separation regimes.

### E.2 Examining Assumption 3.2 in practice

We detail the process for examining $\widehat{P}$ in $\mathbf{CN}_\alpha$, whose goal is to estimate whether $P$ meets the data assumptions.

Given $\widehat{P}$ and an $O(1)$-approximate center set $\widetilde{C}_k$ for $k$-MEANS problem, let $\widetilde{\mathrm{OPT}}(k) := \mathrm{cost}(\widehat{P}, \widetilde{C}_k)$. Correspondingly, compute an $O(1)$-approximate center set $\widetilde{C}_{k-1}$, let $\widetilde{\mathrm{OPT}}(k-1) := \mathrm{cost}(\widehat{P}, \widetilde{C}_{k-1})$. For *cost stability* assumption, we directly verify whether

$$\frac{\widetilde{\mathrm{OPT}}(k-1)}{\widetilde{\mathrm{OPT}}(k)} \geq 1 + O(\alpha) \cdot \left(1 + \frac{\theta n d \log^2(kd/\sqrt{\alpha - 1})}{\widetilde{\mathrm{OPT}}(k)}\right).$$

Suppose $P$ does not meet the stability assumption, for example, there exists a solution $C_{k-1}$ for $(k-1)$-Means problem, such that $\frac{\mathrm{cost}(P, C_{k-1})}{\mathrm{OPT}_P} < 1 + \gamma$. Then it's likely that a corresponding solution for $\widehat{P}$ will also violate the stability assumption.

For the assumption that $r_i \leq 8\bar{r}_i$, consider removing the top 1% of points in $\widehat{P}$ with the greatest distances to $\widetilde{C}$, resulting in a trimmed dataset $\widetilde{P}$. Recompute $r_i, \bar{r}_i$ for this adjusted dataset and $\widetilde{C}$, and check if $r_i \leq 8\bar{r}_i$ holds for every $i \in [k]$.

### E.3 Impact of Cost Estimation of $\mathrm{OPT}_P$ on Algorithm Performance

We discuss how the estimation of the optimal cost $\mathrm{OPT}_P$ affects the performance of $\mathbf{CN}_\alpha$ and $\mathbf{CN}$. Generally speaking, a $c$-estimate ($c \geq 2$) of the optimal cost on $P$ worsens the error guarantee for $\mathbf{CN}$ (using $\mathrm{Err}$ measure) and hardens the data assumption for $\mathbf{CN}_\alpha$ (using $\mathrm{Err}_\alpha$ measure). Below we illustrate these points.

Suppose we are given a center set $\widehat{C}$ for coreset construction, whose clustering cost is $\mathrm{cost}(\widehat{P}, \widehat{C}) = c \cdot \mathrm{OPT}_P$ for some $c > 1$ ($c$-estimate), where $P$ is the underlying dataset and $\widehat{P}$ is a given noisy dataset.

**Impact on CN.** Let $S$ be a coreset derived from CN. Employing a $c$-approximate estimate $\widehat{C}$ can weaken the quality bound of $S$ by up to a factor of $c$ in the following sense: With an exact estimate of $\mathrm{OPT}_P$, Theorem 3.1 provides a bound on the worst-case quality of an $\alpha$-approximate center set of $S$ as

$$r_P(S, \alpha) \leq \left(1 + \varepsilon + O\left(\frac{\theta n d}{\mathrm{OPT}_P} + \sqrt{\frac{\theta n d}{\mathrm{OPT}_P}}\right)\right)^2 \cdot \alpha,$$

In contrast, using CN with the approximate estimate $\widehat{C}$ yields the following bound for the derived coreset $S$:

$$r_P(S, \alpha) \leq \left(1 + c \cdot \varepsilon + O\left(\frac{\theta n d}{\mathrm{OPT}_P} + \sqrt{\frac{\theta n d}{\mathrm{OPT}_P}}\right)\right)^2 \cdot \alpha.$$

Compared to the exact estimation case, this bound increases the error term from $\varepsilon$ to $c \cdot \varepsilon$. This arises because the importance sampling procedure from [10], used in **CN**, introduces an additional error term $\varepsilon \cdot \frac{\text{cost}(P,\widehat{C})}{\text{OPT}_P} = c \cdot \varepsilon$ in its analysis, whereas the term $O\left(\frac{\theta n d}{\text{OPT}_P} + \sqrt{\frac{\theta n d}{\text{OPT}_P}}\right)$ results from $\text{Err}(\widehat{P}, P)$ and remains unaffected by the quality of the estimation.

**Impact on $\mathbf{CN}_\alpha$.** Let $S$ be a coreset derived from $\mathbf{CN}_\alpha$. Using a $c$-approximation estimate $\widehat{C}$ weakens Assumption 3.2 by a factor of $c$; specifically, the required cost-stability constant $\gamma$ increases by this factor. Consequently, the range of datasets satisfying the theoretical bounds of $\mathbf{CN}_\alpha$ (as stated in Theorem 3.3) shrinks. This occurs because the performance guarantee of $\mathbf{CN}_\alpha$ relies on correctly identifying $k$ separated clusters, a task complicated by the approximation error in $\widehat{C}$. Increasing $\gamma$ by a factor of $c$ ensures these clusters are accurately identified, restoring the desired guarantees.

Additionally, using a $c$-estimate may still result in a high-qualified coreset as with perfect estimate in practice.

# F    Extensions of Theorems 3.1 and 3.3

This section extends Theorems 3.1 and 3.3 to other noise models and general $(k, z)$-CLUSTERING.

## F.1    Extension to other noise models

**Noise model II.**    Recall that under noise model II, for every $i \in [n]$ and $j \in [d]$, $\widehat{p}_{i,j} = p_{i,j} + \xi_{p,j}$, where $\xi_{p,j}$ is drawn from $D_j$, a probability distribution on $\mathbb{R}$ with mean 0, variance $\sigma^2$, and satisfying the Bernstein condition. The following Theorem shows the performance of coreset generated using the Err metric under noise model II. The only difference from Theorem 3.1 is that we replace $\theta$ by $\sigma^2$.

**Theorem F.1** (**Coreset using** Err **under noise model II**). *Let $\widehat{P}$ be drawn from $P$ via noise model II with known $\sigma^2 \geq 0$. Let $\varepsilon \in (0, 1)$ and fix $\alpha \geq 1$. Let $\mathcal{A}$ be an algorithm that constructs a weighted subset $S \subset \widehat{P}$ for $k$-MEANS of size $\mathcal{A}(\varepsilon)$ and with guarantee $\text{Err}(S, \widehat{P}) \leq \varepsilon$. Then*

$$\text{Err}(S, P) \leq \varepsilon + O(\tfrac{\sigma^2 n d}{\text{OPT}_P} + \sqrt{\tfrac{\sigma^2 n d}{\text{OPT}_P}}) \text{ and } r_P(S, \alpha) \leq (1 + \varepsilon + O(\tfrac{\sigma^2 n d}{\text{OPT}_P} + \sqrt{\tfrac{\sigma^2 n d}{\text{OPT}_P}}))^2 \cdot \alpha.$$

*Proof.* By a similar argument as that of Theorem 3.1, we only need to prove the following property:

$$\mathbb{E}_{\widehat{P}}\left[\sum_{p \in P} \|\xi_p\|_2^2\right] = O(\sigma^2 n d), \text{ and } \text{Var}_{\widehat{P}}\left[\sum_{p \in P} \|\xi_p\|_2^2\right] = O(\sigma^2 n d^2),$$

which again holds by the Bernstein condition. $\qquad\square$

For the $\text{Err}_\alpha$ metric, we first give the data assumption under noise model II, adapting from Assumption 3.2.

**Assumption F.2** (**Data assumption under noise model II**). *Given $\alpha \geq 1$ and $\sigma^2 \geq 0$, assume $P$ is $\gamma$-cost-stable with*

$$\gamma = O(\alpha) \cdot \left(1 + \frac{\sigma^2 n d \log^2(\frac{kd}{\sqrt{\alpha-1}})}{\text{OPT}_P}\right),$$

*and that $r_i \leq 8\bar{r}_i$ for all $i \in [k]$.*

**Theorem F.3** (**Coreset using the $\text{Err}_\alpha$ metric under noise model II**). *Let $\widehat{P}$ be an observed dataset drawn from $P$ under the noise model II with known parameter $\sigma^2 \in [0, \frac{\text{OPT}_P}{nd}]$. Let $\varepsilon \in (0, 1)$ and fix $\alpha \in [1, 2]$. Under Assumption F.2, there exists a randomized algorithm that constructs a weighted $S \subset \widehat{P}$ for $k$-MEANS of size $O(\frac{k \log k}{\varepsilon - \frac{\sqrt{\alpha-1}\sigma^2 nd}{\alpha \text{OPT}_P}} + \frac{(\alpha-1)k \log k}{(\varepsilon - \frac{\sqrt{\alpha-1}\sigma^2 nd}{\alpha \text{OPT}_P})^2})$ and with guarantee $\text{Err}_\alpha(S, \widehat{P}) \leq \varepsilon$. Moreover,*

$$\text{Err}_\alpha(S, P) \leq \varepsilon + O(\tfrac{\sigma^2 kd}{\text{OPT}_P} + \tfrac{\sqrt{\alpha-1}}{\alpha} \cdot \tfrac{\sqrt{\sigma^2 kd\text{OPT}_P} + \sigma^2 nd}{\text{OPT}_P}) \text{ and}$$

$$r_P(S, \alpha) \leq (1 + \varepsilon + O(\tfrac{\sigma^2 kd}{\text{OPT}_P} + \tfrac{\sqrt{\alpha-1}}{\alpha} \cdot \tfrac{\sqrt{\sigma^2 kd\text{OPT}_P} + \sigma^2 nd}{\text{OPT}_P})) \cdot \alpha.$$

*Proof.* By a similar argument as that of Theorem 3.3, we have the following property:

- $\mathbb{E}_{\widehat{P}}\left[\sum_{p\in P}\|\xi_p\|_2^2\right] = O(\sigma^2 nd)$, and $\mathrm{Var}_{\widehat{P}}\left[\sum_{p\in P}\|\xi_p\|_2^2\right] = O(\sigma^2 nd^2)$,

- $\mathbb{E}_{\widehat{P}}\left[\sum_{p\in P}\langle\xi_p, p-c\rangle\right] = 0$ and $\mathrm{Var}_{\widehat{P}}\left[\sum_{p\in P}\langle\xi_p, p-c\rangle\right] = \sigma^2 \cdot \mathrm{cost}(P, c)$.

which holds by the Bernstein condition.

Similar to the proof of Lemma D.12, for any $i \in [k]$, $\mathbb{E}_{\tilde{P}_i}\left[\|\sum_{p\in P_i}\xi_p\|_2\right] \le O(\sqrt{\sigma^2 n_i d})$ and

$$\mathrm{Var}_{\tilde{P}_i}\left[\|\sum_{p\in P_i}\xi_p\|_2\right] \le \mathbb{E}_{\tilde{P}_i}\left[\|\sum_{p\in P_i}\xi_p\|_2^2\right] = \mathbb{E}_{\tilde{P}_i}\left[\sum_{p\in P_i}\|\xi_p\|_2^2\right] = O(\sigma^2 n_i d),$$

Then by Chebyshev's inequality, with high probability, $\left\|\sum_{p\in P_i}\xi_p\right\|_2 \le O(\sqrt{\sigma^2 n_i d})$, which implies

$$\left\|\mu(\tilde{P}_i) - \mu(P_i)\right\|_2^2 = \|\frac{\sum_{p\in P_i}\xi_p}{n_i}\|_2^2 \le O(\frac{\sigma^2 d}{n_i}).$$

The remaining steps remain the same as in Theorem 3.3. $\qquad\square$

**Non-independent noise across dimensions.** For simplicity, we consider a specific setting where the covariance matrix of each noise vector $\xi_p$ is $\Sigma \in \mathbb{R}^{d\times d}$. Note that under the noise model II, $\Sigma = \sigma^2 \cdot I_d$ when each $D_j = N(0, \sigma^2)$. In contrast, this setting considers non-independent noise across dimensions.

Note that the above proofs rely on certain concentration properties of the terms $\sum_{p\in P}\|\xi_p\|_2^2$ and $\sum_{p\in P}\langle\xi_p, p-c\rangle$. In general, $\mathbb{E}[\|\xi_p\|_2^2] = \mathrm{trace}(\Sigma)$. Hence, by a similar argument as in the proof of Claim D.6, one can show that $\sum_{p\in P}\|\xi_P\|_2^2$ concentrates on $n \cdot \mathrm{trace}(\Sigma)$ and $\sum_{p\in P}\langle\xi_p, p-c\rangle \le O(\sqrt{\mathrm{trace}(\Sigma)\cdot\mathrm{cost}(P,c)})$. The only difference with Theorem 3.1 and 3.3 is that we replace the original variance term $\theta d$ to $\mathrm{trace}(\Sigma)$.

**Theorem F.4** (**Coreset using** Err **under non-independent noise model**)**.** *Let $\widehat{P}$ be drawn from $P$ under non-independent noise model where each $\xi_p$ is drawn from $D_j = N(0, \Sigma)$. Let $\varepsilon \in (0, 1)$ and fix $\alpha \ge 1$. Let $\mathcal{A}$ be an algorithm that constructs a weighted subset $S \subset \widehat{P}$ for $k$-MEANS of size $\mathcal{A}(\varepsilon)$ and with guarantee $\mathrm{Err}(S, \widehat{P}) \le \varepsilon$. Then*

$$\mathrm{Err}(S, P) \le \varepsilon + O(\frac{n\cdot\mathrm{trace}(\Sigma)}{\mathsf{OPT}_P} + \sqrt{\frac{n\cdot\mathrm{trace}(\Sigma)}{\mathsf{OPT}_P}})\ and$$

$$r_P(S, \alpha) \le (1 + \varepsilon + O(\frac{n\cdot\mathrm{trace}(\Sigma)}{\mathsf{OPT}_P} + \sqrt{\frac{n\cdot\mathrm{trace}(\Sigma)}{\mathsf{OPT}_P}}))^2 \cdot \alpha.$$

*Proof.* By a similar argument as that of Theorem 3.1, we only need to prove the following property:

$$\mathbb{E}_{\widehat{P}}\left[\sum_{p\in P}\|\xi_p\|_2^2\right] = O(n\cdot\mathrm{trace}(\Sigma)),\ and\ \mathrm{Var}_{\widehat{P}}\left[\sum_{p\in P}\|\xi_p\|_2^2\right] = O(n\cdot\mathrm{trace}(\Sigma)^2).$$

The remaining steps of the proof remain the same as in Theorem 3.1. $\qquad\square$

For the $\mathrm{Err}_\alpha$ metric, we also extend Theorem 3.3 to this noise model.

**Assumption F.5** (**Data assumption under non-independent noise model**)**.** Given $\alpha \ge 1$ and a covariance matrix $\Sigma \in \mathbb{R}^{d\times d}$, assume $P$ is $\gamma$-cost-stable with

$$\gamma = O(\alpha) \cdot \left(1 + \frac{n\cdot\mathrm{trace}(\Sigma)\cdot\log^2(\frac{kd}{\sqrt{\alpha-1}})}{\mathsf{OPT}_P}\right),$$

and that $r_i \le 8\bar{r}_i$ for all $i \in [k]$.

**Theorem F.6** (**Coreset using the** $\mathrm{Err}_\alpha$ **metric under non-independent noise model**). *Let $\widehat{P}$ be an observed dataset drawn from $P$ under non-independent noise model where each $\xi_p$ is drawn from $D_j = N(0, \Sigma)$ with $\mathrm{trace}(\Sigma) \in [0, \frac{\mathsf{OPT}_P}{nd}]$. Let $\varepsilon \in (0,1)$ and fix $\alpha \in [1,2]$. Under Assumption F.5, there exists a randomized algorithm that constructs a weighted $S \subset \widehat{P}$ for $k$-MEANS of size $O(\frac{k \log k}{\varepsilon - \frac{\sqrt{\alpha-1}n \cdot \mathrm{trace}(\Sigma)}{\alpha \mathsf{OPT}_P}} + \frac{(\alpha-1)k \log k}{(\varepsilon - \frac{\sqrt{\alpha-1}n \cdot \mathrm{trace}(\Sigma)}{\alpha \mathsf{OPT}_P})^2})$ and with guarantee $\mathrm{Err}_\alpha(S, \widehat{P}) \leq \varepsilon$. Moreover,*

$$\mathrm{Err}_\alpha(S, P) \leq \varepsilon + O\Big(\frac{k \cdot \mathrm{trace}(\Sigma)}{\mathsf{OPT}_P} + \frac{\sqrt{\alpha-1}}{\alpha} \cdot \frac{\sqrt{k \cdot \mathrm{trace}(\Sigma) \cdot \mathsf{OPT}_P} + n \cdot \mathrm{trace}(\Sigma)}{\mathsf{OPT}_P}\Big) \text{ and}$$

$$r_P(S, \alpha) \leq (1 + \varepsilon + O\Big(\frac{k \cdot \mathrm{trace}(\Sigma)}{\mathsf{OPT}_P} + \frac{\sqrt{\alpha-1}}{\alpha} \cdot \frac{\sqrt{k \cdot \mathrm{trace}(\Sigma) \cdot \mathsf{OPT}_P} + n \cdot \mathrm{trace}(\Sigma)}{\mathsf{OPT}_P}\Big)) \cdot \alpha.$$

*Proof.* By a similar argument as that of Theorem 3.3, we have the following property:

- $\mathbb{E}_{\widehat{P}}\Big[\sum_{p \in P} \|\xi_p\|_2^2\Big] = O(n \cdot \mathrm{trace}(\Sigma))$, and $\mathrm{Var}_{\widehat{P}}\Big[\sum_{p \in P} \|\xi_p\|_2^2\Big] = O(n \cdot \mathrm{trace}(\Sigma)^2)$,

- $\mathbb{E}_{\widehat{P}}\Big[\sum_{p \in P}\langle \xi_p, p - c\rangle\Big] = 0$ and $\mathrm{Var}_{\widehat{P}}\Big[\sum_{p \in P}\langle \xi_p, p - c\rangle\Big] = \mathrm{trace}(\Sigma) \cdot \mathrm{cost}(P, c)$.

which holds by the Bernstein condition.

Similar to the proof of Lemma D.12, for any $i \in [k]$,

$$\mathrm{Var}_{\tilde{P}_i}\Big[\|\sum_{p \in P_i} \xi_p\|_2\Big] \leq \mathbb{E}_{\tilde{P}_i}\Big[\|\sum_{p \in P_i} \xi_p\|_2^2\Big] = \mathbb{E}_{\tilde{P}_i}\Big[\sum_{p \in P_i} \|\xi_p\|_2^2\Big] = O(n_i \cdot \mathrm{trace}(\Sigma)),$$

Then by Chebyshev's inequality, with high probability, $\left\|\sum_{p \in P_i} \xi_p\right\|_2 \leq O(\sqrt{n_i \cdot \mathrm{trace}(\Sigma)})$, which implies

$$\left\|\mu(\tilde{P}_i) - \mu(P_i)\right\|_2^2 = \|\frac{\sum_{p \in P_i} \xi_p}{n_i}\|_2^2 \leq O\Big(\frac{\mathrm{trace}(\Sigma)}{n_i}\Big).$$

The remaining steps remain the same as in Theorem 3.3. $\qquad\square$

## F.2 Extension to $(k, z)$-CLUSTERING

The following theorem extends Theorem 3.1 to $(k, z)$-CLUSTERING, under both noise models I and II.

**Theorem F.7** (($k, z$)-**CLUSTERING coreset using the** $\mathrm{Err}_\alpha$ **metric in the presence of noise**). *Let $\widehat{P}$ be drawn from $P$ via noise model I (or II) with known $\theta \geq 0$ (or $\sigma^2 \geq 0$). Let $\varepsilon \in (0,1)$ and fix $\alpha \geq 1$. Let $\mathcal{A}$ be an algorithm that constructs a weighted subset $S \subset \widehat{P}$ for $k$-MEANS of size $\mathcal{A}(\varepsilon)$ and with guarantee $\mathrm{Err}(S, \widehat{P}) \leq \varepsilon$. Then for noise model I,*

$$\mathrm{Err}(S, P) \leq \varepsilon + O\Big(\frac{\theta n d^{z/2}}{\mathsf{OPT}_P} + \sqrt[z]{\frac{\theta n d^{z/2}}{\mathsf{OPT}_P}}\Big) \text{ and } r_P(S, \alpha) \leq (1 + \varepsilon + O\Big(\frac{\theta n d^{z/2}}{\mathsf{OPT}_P} + \sqrt[z]{\frac{\theta n d^{z/2}}{\mathsf{OPT}_P}}\Big))^2 \cdot \alpha.$$

*For noise model II,*

$$\mathrm{Err}(S, P) \leq \varepsilon + O\Big(\frac{\sigma^z n d^{z/2}}{\mathsf{OPT}_P} + \sqrt[z]{\frac{\sigma^z n d^{z/2}}{\mathsf{OPT}_P}}\Big) \text{ and } r_P(S, \alpha) \leq (1 + \varepsilon + O\Big(\frac{\sigma^z n d^{z/2}}{\mathsf{OPT}_P} + \sqrt[z]{\frac{\sigma^z n d^{z/2}}{\mathsf{OPT}_P}}\Big))^2 \cdot \alpha.$$

*Proof.* The case of $k$-MEANS has been proved in Theorem 3.1. Now we show how to extend to $(k, z)$-CLUSTERING. For the upper bound, the main difference is that we have

$$|d^z(p, C) - d^z(\widehat{p}, C)| \leq O_z\left(\|\xi_p\|_2^z + \|\xi_p\|_2 \cdot d^{z-1}(p, C)\right),$$

Table 2: Datasets used in our experiments. $\gamma$ represents the cost stability constant. $r_i$ and $\bar{r}_i$ are as defined in Assumption 3.2. Note that the assumption $\max_i \frac{r_i}{\bar{r}_i} < 8$ holds for both datasets. The `Census1990` dataset consists of 2458285 data points and we subsample 100000 from them. We also drop the categorical features of the datasets and only keep the continuous features for clustering.

| Dataset | Size | Dim | $k$ | $\gamma$ | $\max_i \frac{r_i}{\bar{r}_i}$ |
|---|---|---|---|---|---|
| Adult | 41188 | 10 | 10 | 0.07 | 7.52 |
| Census1990 | $10^5$ | 68 | 10 | 0.03 | 5.93 |

where $O_z(\cdot)$ hides constant factor $2^{O(z)}$. Similar to Claim D.6, we assume $\sum_{p \in P} \|\xi_p\|_2^z \leq O_z(\theta n d^{z/2})$ by the Bernstein condition, which happens with probability at least 0.9. Then we have

$$\frac{|\text{cost}_z(P, C) - \text{cost}_z(\widehat{P}, C)|}{\text{cost}_z(\widehat{P}, C)}$$

$$\leq \frac{O_z\left(\sum_{p \in P} \|\xi_p\|_2^z + \|\xi_p\|_2 \cdot d^{z-1}(p, C)\right)}{\text{cost}_z(\widehat{P}, C)}$$

$$\leq O\left(\frac{\theta n d}{\text{OPT}}\right) + \frac{O_z\left(\sum_{p \in P} \|\xi_p\|_2 \cdot d^{z-1}(p, C)\right)}{\text{OPT}} \qquad \text{(by assumption)}$$

$$\leq O\left(\frac{\theta n d^{\frac{z}{2}}}{\text{OPT}}\right) + \frac{O_z\left(\sqrt[z]{(\sum_{p \in P} \|\xi_p\|_2^z)(\sum_{p \in P} d^z(p, C))^{z-1}}\right)}{\text{OPT}} \qquad \text{(Generalized Hölder inequality)}$$

$$\leq O\left(\frac{\theta n d}{\text{OPT}}\right) + \sqrt[z]{\frac{O(\theta n d^{z/2})}{\text{OPT}}} \qquad \text{(by assumption)}$$

$$\leq O\left(\frac{\theta n d}{\text{OPT}} + \sqrt[z]{\frac{\theta n d^{z/2}}{\text{OPT}}}\right), \qquad \text{(Defn. of OPT)}$$

which completes the proof for $(k, z)$-CLUSTERING under noise model I.

Similarly, for $(k, z)$-CLUSTERING under noise model II, we only need to prove the following property:

$$\mathbb{E}_{\widehat{P}}\left[\sum_{p \in P} \|\xi_p\|_2^2\right] = O_z(\sigma^z n d^{z/2}), \text{ and } \text{Var}_{\widehat{P}}\left[\sum_{p \in P} \|\xi_p\|_2^2\right] = O_z(\sigma^{2z} n d^z),$$

which again holds by the Bernstein condition. This completes the proof. $\qquad \square$

The extension of Theorem 3.3 to general $(k, z)$-CLUSTERING introduces additional technical challenges, as the optimal center for $k = 1$ is not the mean point, making it more difficult to control the location of centers. Adapting the use of the $\text{Err}_\alpha$ metric to general $(k, z)$-CLUSTERING remains an interesting direction for future work.

# G  Additional empirical results

In this section, we provide the implementation details of **CN** and **CN**$_\alpha$, and give more empirical results to corroborate our findings in Section 4. All experiments are conducted using Python 3.11 on an Apple M3 Pro machine with an 11-core CPU, 14-core GPU, and 36 GB of memory.

## G.1  Implementation details of CN and CN$_\alpha$

**Implementation of CN.**  Given the perturbed dataset $\widehat{P}$ and budget $\varepsilon > 0$, **CN** first computes an approximate optimal center set $\widetilde{C}$ of $\widehat{P}$ by $k$-means++ with max_iter $= 5$ and computes $\widetilde{OPT} = \text{cost}_2(\widehat{P}, \widetilde{C})$. Then it constructs a coreset by the importance sampling algorithm [10] with coreset size $|S| = 3k^{1.5}\varepsilon^{-2}$, which represents the size bound derived by $\text{Err}$. The runtime of **CN** is $O(ndk)$.

Table 3: Result of census1990 dataset under noise model I with Gaussian noise. $|S|$ represents the coreset size, $\widetilde{r}_S$ represents its empirical approximation ratio, and $\kappa_S$ denotes the ratio of its empirical approximation ratio over the theoretical bound.

(a) $\theta = 0$

| | $\varepsilon$ | 0.1 | 0.15 | 0.2 | 0.25 | 0.3 |
|---|---|---|---|---|---|---|
| $|S|$ | **CN** | 9486 | 4216 | 2371 | 1517 | 1054 |
| | $\mathbf{CN}_\alpha$ | 6835 | 3260 | 1922 | 1320 | 960 |
| $\widetilde{r}_S$ | **CN** | 1.028 | 1.026 | 1.043 | 1.052 | 1.056 |
| | $\mathbf{CN}_\alpha$ | 1.057 | 1.067 | 1.079 | 1.071 | 1.087 |
| $\kappa_S$ | **CN** | 0.849 | 0.776 | 0.724 | 0.673 | 0.625 |
| | $\mathbf{CN}_\alpha$ | 0.961 | 0.928 | 0.899 | 0.857 | 0.836 |

(b) $\theta = 0.01$

| | $\varepsilon$ | 0.1 | 0.15 | 0.2 | 0.25 | 0.3 |
|---|---|---|---|---|---|---|
| $|S|$ | **CN** | 9486 | 4216 | 2371 | 1517 | 1054 |
| | $\mathbf{CN}_\alpha$ | 6779 | 3260 | 1940 | 1320 | 960 |
| $\widetilde{r}_S$ | **CN** | 1.038 | 1.030 | 1.043 | 1.052 | 1.060 |
| | $\mathbf{CN}_\alpha$ | 1.058 | 1.059 | 1.063 | 1.071 | 1.071 |
| $\kappa_S$ | **CN** | 0.655 | 0.602 | 0.565 | 0.530 | 0.498 |
| | $\mathbf{CN}_\alpha$ | 0.945 | 0.905 | 0.872 | 0.843 | 0.812 |

(c) $\theta = 0.05$

| | $\varepsilon$ | 0.1 | 0.15 | 0.2 | 0.25 | 0.3 |
|---|---|---|---|---|---|---|
| $|S|$ | **CN** | 9486 | 4216 | 2371 | 1517 | 1054 |
| | $\mathbf{CN}_\alpha$ | 6834 | 3260 | 1928 | 1320 | 960 |
| $\widetilde{r}_S$ | **CN** | 1.024 | 1.036 | 1.057 | 1.053 | 1.059 |
| | $\mathbf{CN}_\alpha$ | 1.069 | 1.061 | 1.065 | 1.097 | 1.091 |
| $\kappa_S$ | **CN** | 0.451 | 0.427 | 0.409 | 0.384 | 0.363 |
| | $\mathbf{CN}_\alpha$ | 0.893 | 0.851 | 0.821 | 0.815 | 0.781 |

(d) $\theta = 0.25$

| | $\varepsilon$ | 0.1 | 0.15 | 0.2 | 0.25 | 0.3 |
|---|---|---|---|---|---|---|
| $|S|$ | **CN** | 9486 | 4216 | 2371 | 1517 | 1054 |
| | $\mathbf{CN}_\alpha$ | 6708 | 3176 | 1922 | 1320 | 960 |
| $\widetilde{r}_S$ | **CN** | 1.041 | 1.041 | 1.054 | 1.086 | 1.088 |
| | $\mathbf{CN}_\alpha$ | 1.079 | 1.067 | 1.104 | 1.122 | 1.106 |
| $\kappa_S$ | **CN** | 0.201 | 0.192 | 0.187 | 0.184 | 0.177 |
| | $\mathbf{CN}_\alpha$ | 0.682 | 0.653 | 0.656 | 0.648 | 0.620 |

**Implementation of $\mathbf{CN}_\alpha$.** Given the perturbed dataset $\widehat{P}$ and budget $\varepsilon > 0$, $\mathbf{CN}_\alpha$ also first computes an approximate optimal center set $\widetilde{C}$ of $\widehat{P}$ by $k$-means++ with max_iter = 5 and computes $\widetilde{OPT} = \text{cost}_2(\widehat{P}, \widetilde{C})$. Next, we decompose $\widehat{P}$ into $k$ clusters $\widehat{P}_i$ by $\widetilde{C}$. For each $i \in [k]$, compute $\widehat{r}_i = \sqrt{\frac{\text{cost}(\widehat{P}_i, \widehat{c}_i)}{|\widehat{P}_i|}}$. Then we compute $P'_i = \widehat{P}_i \cap B_i$, where ball $B_i := B(\widehat{c}_i, R_i)$ with $R_i := \widehat{r}_i + \sqrt{d} \log 10(1 + \theta k d)$. For each $i \in [k]$, take a uniform sample $S_i$ of size $\min(|P'_i|, \frac{9}{\varepsilon} + \frac{6}{\varepsilon^2})$ as an approximation of theoretical results. Finally, it returns $S = \bigcup_{i \in [k]} S_i$ with $w(p) = \frac{|P'_i|}{S_i}$ for $p \in S_i$. Similarly, the runtime of $\mathbf{CN}_\alpha$ is $O(ndk)$.

## G.2  Results on the Census1990 dataset

The setup has been shown in Section 4. See Table 3 for results. All observations are consistent with that for Adult.

## G.3  Results under other noise settings

We present additional results under various noise settings using the Adult dataset, with the same noise levels and tolerance parameters as in Section 4. Tables 4 and 5 evaluate coreset performance under noise model I, replacing Gaussian noise with Laplacian and Uniform noise, respectively. The results are consistent with Table 1, validating the robustness of our theoretical findings across different noise types. Table 6 assesses performance under noise model II, illustrating the practical relevance of Theorem F.3, an extension of Theorem 3.3. Table 7 considers non-independent noise, as analyzed in Theorem F.6, and again shows consistency with Table 1. Overall, these results further support our theoretical analysis, confirming that coreset performance under noise is primarily governed by the noise variance.

Table 4: Result of `Adult` dataset under noise model I with Laplacian noise $\mathrm{Lap}(0, \frac{1}{\sqrt{2}})$. This choice of Laplacian noise ensures a variance of 1 per coordinate, matching that of the Gaussian noise. $|S|$ represents the coreset size, $\widetilde{r}_S$ represents its empirical approximation ratio, and $\kappa_S$ denotes the tightness ratio of its empirical approximation ratio over the theoretical bound.

(a) $\theta = 0$

| $\varepsilon$ | | 0.1 | 0.15 | 0.2 | 0.25 | 0.3 |
|---|---|---|---|---|---|---|
| $|S|$ | **CN** | 9486 | 4216 | 2371 | 1517 | 1054 |
| | $\mathbf{CN}_\alpha$ | 6428 | 3178 | 1940 | 1320 | 960 |
| $\widehat{r}_S$ | **CN** | 1.048 | 1.072 | 1.121 | 1.179 | 1.310 |
| | $\mathbf{CN}_\alpha$ | 1.138 | 1.096 | 1.173 | 1.155 | 1.153 |
| $\kappa_S$ | **CN** | 0.866 | 0.810 | 0.778 | 0.755 | 0.775 |
| | $\mathbf{CN}_\alpha$ | 1.035 | 0.953 | 0.978 | 0.924 | 0.887 |

(b) $\theta = 0.01$

| $\varepsilon$ | | 0.1 | 0.15 | 0.2 | 0.25 | 0.3 |
|---|---|---|---|---|---|---|
| $|S|$ | **CN** | 9486 | 4216 | 2371 | 1517 | 1054 |
| | $\mathbf{CN}_\alpha$ | 6445 | 3178 | 1940 | 1320 | 960 |
| $\widehat{r}_S$ | **CN** | 1.036 | 1.031 | 1.093 | 1.118 | 1.243 |
| | $\mathbf{CN}_\alpha$ | 1.083 | 1.111 | 1.120 | 1.188 | 1.173 |
| $\kappa_S$ | **CN** | 0.600 | 0.554 | 0.547 | 0.522 | 0.542 |
| | $\mathbf{CN}_\alpha$ | 0.956 | 0.940 | 0.908 | 0.926 | 0.880 |

(c) $\theta = 0.05$

| $\varepsilon$ | | 0.1 | 0.15 | 0.2 | 0.25 | 0.3 |
|---|---|---|---|---|---|---|
| $|S|$ | **CN** | 9486 | 4216 | 2371 | 1517 | 1054 |
| | $\mathbf{CN}_\alpha$ | 6445 | 3178 | 1940 | 1320 | 960 |
| $\widehat{r}_S$ | **CN** | 1.030 | 1.075 | 1.217 | 1.095 | 1.281 |
| | $\mathbf{CN}_\alpha$ | 1.081 | 1.113 | 1.174 | 1.116 | 1.147 |
| $\kappa_S$ | **CN** | 0.370 | 0.364 | 0.389 | 0.331 | 0.367 |
| | $\mathbf{CN}_\alpha$ | 0.855 | 0.847 | 0.860 | 0.789 | 0.783 |

(d) $\theta = 0.25$

| $\varepsilon$ | | 0.1 | 0.15 | 0.2 | 0.25 | 0.3 |
|---|---|---|---|---|---|---|
| $|S|$ | **CN** | 9486 | 4216 | 2371 | 1517 | 1054 |
| | $\mathbf{CN}_\alpha$ | 6445 | 3178 | 1940 | 1320 | 960 |
| $\widehat{r}_S$ | **CN** | 1.036 | 1.073 | 1.132 | 1.167 | 1.384 |
| | $\mathbf{CN}_\alpha$ | 1.158 | 1.125 | 1.183 | 1.204 | 1.221 |
| $\kappa_S$ | **CN** | 0.130 | 0.130 | 0.132 | 0.132 | 0.151 |
| | $\mathbf{CN}_\alpha$ | 0.603 | 0.571 | 0.586 | 0.581 | 0.576 |

Table 5: Result of `Adult` dataset under noise model I with Uniform noise $U[-\sqrt{3}, \sqrt{3}]$. This choice of Uniform noise ensures a variance of 1 per coordinate, matching that of the Gaussian noise. $|S|$ represents the coreset size, $\widetilde{r}_S$ represents its empirical approximation ratio, and $\kappa_S$ denotes the tightness ratio of its empirical approximation ratio over the theoretical bound.

(a) $\theta = 0$

| $\varepsilon$ | | 0.1 | 0.15 | 0.2 | 0.25 | 0.3 |
|---|---|---|---|---|---|---|
| $|S|$ | **CN** | 9486 | 4216 | 2371 | 1517 | 1054 |
| | $\mathbf{CN}_\alpha$ | 6445 | 3178 | 1940 | 1320 | 960 |
| $\widehat{r}_S$ | **CN** | 1.057 | 1.139 | 1.043 | 1.281 | 1.265 |
| | $\mathbf{CN}_\alpha$ | 1.106 | 1.097 | 1.142 | 1.169 | 1.155 |
| $\kappa_S$ | **CN** | 0.873 | 0.861 | 0.724 | 0.820 | 0.748 |
| | $\mathbf{CN}_\alpha$ | 1.006 | 0.954 | 0.951 | 0.935 | 0.889 |

(b) $\theta = 0.01$

| $\varepsilon$ | | 0.1 | 0.15 | 0.2 | 0.25 | 0.3 |
|---|---|---|---|---|---|---|
| $|S|$ | **CN** | 9486 | 4216 | 2371 | 1517 | 1054 |
| | $\mathbf{CN}_\alpha$ | 6445 | 3178 | 1940 | 1320 | 960 |
| $\widehat{r}_S$ | **CN** | 1.026 | 1.050 | 1.128 | 1.224 | 1.158 |
| | $\mathbf{CN}_\alpha$ | 1.089 | 1.125 | 1.098 | 1.208 | 1.130 |
| $\kappa_S$ | **CN** | 0.594 | 0.564 | 0.564 | 0.571 | 0.505 |
| | $\mathbf{CN}_\alpha$ | 0.962 | 0.951 | 0.891 | 0.942 | 0.848 |

(c) $\theta = 0.05$

| $\varepsilon$ | | 0.1 | 0.15 | 0.2 | 0.25 | 0.3 |
|---|---|---|---|---|---|---|
| $|S|$ | **CN** | 9486 | 4216 | 2371 | 1517 | 1054 |
| | $\mathbf{CN}_\alpha$ | 6445 | 3145 | 1940 | 1320 | 960 |
| $\widehat{r}_S$ | **CN** | 1.061 | 1.051 | 1.092 | 1.151 | 1.208 |
| | $\mathbf{CN}_\alpha$ | 1.131 | 1.133 | 1.136 | 1.163 | 1.111 |
| $\kappa_S$ | **CN** | 0.381 | 0.356 | 0.349 | 0.348 | 0.346 |
| | $\mathbf{CN}_\alpha$ | 0.894 | 0.862 | 0.832 | 0.822 | 0.759 |

(d) $\theta = 0.25$

| $\varepsilon$ | | 0.1 | 0.15 | 0.2 | 0.25 | 0.3 |
|---|---|---|---|---|---|---|
| $|S|$ | **CN** | 9486 | 4216 | 2371 | 1517 | 1054 |
| | $\mathbf{CN}_\alpha$ | 6445 | 3178 | 1940 | 1320 | 960 |
| $\widehat{r}_S$ | **CN** | 1.064 | 1.061 | 1.099 | 1.228 | 1.295 |
| | $\mathbf{CN}_\alpha$ | 1.117 | 1.136 | 1.217 | 1.220 | 1.214 |
| $\kappa_S$ | **CN** | 0.133 | 0.128 | 0.128 | 0.139 | 0.141 |
| | $\mathbf{CN}_\alpha$ | 0.582 | 0.576 | 0.602 | 0.589 | 0.573 |

Table 6: Result of `Adult` dataset under noise model II with Gaussian noise $N(0, \sigma^2)$. Here, $\sigma^2$ controls the variance of noise, playing the same role as $\theta$ under noise model I. $|S|$ represents the coreset size, $\widetilde{r}_S$ represents its empirical approximation ratio, and $\kappa_S$ denotes the tightness ratio of its empirical approximation ratio over the theoretical bound.

(a) $\sigma^2 = 0$

| $\varepsilon$ | | 0.1 | 0.15 | 0.2 | 0.25 | 0.3 |
|---|---|---|---|---|---|---|
| $|S|$ | **CN** | 9486 | 4216 | 2371 | 1517 | 1054 |
| | **CN$_\alpha$** | 6445 | 3178 | 1940 | 1320 | 960 |
| $\widehat{r}_S$ | **CN** | 1.023 | 1.071 | 1.133 | 1.219 | 1.220 |
| | **CN$_\alpha$** | 1.085 | 1.126 | 1.170 | 1.156 | 1.140 |
| $\kappa_S$ | **CN** | 0.845 | 0.810 | 0.787 | 0.780 | 0.722 |
| | **CN$_\alpha$** | 0.987 | 0.979 | 0.975 | 0.925 | 0.877 |

(b) $\sigma^2 = 0.01$

| $\varepsilon$ | | 0.1 | 0.15 | 0.2 | 0.25 | 0.3 |
|---|---|---|---|---|---|---|
| $|S|$ | **CN** | 9486 | 4216 | 2371 | 1517 | 1054 |
| | **CN$_\alpha$** | 6445 | 3178 | 1940 | 1320 | 960 |
| $\widehat{r}_S$ | **CN** | 1.039 | 1.066 | 1.075 | 1.104 | 1.280 |
| | **CN$_\alpha$** | 1.096 | 1.120 | 1.137 | 1.164 | 1.161 |
| $\kappa_S$ | **CN** | 0.602 | 0.573 | 0.538 | 0.515 | 0.558 |
| | **CN$_\alpha$** | 0.967 | 0.947 | 0.922 | 0.908 | 0.871 |

(c) $\sigma^2 = 0.05$

| $\varepsilon$ | | 0.1 | 0.15 | 0.2 | 0.25 | 0.3 |
|---|---|---|---|---|---|---|
| $|S|$ | **CN** | 9486 | 4216 | 2371 | 1517 | 1054 |
| | **CN$_\alpha$** | 6445 | 3178 | 1940 | 1320 | 960 |
| $\widehat{r}_S$ | **CN** | 1.045 | 1.074 | 1.218 | 1.135 | 1.234 |
| | **CN$_\alpha$** | 1.098 | 1.115 | 1.129 | 1.123 | 1.139 |
| $\kappa_S$ | **CN** | 0.375 | 0.363 | 0.389 | 0.343 | 0.353 |
| | **CN$_\alpha$** | 0.869 | 0.849 | 0.828 | 0.794 | 0.778 |

(d) $\sigma^2 = 0.25$

| $\varepsilon$ | | 0.1 | 0.15 | 0.2 | 0.25 | 0.3 |
|---|---|---|---|---|---|---|
| $|S|$ | **CN** | 9486 | 4216 | 2371 | 1517 | 1054 |
| | **CN$_\alpha$** | 6445 | 3178 | 1940 | 1320 | 960 |
| $\widehat{r}_S$ | **CN** | 1.042 | 1.098 | 1.127 | 1.292 | 1.323 |
| | **CN$_\alpha$** | 1.079 | 1.125 | 1.217 | 1.215 | 1.231 |
| $\kappa_S$ | **CN** | 0.130 | 0.133 | 0.132 | 0.146 | 0.145 |
| | **CN$_\alpha$** | 0.562 | 0.571 | 0.602 | 0.587 | 0.581 |

Table 7: Result of `Adult` dataset under noise model II with non-independent Gaussian noise $N(0, \sigma^2 \Sigma)$ with $\mathrm{trace}(\Sigma) = d$. We first generate the covariance matrix $\Sigma$ randomly, and then apply it for different noise levels $\sigma^2$. This choice of $\mathrm{trace}(\Sigma)$ ensures a variance of $\sigma^2 d$ per point, matching that of noise model II. $|S|$ represents the coreset size, $\widetilde{r}_S$ represents its empirical approximation ratio, and $\kappa_S$ denotes the tightness ratio of its empirical approximation ratio over the theoretical ratio.

(a) $\sigma^2 = 0$

| $\varepsilon$ | | 0.1 | 0.15 | 0.2 | 0.25 | 0.3 |
|---|---|---|---|---|---|---|
| $|S|$ | **CN** | 9486 | 4216 | 2371 | 1517 | 1054 |
| | **CN$_\alpha$** | 6445 | 3178 | 1940 | 1320 | 960 |
| $\widehat{r}_S$ | **CN** | 1.029 | 1.051 | 1.101 | 1.265 | 1.227 |
| | **CN$_\alpha$** | 1.064 | 1.149 | 1.136 | 1.119 | 1.175 |
| $\kappa_S$ | **CN** | 0.851 | 0.795 | 0.764 | 0.810 | 0.726 |
| | **CN$_\alpha$** | 0.967 | 0.999 | 0.947 | 0.895 | 0.904 |

(b) $\sigma^2 = 0.01$

| $\varepsilon$ | | 0.1 | 0.15 | 0.2 | 0.25 | 0.3 |
|---|---|---|---|---|---|---|
| $|S|$ | **CN** | 9486 | 4216 | 2371 | 1517 | 1054 |
| | **CN$_\alpha$** | 6445 | 3178 | 1940 | 1320 | 960 |
| $\widehat{r}_S$ | **CN** | 1.022 | 1.039 | 1.133 | 1.147 | 1.240 |
| | **CN$_\alpha$** | 1.077 | 1.165 | 1.150 | 1.088 | 1.164 |
| $\kappa_S$ | **CN** | 0.592 | 0.559 | 0.566 | 0.535 | 0.541 |
| | **CN$_\alpha$** | 0.951 | 0.985 | 0.933 | 0.848 | 0.873 |

(c) $\sigma^2 = 0.05$

| $\varepsilon$ | | 0.1 | 0.15 | 0.2 | 0.25 | 0.3 |
|---|---|---|---|---|---|---|
| $|S|$ | **CN** | 9486 | 4216 | 2371 | 1517 | 1054 |
| | **CN$_\alpha$** | 6418 | 3178 | 1940 | 1320 | 960 |
| $\widehat{r}_S$ | **CN** | 1.038 | 1.036 | 1.043 | 1.152 | 1.337 |
| | **CN$_\alpha$** | 1.083 | 1.086 | 1.141 | 1.165 | 1.167 |
| $\kappa_S$ | **CN** | 0.373 | 0.350 | 0.333 | 0.348 | 0.383 |
| | **CN$_\alpha$** | 0.857 | 0.826 | 0.836 | 0.824 | 0.797 |

(d) $\sigma^2 = 0.25$

| $\varepsilon$ | | 0.1 | 0.15 | 0.2 | 0.25 | 0.3 |
|---|---|---|---|---|---|---|
| $|S|$ | **CN** | 9486 | 4216 | 2371 | 1517 | 1054 |
| | **CN$_\alpha$** | 6445 | 3178 | 1940 | 1320 | 960 |
| $\widehat{r}_S$ | **CN** | 1.090 | 1.145 | 1.221 | 1.219 | 1.304 |
| | **CN$_\alpha$** | 1.129 | 1.199 | 1.202 | 1.193 | 1.239 |
| $\kappa_S$ | **CN** | 0.136 | 0.138 | 0.143 | 0.138 | 0.142 |
| | **CN$_\alpha$** | 0.588 | 0.609 | 0.595 | 0.576 | 0.584 |

