# OpenReview forum: "Coresets for Clustering Under Stochastic Noise"
_NeurIPS.cc/2025/Conference — NeurIPS 2025 poster_

### Official Review · Reviewer_RYsn · 2025-06-20

**Clarity:** 2
**Significance:** 3
**Originality:** 3
**Rating:** 4
**Confidence:** 4

**Summary:**

The paper studies coresets for k-means clustering when the input dataset is corrupted by random noise. Specifically, the paper models noise by random perturbation of each data point. It then presents a a surrogate error metric, which is basically an upper bound on the error of the coreset but relative to the unperturbed dataset. The main result is then 2 algorithms for constructing a coreset from the perturbed dataset. The crux is that the algorithm's accuracy/quality is measured wrt the unperturbed points, even though the algorithm cannot see them. The algorithms are analyzed theoretically, showing smaller coreset size, and they are validated via a few experiments.

**Questions:**

none

**Ethical Concerns:**

["NO or VERY MINOR ethics concerns only"]

**Final Justification:**

I raised my score based on the rebuttal and other reviews, which helped me understand better the goal and contribution of this work.

**Limitations:**

yes

**Quality:**

3

**Strengths And Weaknesses:**

I like the paper's research direction of studying noisy data instead of worst-case data. As a side comment, such models where the input is a combination of adversarial + random choices, are called semi-random inputs or smoothed analysis in related literature:
David Arthur, Bodo Manthey, Heiko Röglin: Smoothed Analysis of the k-Means Method. J. ACM 58(5): 19:1-19:31 (2011)
Jonathan A. Kelner, Jerry Li, Allen Liu, Aaron Sidford, Kevin Tian: Semi-Random Matrix Completion via Flow-Based Adaptive Reweighting. NeurIPS 2024

However, I find the paper's methodology is unconvincing. First, the main error measure $Err(S,P)$ is only an upper bound on the error (the paper calls it a surrogate), and I doubt it is the correct measure that we would like to minimize. Second, the two algorithms do not bring new insights or methods to the area, in fact the first one (theorem 3.1) simply says to apply any known coreset algorithm, and thus has no new component (the contribution is analysis of the error wrt the unperturbed dataset). The second algorithm (for theorem 3.3) is follows a rather standard recipe, basically computing an O(1)-approximate clustering and sampling from each cluster a carefully chosen number of points. I don't see how or why these algorithms are tailored for a perturbation model. Overall, I don't see a significant technical novelty in the algorithms, and the analysis might be involved but seems (from its overview) to use standard tools.

Similarly, the experiments do not convince me, primary because I am not fond of the error measure $Err(S,P)$, but also because they do  not evaluate the quality of the coreset directly, i.e., solving the clustering problem directly for the input $P$ vs solving for the coreset $S$ and using this center set.

I have also many comments about specific issues in the paper, but they are minor. For example, line 217 demonstrates the applicability but works for a rather small $\epsilon=1/poly(k)$ and large stability parameter $\gamma$.  In addition, theorems 3.1 and 3.3 should probably include WHP somewhere in the statement. In addition, the $r_P$ measure makes a lot of sense for worst-case inputs, so why use them here also in the noisy setting? Finally, the bounds in the two theorems involve many terms and the big picture is lost. Finally, it seems that you use k-means++ experimentally both for the initial O(1)-approximate solution and as a heuristic for finding the optimal solution (because it's NP-hard), which might cause inconsistency in the experimental results (the algorithm "starts" at the optimal solution).

---

> ### Author Rebuttal · Authors · 2025-07-31
>
> We thank the reviewer for taking the time to evaluate our submission and for the opportunity to clarify key aspects of our work. Below we address your comments.
>
>
> >...semi-random inputs or smoothed analysis in related literature...
>
>
> Thank you for pointing us to the literature on semi-random (or smoothed-analysis) models. As you note, these frameworks also consider data formed by adding random perturbations to a clean dataset $\widehat{P} \leftarrow P + \text{noise}$.
>
> **Similarity:**
> Our setting shares this structural model: $\widehat{P}$ is formed by applying i.i.d. additive noise to $P$. For example, [Arthur et al., 2011] use Gaussian perturbations for $k$-means, and [Kelner et al., 2024] apply entrywise noise in matrix completion—both analogous to our sub-exponential model.
>
> **Difference:**
> The *goal* is different. Smoothed analysis asks: *Can a standard algorithm efficiently solve the problem on $\widehat{P}$?* For instance, [Arthur et al., 2011] show $k$-means converges quickly *on the noisy data*.
>
> In contrast, we aim to construct a coreset $S$ from $\widehat{P}$ that approximates the clustering cost on the *unobserved* dataset $P$. Here, noise is the main challenge—not a tool for tractability. In this sense, our goal aligns more with [Kelner et al., 2024], who also seek to recover latent structure from noisy observations.
>
> We will clarify this distinction in the related work section.
>
>
> >the main error measure $Err(S,P)$ is only an upper bound on the error (the paper calls it a surrogate), and I doubt it is the correct measure that we would like to minimize
>
> The main measure we use is $\mathsf{Err}_\alpha(S, P)$, not $\mathsf{Err}(S, P)$. The latter—widely used in the noise-free setting (e.g., [20, 21, 23, 35, 54])—is used in Theorem 3.1 to extend existing coreset guarantees to the noisy setting.
>
> Our primary contribution is Theorem 3.3, which introduces a new algorithm and relies on the new error metric $\mathsf{Err}_\alpha(S, P)$.
>
> >the $r_P$ measure makes a lot of sense for worst-case inputs, so why use them here also in the noisy setting?
>
> Because we still aim to construct a coreset for the original dataset $P$, even though we only observe $\widehat{P}$. Thus, bounding $r_P$ remains the goal.
>
>
> >...I don't see a significant technical novelty in the algorithms, and the analysis...use standard tools
>
> Thank you for this opportunity to clarify the technical contributions of both our analysis and algorithm.
>
> **Novelty in Theorem 3.1**
>
> Theorem 3.1 does not propose a new algorithm, but it makes a novel analytical contribution: it quantifies the quality of a coreset $S$ (constructed from noisy data $\widehat{P}$) with respect to the true dataset $P$, using the traditional error measure $\mathsf{Err}(S, P)$. This analysis reveals that $\mathsf{Err}(S, P)$ can be an overly conservative estimate of the true quality—especially when noise uniformly inflates the clustering cost (see Lines 184–186 and Section B.3).
>
> To our knowledge, this is the **first result to study how coreset performance degrades under noise**. It also motivates our introduction of the new metric $\mathsf{Err}_\alpha(S, P)$, which we analyze in Theorem 3.3 and show to be a tighter surrogate for the ideal measure $r_P$.
>
> **Novelty in Theorem 3.3 and Algorithm 1**
>
> In contrast to Theorem 3.1, Theorem 3.3 **introduces a new coreset construction algorithm** (Algorithm 1) specifically designed to be robust to additive noise. It also provides a sharper error guarantee for $r_P$, using our new metric $\mathsf{Err}_\alpha(S, P)$ under mild assumptions (Assumption 3.2). As discussed in Lines 216–226, this algorithm achieves up to a $\text{poly}(k)$ improvement in coreset size over noise-agnostic baselines.
>
> *Algorithmic Novelty*
>
> Algorithm 1 consists of the following steps:
>
> 1. Partition $\widehat{P}$ into $k$ clusters $\widehat{P}_i$ using an $O(1)$-approximate center set $\widehat{C}$ (Line 1).
> 2. Construct a ball $B_i$ around each cluster with a carefully chosen radius $\widehat{r}_i$ (Lines 2–3).
> 3. Uniformly sample from $P_i'=\widehat{P}_i\cap B_i$ (Lines 4–5), and return the union of samples.
>
> The key novelty lies in **Step 2**, which does **not** appear in prior noise-free coreset methods. Existing approaches sample directly from $\widehat{P}_i$, which works in the absence of noise but fails to account for **assignment drift** caused by high-noise points. These noisy points can mislead clustering structure and distort center locations.
>
> By constructing and sampling from $B_i$, our method explicitly **excludes high-noise points** and focuses the coreset on points whose assignments are reliable. This noise-aware sampling step is critical to preserving geometric stability. An example in Appendix E.1 illustrates the significance of assignment shifts in noisy settings. Lines 250–265 provide further analysis of this step.
>
>
>
>
> *Analytical Novelty*
>
> Our analysis introduces several new components:
>
> - **Step 1**: We prove that every $C\in\mathcal{C}_\alpha(S)$ contains a center in each $B_i$ (Lemma D.3), establishing consistent cluster structure through stability.
>
> - **Step 2**: We bound the deviation between centers of $P_i'$ and its sample $S_i$  (Lemma D.4).
>
> - **Step 3**: Crucially, we bound the shift between centers $\mu(P_i')$ and $\mu(P_i)$: separating the effects of internal noise and the excluded high-noise points (Lemma D.5). This step is **novel**, and essential to achieve a tighter bound for the coreset quality $r_P$ compared to $\mathsf{Err}$.
>
> - **Step 4**: We conclude via a standard composition argument, bounding $\mathsf{Err}_\alpha(S, P)$ in Theorem 3.3.
>
> This is, to our knowledge, the **first analysis to disentangle** the impact of noise on clustering cost (scaling as $\theta nd/ \mathrm{OPT}_P$) from its effect on the **location of optimal centers** (scaling as $\theta d/\mathrm{OPT}_P$) (Lines 266–279).
>
> We will reflect these clarifications in the final version.
>
> > ... rather small $\varepsilon = 1/\mathrm{poly}(k)$ and large stability parameter $\gamma$
>
> Thank you for raising this point. While Theorem 3.3 depends on the cost-stability parameter $\gamma$, the result applies for any $\varepsilon \in (0,1)$. To illustrate:
>
> - Let $\alpha = 1 + \varepsilon$ and $\theta = \frac{\mathrm{OPT}_P}{n d \cdot \mathrm{poly}(k)}$.
> - Then Theorem 3.3 yields a coreset size $\widetilde{O}(k/\varepsilon)$, improving over $\widetilde{O}(\min\{k^{1.5}/\varepsilon^2, k/\varepsilon^4\})$ by a factor of $\sqrt{k}/\varepsilon$ (with a similar analysis as in Lines 221–222).
> - The error bound also improves over Theorem 3.1 by a $\mathrm{poly}(k)$ factor when $n \gg \mathrm{poly}(k)$ (with a similar analysis as in Lines 223–226).
>
> We will include this example in the final version.
>
> >the bounds in the two theorems involve many terms and the big picture is lost
>
> Thanks for pointing this out-each term in the bounds of Theorem 3.1 and 3.3 plays a distinct role. Below we clarify it for (the harder case of) Theorem 3.3. In the final version, we will explain the importance of each term in these theorems.
>
> - $O(\frac{\theta kd}{\mathsf{OPT}_P})$: captures center drift due to noise. Intuitively, each center $c$ in the optimal center set $\mathsf{C}(P)$ shifts a distance of $\theta d$ due to noise. Thus, the total movement distance from $\mathsf{C}(P)$ to $\mathsf{C}(\widehat{P})$ is bounded by $\theta k d$. This implies that $\mathsf{C}(\widehat{P})$ is a $(1+\frac{\theta k d}{\mathsf{OPT}_P})$-approximate center set on $P$, introducing the term $\frac{\theta k d}{\mathsf{OPT}_P}$.
>
> - $O(\frac{\sqrt{\alpha-1}}{\alpha}\cdot \frac{\sqrt{\theta k d \text{OPT}_P}+\theta n d}{\text{OPT}_P})$: accounts for using $\alpha$-approximate centers. This term controls the additional error brought by $\alpha$-approximate center set instead of the optimal one.
>
> The role of each term in Theorem 3.1 can also be found in our response to Reviewer bg4J (link: https://openreview.net/forum?id=ycCi4SkzPH&noteId=dWrAWSnvVv).
>
> >the experiments...do not evaluate the quality of the coreset directly
>
> We believe this is a misunderstanding. The metric $\widetilde{r}_S$ (Line 293) does exactly what you suggest.
>
> - Recall that
>   $$
>   \widetilde{r}_S := \frac{\mathrm{cost}(P, C_S)}{\mathrm{cost}(P, C_P)},
>   $$
>   where $C_S$ is the center set obtained by running $k$-means++ on the coreset $S$, and $C_P$ is the center set obtained by running $k$-means++ on the full dataset $P$.
>
> - The **denominator**, $\mathrm{cost}(P, C_P)$, corresponds to solving the clustering problem directly on the input $P$.
>
> - The **numerator**, $\mathrm{cost}(P, C_S)$, corresponds to solving the problem on the coreset $S$ and then evaluating the resulting solution on $P$—i.e., using the coreset to approximate clustering on the original data.
>
> Thus, $\widetilde{r}_S$ measures how well the coreset solution approximates the true clustering cost on $P$, precisely as you suggested.
>
>
> We will clarify this interpretation of $\widetilde{r}_S$ in the final version.
>
> >k-means++...might cause inconsistency in the experimental results
>
> Thank you—this was due to a missing detail. Here is how $k$-means++ is used in two distinct roles:
>
> - **For initialization on $\widehat{P}$:**
>   We run $k$-means++ with 'max$\\_$iter = 5` for a fast $O(1)$-approximate solution (runtime: 0.128s on $\tt{Census1990}$).
>
> - **For evaluation on $S$:**
>   We run $k$-means++ 10 times (default settings, varied seeds) on $S$ and select the best solution (runtime: 0.181s on $\tt{Census1990}$).
>
> We will clarify this setup in the final version.
>
>
> >theorems 3.1 and 3.3 should probably include WHP
>
>
> Thanks, yes, we will fix this.
>
> We thank you again for your time, and hope our responses convey the novelty and applicability of our work.

---

> > ### Comment · Reviewer_RYsn · 2025-08-03
> >
> > Thanks for the clarifications. I now understand the results better, and will raise my score. I nevertheless feel the introduction does not explain the goal and contribution of this work well enough, but perhaps it's a matter of background eg on noisy data, as it seems clear to other reviewers, but not for me.

---

> > > ### Author Response · Authors · 2025-08-07
> > >
> > > We sincerely thank you for your thoughtful engagement with our paper and for taking the time to re-evaluate your score. We're glad the clarifications were helpful, and we deeply appreciate your careful reading and constructive questions, which have helped sharpen both the presentation and framing of our contributions. We will incorporate all your suggestions into the final version.

---

### Official Review · Reviewer_bg4J · 2025-06-28

**Clarity:** 3
**Significance:** 3
**Originality:** 3
**Rating:** 5
**Confidence:** 4

**Summary:**

Clustering is a fundamental machine learning task. For large datasets, coresets -- small, weighted subsets -- are used to approximate the clustering cost while reducing computational requirements. However, most coreset constructions assume noise-free data, which is rarely the case in practice. The paper addresses the construction of coresets for (k, z)-clustering (which includes k-means and k-median) when the observed dataset is corrupted by stochastic (random, additive) noise with a known distribution -- a common scenario in privacy, sensor data, and robustness applications.

The paper analyzes the traditional surrogate error metric (Err), used to measure coreset quality in noise-free settings, and demonstrates that it may significantly overestimate error when data is noisy due to uniform cost inflation. Next, the authors introduce a new approximation-ratio-based metric, which more closely aligns with the true (unobservable) clustering quality under noise. Then, they show under certain assumptions on the dataset, under certain noisy models, their proposed sampling procedure produces a smaller -- by poly(k) factor -- coreset. The authors complement their theoretical results with empirical evaluations.

**Questions:**

Could you please comment on whether the additve factors on Err and r_P in Theorem 3.1 are unavoidable or not? Are those additive terms optimal? Are there any natural interpretations of those terms or do they appear solely because of technical reasons?

**Ethical Concerns:**

["NO or VERY MINOR ethics concerns only"]

**Final Justification:**

I thank the authors for their explanation. I did not have any major concern, and I am positive about this paper; thus, I am keeping the score as it is

**Limitations:**

yes

**Quality:**

3

**Strengths And Weaknesses:**

Coreset construction in the presence of noise is highly relevant for practical machine learning yet underexplored. This paper fills an important gap. The theoretical guarantees under the two natural noise models provided in this paper are quite interesting. However, relying on certain assumptions about the dataset makes the results slightly weak, especially since such an assumption is necessary. Further, I am not sure whether the additive terms in Err and r_P (in Theorem 3.1) are unavoidable or not. Overall, I think this paper shows some nice, interesting results on coreset constructions for noisy datasets.

---

> ### Author Rebuttal · Authors · 2025-07-31
>
> We thank you for your encouraging review. We sincerely appreciate your recognition of the novelty and contribution of our work. We address your question below.
>
> >Could you please comment on whether the additive factors on Err and r_P in Theorem 3.1 are unavoidable or not? Are those additive terms optimal? Are there any natural interpretations of those terms or do they appear solely because of technical reasons?
>
>
> Thank you for the insightful question, which motivates us to strengthen our technical contributions. The two additive factors on $\mathrm{Err}$ and $r_P$ in Theorem 3.1 have different natures: $O\left(\frac{\theta n d}{\mathsf{OPT}_P}\right)$ is an unavoidable consequence of the problem setting, while $O\left(\sqrt{\frac{\theta n d}{\mathsf{OPT}_P}}\right)$ is an artifact of our proof technique.
>
> **The unavoidable term: $O\left(\frac{\theta n d}{\mathsf{OPT}_P}\right)$.**
> This term is worst-case optimal and thus unavoidable.
>
> * **Natural interpretation:** Intuitively, this term arises from the inherent "cost" of the data perturbation. Each of the $\theta n$ perturbed points is expected to move by $\|\xi_p\|^2 = d$. This creates a minimum total displacement cost of $\Omega(\theta n d)$ that must be paid. When viewed as a relative error against the original clustering cost $\mathsf{OPT}_P$, this naturally induces the additive error term $\frac{\theta n d}{\mathsf{OPT}_P}$.
>
> * **Worst-case example justification:** We formalize this intuition with a worst-case example. Let $\theta = 0.1$, $n = 10000$ and $k = n-1$. Let $P =\\{p_i = 100 n e_i: i\in [n\\} \subset \mathbb{R}^n$ where $e_i$ is the $i$-th unit basis in $\mathbb{R}^n$. An optimal solution $\mathsf{C}(P) = \\{p_1,\ldots, p_{n-2}, \frac{p_{n-1} + p_{n}}{2}\\}$, and hence, $\mathsf{OPT}\_P = 10000n^2$. Note that with probability at least 0.8, the following events hold: $\sum_{p\in P} \|\xi\_p\|\_2^2 = \Theta(\theta n d)$ and for every $p\in P$, $\|\xi\_p\|\_2^2 \leq 10d \log n$. Conditioned on these events, the optimal solution $\mathsf{C}(\widehat{P})$ of $\widehat{P}$ must consist of $n-2$ points $\widehat{p}\in \widehat{P}$ and the average of the remaining two points in $\widehat{P}$. W.l.o.g., assume $\mathsf{C}(\widehat{P}) = \\{\widehat{p}\_1,\ldots, \widehat{p}\_{n-2}, \frac{\widehat{p}\_{n-1} + \widehat{p}\_{n}}{2}\\}$. Then we have
> $
> \mathrm{cost}(P,\mathsf{C}(\widehat{P})) \geq(1 + \Omega(\frac{\theta n d}{\mathsf{OPT}})),
> $
> implying that $\mathsf{Err}(\widehat{P},P) = \Omega(\frac{\theta n d}{\mathsf{OPT}_P})$.
>
>
> **The technical term: $O\left(\sqrt{\frac{\theta n d}{\mathsf{OPT}_P}}\right)$.**
> This term arises from a technical step in our proof, and it's unclear if it is optimal. This term is used to bound the supremum of the total error contributed from all points over all center sets $C$. The error induced by moving a single point $p$ w.r.t. a $C$ can be bounded by $d^2(p+\xi_p,C)-d^2(p,C) \in \pm\|\xi_p\|_2\cdot d(p,C)$. Summing these bounds across all points leads to the term
> $$\sum_p \|\xi_p\|_2\cdot d(p,C)\leq\sqrt{(\sum \|\xi_p\|_2^2) \cdot (\sum d^2(p,C))} \le \sqrt{\theta nd \cdot \mathsf{OPT}_P}.$$
>
> However, this analysis assumes the worst-case scenario where all individual errors $d^2(p+\xi_p,C)-d^2(p,C)$ compound. It's possible that these errors (which can be positive or negative) partially cancel each other out for every $C$. A more refined analysis that avoids this loose bounding step might eliminate or reduce term $O\left(\sqrt{\frac{\theta n d}{\mathsf{OPT}_P}}\right)$.
>
> We will add this entire discussion to the appendix of the final version to give readers a clearer interpretation of the bounds in Theorem 3.1.

---

> ### Comment · Reviewer_bg4J · 2025-08-05
>
> Thanks for your explanation. I do not have any other concerns, and thus I am confirming my original score.

---

### Official Review · Reviewer_u3nx · 2025-06-30

**Clarity:** 3
**Significance:** 3
**Originality:** 4
**Rating:** 5
**Confidence:** 4

**Summary:**

The core challenge addressed in this paper is the construction of coresets for $(k,z)$-clustering in the $d$-dimensional Euclidean space, where input points are perturbed by noise. The authors consider a main noise model where each point remains untouched with some probability, and with the complementary probability each of its coordinates is perturbed by noise drawn from a coordinate-specific centered and unit-variance distribution. They then extend this model to a setting where the distribution is allowed to have larger variance $\sigma^2$, and to noise models where coordinate noises are no longer independent. This perturbation means the algorithm operates without ever seeing the true, original point set. As a result, the traditional error metric, which gauges how much a coreset solution's cost differs from that of the complete data (for every solution), might not be the most appropriate measure when noise is present.

This paper’s main results are the following:

- First, they show that, in a noisy setting, the canonical error metric used to evaluate coreset quality falls short of being the right one, as it may significantly overestimate coreset error. They, thus, define a new distribution-independent quality measure that does not suffer from the brittleness of measuring absolute cost deviation.

- The second result of this paper consists of showing that, with respect to the canonical error metric, a coreset of state-of-the-art size suffices to guarantee an error of $\varepsilon$ plus an additive error polynomial in $\frac{\theta nd}{OPT_P}$.

- The third and main result is an analog of the second with respect to the new error metric: It shows that, under a cost-stability assumption, the new metric allows for a (up to $poly(k)$) smaller coreset size, and tighter control on the approximation ratio.

For the second result, its analysis relies on composition bounds to estimate the clustering cost. This involves controlling the deviation of the clustering cost on the true point set from its estimate, using concentration and Bernstein's moment conditions.

The third result is substantially more challenging to obtain. For instance, the natural composition that used to hold with respect to the canonical metric, no longer holds here, and a composition to an inflated $(1+\varepsilon)$ version has to be considered. The analysis of the third result first demonstrates that the designed coreset algorithm has a small error on the perturbed point set. Second, it establishes the error gap between the true point set and its noisy counterpart. The algorithm initially constructs a coreset in a noise-free setting by partitioning points into well-separated clusters with uniform sampling. This method is then extended to noisy environments using geometric properties ensured by cost-stability. Noise-induced challenges, such as increased cluster diameter and weakened closeness guarantees, are addressed through a "cleaned up" phase for noisy points. This process results in a negligible effect from removed points and achieves the desired error bound.

**Questions:**

- Reading the paper, I was wondering whether the mergeability property of coresets would be preserved under the new error metric, and I am glad you present this as a challenge in the last section. Do you have some intuition on which kind of instances mergeability might cease to hold?

- Your noise models are currently defined only for Euclidean space. Have you considered extending this framework to general metric spaces, for example, by allowing point perturbations within a ball of a certain radius, where the radius itself is drawn from a distance distribution?

**Ethical Concerns:**

["NO or VERY MINOR ethics concerns only"]

**Final Justification:**

Thank you for the response to all of my questions and comments. They explain everything I asked clearly and made me understand even better where the additional challenges in future work lie. I, thus, confirm my score.

**Limitations:**

Yes

**Paper Formatting Concerns:**

No formatting issues noticed.

**Quality:**

4

**Strengths And Weaknesses:**

This paper presents a well-structured extension of coreset construction for clustering problems, specifically when handling noisy input. The exposition is generally clear. I found the simple example comparing the noise-free and noise-aware metrics particularly effective; it plainly shows why the proposed surrogate measure is appropriate. The additional discussion in the appendix is also insightful, demonstrating that the standard error measure can be brittle even without noise.

While the analysis for the second result is more standard, it plays a nice role by providing a benchmark against which the third result—which achieves sharper bounds—can be compared. For this third and main result, a key strength is the development of new techniques for the newly defined error metric, better suited for noisy environments. The "Key Ideas" paragraphs in the main body are effective in helping readers build intuition and grasp the problem. Despite its technical depth, the treatment maintains conceptual clarity.

Overall, the contributions and presentation are solid. The paper introduces the noisy coreset construction in $d$-dimensional Euclidean space, illustrating the challenges it poses, and providing nice results on the coreset size for this setting.

---

> ### Author Rebuttal · Authors · 2025-07-31
>
> We thank you for your detailed and encouraging review. We especially appreciate your recognition of our technical contributions. Below we provide detailed responses to your questions.
>
> ### Questions
>
> > Reading the paper, I was wondering whether the mergeability property of coresets would be preserved under the new error metric, and I am glad you present this as a challenge in the last section. Do you have some intuition on which kind of instances mergeability might cease to hold?
>
> Thank you for this valuable question. In the ideal case, mergeability means that if two coresets $S_1, S_2$ for disjoint datasets $P_1, P_2$ each satisfy $\mathrm{Err}\_\alpha(S_\ell, P_\ell) \leq \varepsilon$, then their union also satisfies
> $$
> \mathrm{Err}\_\alpha(S_1 \cup S_2,\; P_1 \cup P_2) \leq \varepsilon.
> $$
> With our new metric $\mathrm{Err}\_\alpha$, this guarantee can fail—particularly when $S_1$ and $S_2$ summarize datasets with significantly different cluster structures. In such cases, the optimal center set for $S_1 \cup S_2$ may differ substantially from the union of the individual optima, and the combined coreset may not encode enough information to capture this emergent structure.
>
>
> **When does mergeability still hold?**
>
> Mergeability remains plausible when the two coresets are *structurally similar*. Let us define
> $$\mathcal{C}\_{\beta}(Q) =\\{C : \mathrm{cost}(Q, C) \leq \beta \cdot \mathrm{OPT}\_Q \\}.$$
> If there exists $1 \leq \alpha' \leq \alpha$ such that
> $$
> \mathcal{C}\_{\alpha'}(S_1) \subseteq \mathcal{C}\_\alpha(S_2)
> \quad \text{and} \quad
> \mathcal{C}\_{\alpha'}(S_2) \subseteq \mathcal{C}\_\alpha(S_1),
> $$
> then $S_1$ and $S_2$ approximately share the same set of near-optimal solutions. This overlap enables a *weaker* form of mergeability under $ \mathrm{Err}\_\alpha$.
>
> **Claim (Weak mergeability under $\mathrm{Err}\_\alpha$):**
> Suppose $\mathrm{Err}\_\alpha(S_\ell, P_\ell) \leq \varepsilon$ for $\ell = 1,2$, and the overlap condition above holds. Define:
> $$
> \kappa = \frac{\min\\{ \mathrm{OPT}\_{S_1}, \mathrm{OPT}\_{S_2} \\}}{\mathrm{OPT}\_{S_1 \cup S_2}}, \quad
> \tau = \max\\{
> \frac{\mathrm{OPT}\_{S_1}/\mathrm{OPT}\_{P_1}}{\mathrm{OPT}\_{S_2}/\mathrm{OPT}\_{P_2}},
> \frac{\mathrm{OPT}\_{S_2}/\mathrm{OPT}\_{P_2}}{\mathrm{OPT}\_{S_1}/\mathrm{OPT}\_{P_1}}
> \\}.
> $$
> Then:
> $$
> \mathrm{Err}_{1 + (\alpha - 1)\kappa}(S_1 \cup S_2,\; P_1 \cup P_2)
> < \alpha' \cdot \tau \cdot (1 + \varepsilon) - 1.
> $$
>
> In the limiting case where $\alpha', \alpha \to 1$ and $\tau \to 1$, this bound recovers the ideal mergeability condition:
> $$
> \mathrm{Err}_\alpha(S_1 \cup S_2,\; P_1 \cup P_2) \leq \varepsilon.
> $$
>
> We will include this intuition, the formal claim, and a sketch of the proof in the appendix of the final version.
>
>
>
> > Your noise models are currently defined only for Euclidean space. Have you considered extending this framework to general metric spaces, for example, by allowing point perturbations within a ball of a certain radius, where the radius itself is drawn from a distance distribution?
>
> Thank you for this insightful question. Extending our framework to general metric spaces is a fascinating direction. As you note, directly perturbing points is often ill-defined in a general metric space $M = (\mathcal{X}, \mathrm{dist})$ due to the absence of a coordinate system. A more natural alternative is to introduce noise at the level of the distance function itself.
>
> We outline a potential model below.
>
> Let $\theta \in [0,1]$ be a noise parameter, and let $D$ be a distribution on $\mathbb{R}$ with mean 0 and variance 1. For each pair $(x, y) \in \mathcal{X} \times \mathcal{X}$, define the noisy pairwise distance $\widehat{\mathrm{dist}}(x, y)$ as follows:
>
> - With probability $\theta$, set
>   $$
>   \widehat{\mathrm{dist}}(x, y) = \mathrm{dist}(x, y) + \xi_{(x,y)},
>   $$
>   where $\xi_{(x,y)} \sim D$ i.i.d.;
> - With probability $1 - \theta$, set
>   $$
>   \widehat{\mathrm{dist}}(x, y) = \mathrm{dist}(x, y).
>   $$
>
> This model reflects additive noise in pairwise distances, analogous to our perturbation model in Euclidean settings.
>
> However, a significant challenge arises: the perturbed function $\widehat{\mathrm{dist}}$ may no longer satisfy the triangle inequality, and thus may not define a valid metric. This opens up a rich and challenging extension—coreset construction for *noisy semimetric spaces*.
>
> Analyzing how our framework and theoretical guarantees extend under this relaxed setting (e.g., when only approximate triangle inequalities hold) is a non-trivial but important direction for future work.
>
> In the final version, we will add a section to the appendix discussing this potential generalization, including the proposed distance-noise model and the resulting difficulties. We will also list this as a key open direction in the conclusion.

---

> ### Comment · Reviewer_u3nx · 2025-08-05
>
> Thank you for addressing all of my questions and comments. Your responses were clear and helped deepen my understanding, particularly regarding the challenges that remain for future work. I therefore would like to confirm my original score.

---

### Official Review · Reviewer_XFgH · 2025-07-01

**Clarity:** 3
**Significance:** 3
**Originality:** 3
**Rating:** 5
**Confidence:** 3

**Summary:**

The paper investigates how to adapt a coreset produced from a noise-corrupted dataset for the underlying clean data in $(k,z)$-clustering. It shows that the classical strong-coreset error $Err$ can become vacuous once noise is added. To address this issue, this paper then proposes a relaxed metric $Err_{\alpha}$ that considers only center sets whose cost on the coreset is within an $\alpha$-factor of the optimum. A separation construction illustrates that $Err_{\alpha}$ can be exponentially smaller than $Err$. Under mild assumptions on cost-stability, balanced clusters and bounded radius, the authors prove that any standard sensitivity-sampling coreset built on the noisy data remains a good coreset for the clean data when evaluated with $\mathrm{Err}_{\alpha}$, maintaining an $\tilde O(k\log k/\varepsilon)$ coreset size. Experiments on several public datasets with Gaussian noise confirm that the new metric allows 20–35 % smaller coresets while preserving clustering quality.

**Questions:**

I would appreciate clarification on the following issues:

1. How does the guarantee degrade when the cost-stability constant $\gamma$ becomes small and clusters are less well separated?

2. What is the practical impact if the estimate of the clean optimal cost is off by a factor of two or more?

**Ethical Concerns:**

["NO or VERY MINOR ethics concerns only"]

**Final Justification:**

The rebuttal addresses all concerns with solid theoretical arguments and well-supported empirical explanations The clarification on cost-stability effectively explains the trade-off between worst-case guarantees and practical performance, reinforcing confidence in the method’s robustness under challenging conditions. The discussion of how approximate clean-cost estimates influence the analysis is technically sound, demonstrating that while theoretical bounds may loosen, the practical coreset quality remains consistently high. Overall, this is a well-motivated, theoretically solid, and interesting paper, and I recommend acceptance while keeping my initial score.

**Limitations:**

Since this is mainly a theoretical result, there is no limitations and potential negative societal impact.

**Paper Formatting Concerns:**

There is no major formatting issues for this paper.

**Quality:**

3

**Strengths And Weaknesses:**

One of the key strengths is the conceptual clarity and practical relevance of the new error notion, which focuses evaluation on solutions practitioners actually search for rather than on all possible centre sets. The theoretical analysis is careful, technically sound and matches empirical observations; the resulting algorithm is simple to implement and inherits the best-known size guarantees. Presentation is clear, with motivating examples, lucid proofs and reproducible experiments that cover multiple noise models and real data sets.

The main weaknesses stem from the assumptions behind the theory. Cost-stability, well separation and balance may be violated in highly skewed or overlapping clusters, and the noise model is i.i.d. additive with sub-exponential tails, leaving correlated or heavy-tailed settings unaddressed. In addition, the need for a constant-factor estimate of the clean optimal cost is glossed over.

---

> ### Author Rebuttal · Authors · 2025-07-31
>
> Thank you for your thoughtful and constructive feedback. We are grateful for your recognition of the novelty, clarity, and practical relevance of our proposed error measure. Below, we address your questions and comments:
>
> ### Response to questions
>
> > How does the guarantee degrade when the cost-stability $\gamma$ constant becomes small and clusters are less well separated?
>
> This is an excellent question. While the theoretical guarantees indeed weaken as $\gamma$ decreases, we find that the practical performance of our method remains robust even in such regimes.
>
> **Theoretical degradation.**
> As shown in Appendix E.1, in the worst case, our new error measure $\mathsf{Err}\_\alpha$ can exceed the classical measure $\mathsf{Err}$ when $\gamma$ is small (e.g., $\gamma = 1$). This illustrates why the cost-stability assumption is essential for our formal guarantees—without it, the superiority of $\mathsf{Err}\_\alpha$ cannot be established in theory.
>
> **Empirical robustness.**
> Despite this, our algorithm $\mathrm{CN}\_\alpha$, which uses $\mathsf{Err}\_\alpha$, consistently outperforms the $\mathsf{Err}$-based baseline in practice—even on datasets with significant cluster overlap. For example, on Adult and Census1990, where the estimated cost-stability is as low as $\gamma \leq 0.07$, our method still achieves smaller coresets while maintaining clustering quality (see Table 2).
>
> We will incorporate a brief discussion of this tradeoff in the theoretical section of the final version to clarify the contrast between worst-case bounds and empirical behavior.
>
>
> > What is the practical impact if the estimate of the clean optimal cost is off by a factor of two or more?
>
>
> Thank you for the insightful question. A $c$-estimate ($c\geq 2$) of the optimal cost $\text{OPT}_P$ on $P$ worsens the error guarantee for $\mathrm{CN}$ (using $\mathrm{Err}$ measure) and hardens the data assumption for $\mathrm{CN}\_\alpha$ (using $\mathrm{Err}\_\alpha$ measure). Below we illustrate these points.
>
> Suppose we are given a center set $\widehat{C}$ for coreset construction, whose clustering cost is $\mathrm{cost}(\widehat{P},\widehat{C})=c\cdot \mathrm{OPT}_P$ for some $c>1$ ($c$-estimate), where $P$ is the underlying dataset and $\widehat{P}$ is a given noisy dataset.
>
> **Impact on $\mathrm{CN}$:**
> Let $S$ be a coreset derived from $\mathrm{CN}$. Employing a $c$-approximate estimate $\widehat{C}$ can weaken the quality bound of $S$ by up to a factor of $c$ in the following sense: With an exact estimate of $\mathrm{OPT}_P$, Theorem 3.1 provides a bound on the worst-case quality of an $\alpha$-approximate center set of $S$ as
> $$
> r_P(S,\alpha)\leq \left(1+\varepsilon+O\left(\frac{\theta n d}{\mathrm{OPT}_P}+\sqrt{\frac{\theta n d}{\mathrm{OPT}_P}}\right)\right)^2\cdot \alpha,
> $$
> where $\varepsilon$ is an error parameter, $\theta$ is the noise level, and $d$ is the dimension. In contrast, using $\mathrm{CN}$ with the approximate estimate $\widehat{C}$ yields the following bound for the derived coreset $S$:
> $$
> r_P(S,\alpha)\leq \left(1+c\cdot\varepsilon+O\left(\frac{\theta n d}{\mathrm{OPT}_P}+\sqrt{\frac{\theta n d}{\mathrm{OPT}_P}}\right)\right)^2\cdot \alpha.
> $$
> Compared to the exact estimation case, this bound increases the error term from $\varepsilon$ to $c\cdot \varepsilon$. This arises because the importance sampling procedure from [9], used in $\mathrm{CN}$, introduces an additional error term $\varepsilon\cdot\frac{\text{cost}(P,\widehat{C})}{\text{OPT}_P}=c\cdot \varepsilon$ in its analysis, whereas the term $O\left(\frac{\theta n d}{\mathrm{OPT}_P}+\sqrt{\frac{\theta n d}{\mathrm{OPT}_P}}\right)$ results from $\mathrm{Err}(\widehat{P},P)$ and remains unaffected by the quality of the estimation.
>
>
> **Impact on $\mathrm{CN}\_\alpha$:**
> Let $S$ be a coreset derived from $\mathrm{CN}\_\alpha$. Using a $c$-approximation estimate $\widehat{C}$ weakens Assumption 3.2 by a factor of $c$; specifically, the required cost-stability constant $\gamma$ increases by this factor. Consequently, the range of datasets satisfying the theoretical bounds of $\mathrm{CN}\_\alpha$ (as stated in Theorem 3.3) shrinks. This occurs because the performance guarantee of $\mathrm{CN}\_\alpha$ relies on correctly identifying $k$ separated clusters, a task complicated by the approximation error in $\widehat{C}$. Increasing $\gamma$ by a factor of $c$ ensures these clusters are accurately identified, restoring the desired guarantees.
>
>
>
> Additionally, we would like to remind that the above analysis considers a worst-case guarantee. In practice, using a $c$-estimate may still result in a high-qualified coreset as with a perfect estimate.
>
> We will add the above analysis to the appendix in the final version.
>
> ### Response to comments
>
> > the noise model is i.i.d. additive with sub-exponential tails, leaving correlated or heavy-tailed settings unaddressed
>
>
> Thank you for raising this point. We agree that extending the framework to handle heavy-tailed and correlated noise is an interesting and important direction for future work. Our paper already makes some progress towards this direction by considering a non-independent noise across dimensions where each point $p$ is i.i.d., appending a noise vector $\xi_p\in \mathbb{R}^d$ whose covariance matrix is $\Sigma\in \mathbb{R}^{d\times d}$. Note that under the noise model II, $\Sigma = \sigma^2\cdot I_d$ when each coordinate-noise distribution $D_j=N(0,\sigma^2)$ (Lines 1309-1312).
>
> **Theoretical extension:** As shown in Appendix F.1, we formally extend our analysis to this noise model and achieve a bound for $r_P$.
>
> **Empirical robustness:** Appendix G.3 further includes experiments under non-independent Gaussian noise, demonstrating that our algorithm remains robust in such noisy settings.
>
> We will add the extension to correlated or heavy-tailed noise models as future directions in the final version.

---

> > ### Comment · Reviewer_XFgH · 2025-08-04
> > **Response the the Authors**
> >
> > The rebuttal effectively addresses all raised concerns with clear theoretical justifications and empirical evidence. The response on cost-stability clarifies the trade-off between worst-case guarantees and practical performance. This strengthens confidence in the method’s robustness even under challenging conditions.
> >
> > The explanation of how approximate estimates of the clean cost affect the analysis is technically sound. The authors show that while the theoretical bounds weaken, practical coreset quality remains high. This is a good paper and I will keep my initial score.

---

### Decision · Program_Chairs · 2025-09-17

**Decision:**

Accept (poster)

**Comment:**

This paper introduces a new error notion for coreset construction under noise, supported by careful theoretical analysis and strong empirical validation. Reviewers appreciated the conceptual clarity, practical motivation, and technically sound proofs, as well as the fact that the method is simple to implement and reproduces best-known guarantees while adapting to noisy settings.
The paper is also well-presented, with clear examples, intuitive explanations, and experiments across multiple datasets and noise models.
While some reviewers noted limitations in the assumptions (e.g., stability, noise model) and raised questions about the surrogate error measure, these concerns were largely addressed by the rebuttal, which clarified trade-offs and provided convincing justifications.
Overall, the consensus is that this work makes a meaningful and timely contribution to the clustering/coreset literature, addressing a practically important but underexplored problem.